

# Detection of slow changes in terrestrial water storage with GRACE and GRACE-FO satellite gravity missions

Julia Pfeffer[1], Anny Cazenave[1,2], Alejandro Blazquez[2,3], Bertrand Decharme[4], Simon Munier[4], Anne Barnoud[1]

[1] Magellium, Ramonville-Saint-Agne, 31520, France
[2] LEGOS, Université de Toulouse, Toulouse, 31400, France
[3] CNES, Toulouse, 31400, France
[4] CNRM/Météo France/CNRS, Toulouse, 31057, France

*Correspondence to*: Julia Pfeffer (julia.pfeffer@magellium.fr)

**Abstract.** The GRACE (Gravity Recovery And Climate Experiment) and GRACE Follow-On (FO) satellite gravity missions enable global monitoring of the mass transport within the Earth's system, leading to unprecedented advances in our understanding of the global water cycle in a changing climate. This study focuses on the quantification of changes in terrestrial water storage based on an ensemble of GRACE and GRACE-FO solutions and two global hydrological models. Significant changes in terrestrial water storage are detected at pluriannual and decadal time-scales in GRACE and GRACE-FO satellite gravity data, that are generally underestimated by global hydrological models. The largest differences (more than 20 cm in equivalent water height) are observed in South America (Amazon, Sao Francisco and Parana river basins) and tropical Africa (Congo, Zambezi and Okavango river basins). Significant differences (a few cm) are observed worldwide at similar time-scales, and are generally well correlated with precipitation. While the origin of such differences is unknown, part of it is likely to be climate-related and at least partially due to inaccurate predictions of hydrological models. Slow changes in the terrestrial water cycle may indeed be overlooked in global hydrological models due to inaccurate meteorological forcing (e.g., precipitation), unresolved groundwater processes, anthropogenic influences, changing vegetation cover and limited calibration/validation datasets. Significant differences between GRACE satellite measurements and hydrological model predictions have been identified, quantified and characterised in the present study. Efforts must be made to better understand the gap between both methods at pluriannual and decadal time-scales, which challenges the use of global hydrological models for the prediction of the evolution of water resources in changing climate conditions.

## 1 Introduction

The GRACE (Gravity Recovery And Climate Experiment; Tapley et al., 2004) and GRACE Follow-On (GRACE-FO; Landerer et al., 2020) missions provide spatio-temporal observations of the gravity field spanning over two decades, sensitive



to the redistribution of masses from the deep Earth's interior to the top of the atmosphere (e.g., Chen et al., 2022). The GRACE and GRACE-FO satellite observations have been widely used to estimate changes in terrestrial water storage (TWS), expressed in equivalent water heights, representing mass anomalies as a layer of water of variable thickness in space and time located at the Earth's surface (e.g., Wahr et al., 1998). Changes in TWS range from a few millimetres to a few ten centimetres from arid (e.g., deserts) to humid (e.g., tropical rain forests) regions of the world, and are dominated first by seasonal changes, then by

decadal changes (e.g., Humphrey et al., 2016). Locally (mostly along the Amazon River), seasonal TWS variations can reach up to 1 or 2 metres. Decadal trends in TWS have been attributed to climate variability (e.g., change in precipitation), direct human impacts (e.g., irrigation) and the combination of both effects (Rodell et al., 2018). Significant groundwater depletion has for example been observed in the Central Valley (California), in response to two extreme and prolonged droughts intensified by groundwater pumping for agriculture, wetland management and domestic use (e.g., Scanlon et al., 2012; Ohja

et al., 2018).

Trends in TWS are often temporary due to climate variability (e.g., Alam et al., 2021) and changes in water consumption policies (e.g., Bhanja et al., 2017). Significant interannual TWS variations detected in large river basins have been attributed to a combination of eight major climate modes, including the El Niño-Southern Oscillation (ENSO), Pacific Decadal

Oscillation, North Atlantic Oscillation, Multidecadal Atlantic Oscillation and Southern Annular Mode (e.g., Pfeffer et al., 2022). Successive droughts and floods events have been associated with a succession of positive (El Niño) and negative (La Niña) phases of ENSO in various regions of the world, such as Australia, Southern Africa or parts of the Amazon River basin (e.g., Ni et al., 2018, Anyah et al., 2018, Xie et al., 2019). Drought (e.g., Thomas et al., 2017) and flood potential (e.g., Sun et al., 2017) indices using GRACE and GRACE-FO observations have been developed to monitor the impact of extreme events

on freshwater resources, taking into account all climatic and anthropogenic mechanisms and all water reservoirs from the surface to deep aquifers.

Beyond monitoring the TWS variability, GRACE and GRACE-FO data have widely been used to constrain poorly observed components of the water mass balance. Typically, TWS changes (dTWS/dt) can be expressed as:

$$\frac{dTWS}{dt} = P - ET - R \qquad (1)$$

and used to constrain the terrestrial water discharge (R) based on independent estimates of the net precipitation (precipitation P minus evapotranspiration ET), with good agreement with available in situ river gauges (e.g., Syed et al., 2009 and 2010). Alternatively, groundwater storage (GWS) variations can be estimated as the difference between the TWS changes estimated from GRACE observations and ice, snow, surface water and soil moisture variations estimated from independent data sources (e.g., Chen et al., 2016; Frappart et al., 2018). These approaches often rely on global hydrological models, land surface models

or land surface reanalyses, to estimate one or several terms of the water mass balance equation, assuming that the water fluxes (e.g., net precipitation, see for example Chandanpurkar et al., 2017) and water storage anomalies from the ice, snow, surface





and soil reservoirs (e.g., Rodell et al., 2007; Bhanja et al., 2016; Thomas and Famiglietti, 2019; Frappart et al., 2019) are modelled with sufficient accuracy, so that the residual gravity signal can be attributed to the variable of interest (i.e. terrestrial freshwater discharge or GWS changes).

If the spatial and temporal variability of TWS is generally well captured, global hydrological models and land surface models tend to underestimate the amplitude of seasonal signals (e.g., Döll et al., 2014) and decadal trends (e.g., Scanlon et al., 2018) when compared to GRACE and GRACE-FO observations. The differences in TWS between satellite gravity observations and model predictions have been shown to depend on the choice of models and river basin considered (e.g., Döll et al., 2014; Wada

et al., 2014; Scanlon et al., 2018; Scanlon et al., 2019; Decharme et al., 2019; Yang et al., 2020 and Felfelani et al., 2017). Seasonal changes in TWS are often underestimated by hydrological and land surface models in tropical, arid and semi-arid basins, and overestimated at higher latitudes in the Northern hemisphere, likely due to insufficient surface and ground water storage estimates in tropical basins, and to a misrepresentation of evapotranspiration and snow physics at higher latitudes (Scanlon et al., 2019). Some models lead to better performance in heavily managed river basins and, on the contrary, to

erroneous trends and seasonal cycles in regions where the natural variability is dominant (e.g., Wada et al., 2014; Scanlon et al., 2019; Felfelani et al., 2017). The performance of models also varies during the recharge and discharge periods, suggesting that some processes (e.g., reservoir operation) may be adequately captured by a model, while other processes (e.g., groundwater dynamics) may be overlooked (Felfelani et al., 2017). The reasons for discrepancies between models and satellite gravity observations remain largely unknown, though improvements in the parameterization of global hydrological and land surface

models are often recommended to reliably predict spatial and temporal changes in TWS, especially regarding aquifers (e.g., Decharme et al., 2019, Scanlon et al., 2019, Felfelani et al., 2017).

This study focuses on the comparison of two global hydrological models, ISBA-CTRIP (Decharme et al., 2019) and WGHM (Müller Schmied et al., 2021), against GRACE-based TWS observations at interannual and decadal time-scales. While the

seasonal variations in TWS have been extensively studied (e.g., Döll et al., 2014; Wada et al., 2014; Scanlon et al., 2019; Decharme et al., 2019 and Felfelani et al., 2017), little attention has been paid to longer time-scales, often only estimated as linear trends (Scanlon et al., 2018; Felfelani et al., 2017). Significant non-linear variability occurs however at interannual time-scales, that may lead to considerable stress on water resources and large uncertainties on climate model projections. Besides, the same model may have different performances at seasonal, interannual and decadal time-scales, as different processes

prevail at such different time scales (e.g., Scanlon et al., 2018, Scanlon et al., 2019; Felfelani et al., 2017). This study will therefore quantify and characterise the amplitude of TWS at interannual and decadal time-scales for 9 GRACE solutions (3 mascon solutions and 6 spherical harmonic solutions) and 2 global hydrological models between April 2002 and December 2016. The differences in TWS will be compared to the dispersion in GRACE solution to evaluate their significance and to precipitation to better understand their origin. Such assessment will allow evaluating the performance of hydrological models



used in CMIP6 (e.g., Voldoire et al., 2019) projections, ISI-MIP (e.g., Herbert & Döll, 2019) projections and in value-added products based on a synergy of satellite data and models.

## 2 Methods

### 2.1 Satellite gravity data

Total terrestrial water storage (TWS) changes have been estimated using the latest release of three mascon solutions from the
JPL (RL06 Version 02, Wiese et a., 2019), CSR (RL06 V02; Save et al., 2016 and Save, 2020) and GSFC (RL06 V01, Loomis et al., 2019a) and six solutions based on spherical harmonic coefficients of the gravitational potential from the JPL (RL06, GRACE-FO, 2019a; Yuan, 2019), CSR (RL06, GRACE-FO, 2019b; Yuan, 2019), GFZ (RL06, Dahle et al., 2018), ITSG (GRACE2018, Mayer-Gürr et al., 2018), COST-G (RL01, Meyer et al., 2020) and CNES-GRGS (RL05, Lemoine and Bourgogne, 2020). The same corrections for the geocenter (Sun et al., 2016), $C_{20}$ coefficients (Loomis et al., 2019b) and GIA
(ICE6G-D by Peltier et al., (2018)) have been applied for mascon and spherical harmonic solutions. The Stokes coefficients from the JPL, CSR, GFZ, ITSG, COST-G and CNES-GRGS solutions, with the aforementioned corrections applied, have been truncated at degree 60, converted to surface mass anomalies expressed as equivalent water height (cm) and projected on the WGS84 ellipsoid using the locally spherical approximation (eq. 27 in Ditmar et al., 2018) implemented in the l3py python package (Akvas, 2018). Systematic errors (i.e., stripes) have been removed from spherical harmonic solutions (except for the
constrained CNES-GRGS solutions) using an anisotropic filter based on the principle of diffusion (Goux et al., 2022), using Daley length scales of 200 and 300 km in the North-South and East-West directions, and a shape of Matern function close to a Gaussian (8 iterations). The diffusive filter allows the conservation of mass within the continental domain, defined here as grid cells where at least 30% of the altitudes from ETOPO1 Global Relief Model (NOAA National Geophysical Data Center, 2009) are above sea level. Small islands ($<100\,000$ km$^2$) have been excluded from the continental domain, because of the
limited spatial resolution of monthly GRACE products (a few hundred kilometres). By default, the GRACE-derived TWS anomalies used in this study is the average of the nine processed GRACE solutions. The uncertainty on GRACE-based TWS anomalies is estimated as the dispersion (minimum to maximum) between the 9 GRACE solutions.

### 2.2 Global hydrological models

Total terrestrial water storage (TWS) TWS changes have also been estimated using the ISBA-CTRIP (Interaction Soil
Biosphere Atmosphere - CNRM (Centre National de Recherches Météorologiques) version of Total Runoff Integrating Pathways) global land surface modelling system (Decharme et al., 2019) and the version 2.2d (Müller Schmied et al., 2021) of the WaterGap Global Hydrological Model (WGHM) including glaciers.

ISBA solves the water and energy balance in the soil, canopy, snow and surface water bodies, and CTRIP simulates discharges
through the global river network, as well as the dynamic of both the seasonal floodplains and the unconfined aquifers. ISBA



and CTRIP are coupled through the land surface interface SURFEX, allowing complex interactions (e.g., floodplain free water evaporation, and upwards capillarity fluxes between groundwaters and superficial soils) between the atmosphere, land surface, soil and aquifer. ISBA-CTRIP is forced at a 3-hourly timestep with the ERA-Interim atmospheric reanalysis (Dee et al., 2011) for air temperature and humidity, wind speed, surface pressure and total radiative fluxes, and with the gauge-based Global Precipitation Climatology Center (GPCC) Full Data Product V6 (Schneider et al., 2014) for precipitation.

WGHM 2.2d simulates changes in water flows and storage using a vertical mass balance for the canopy, snow and soil and a lateral mass balance for the surface water bodies and groundwater (Müller Schmied et al., 2021). WGHM is coupled with a global water use model, taking into account water impoundment in artificial reservoirs and regulated lakes and water withdrawals for irrigation, livestock, domestic use, manufacturing and thermal power (Müller Schmied et al., 2021). Anthropogenic water withdrawals/impoundments are assumed to only impact surface waters and groundwaters (Müller Schmied et al., 2021). In addition, water storage changes in continental glaciers have been simulated with the Global Glacier Model (Marzeion et al., 2012) and added as an input to WaterGap (Caceres et al., 2022). The WGHM uses meteorological input data from WFDEI (Weedon et al., 2014) also based on the ERA-Interim atmospheric reanalysis for air temperature and solar radiation and GPCC for precipitation. Two model variants are available using different irrigation efficiencies (optimal and 70% of optimal) (Döll et al., 2014b). Both being equally plausible given the limited datasets available to characterise groundwater abstractions for irrigation, we averaged the two variants in the present study.

### 2.3 Lake data

Lake water storage anomalies have then been added to the predicted TWS anomalies from ISBA-CTRIP and WGHM. Indeed, although WGHM2.2d includes artificial and natural lakes in its framework, large differences were observed between the observed and predicted TWS anomalies around large lakes (e.g., American and African Great Lakes, Caspian Sea, Volta Lake), that were greatly reduced with the application of a lake correction (Appendix A).

Changes in lake volume were estimated for 100 lakes during the whole GRACE period from the hydroweb database (https://hydroweb.theia-land.fr/), based on a combination of lake level measurements from satellite altimetry and lake area measurements from satellite imagery (e.g., Cretaux et al., 2016). Then lake volume changes are converted into equivalent water heights (m) over a regular 1x1 degree grid, using the GLWD (Global Lakes and Wetlands Database) shapes for lakes larger than 5000 km$^2$ as detailed in Blazquez et al. (in preparation).

### 2.4 Precipitation data

Precipitation is estimated using the gauge-based Global Precipitation Climatology Center (GPCC) Full Data Product V6 (Schneider et al., 2014) and the IMERG (Integrated Multi-satellitE Retrievals for GPM) data product (Huffman et al., 2019)



based on the TRMM (Tropical Rainfall Measuring Mission: 2000-2015) and GPM (Global Precipitation Measurement: 2014 - present) satellite data.

## 2.5 Data processing

The period of common availability for all datasets spans from April 2002 (first estimation of TWS changes with GRACE data) to December 2016 (latest estimation of TWS changes with WGHM data). All time-series have been monthly averaged. Months with missing data are excluded from all datasets, leaving 141 valid months between April 2002 and December 2016. All dataset were interpolated to a regular 1°x1° grid using the conservative algorithm from xESMF (Zhuang et al., 2020), allowing to preserve the integral of the surface mass anomalies across the grid conversion (i.e., the water mass anomaly over a 1°x1° grid cell is equal to the area-weighted average of the mass anomalies from overlapping cells in the source grid). Because this study focuses on interannual to decadal changes in total terrestrial water storage, regions where observed mass changes are known to be dominated by other processes have been masked. These include the oceans, ice-covered regions such as Antarctica, Greenland, Arctic islands, and regions impacted by very large earthquakes (Sumatra, Tohoku, Maule) defined by Tang et al. (2020). Seasonal signals have been removed by least-squares adjustment of annual and semi-annual sinusoids. Finally, to be able to compare higher-resolution hydrology products to GRACE-based TWS anomalies, a diffusive filter with an isotropic Daley length of 250 km has been applied to all products. In the following, we refer to the fully processed time-series as TWS anomalies. Residual TWS anomalies (sometimes shortened as residuals) refer to the difference between the TWS anomalies estimated with the average GRACE solution and the TWS anomalies estimated with one of the two global hydrological models considered in this study (either ISBA-CTRIP or WGHM). The amplitude of the interannual variability is expressed as the range at 95% CL of fully processed TWS anomalies. The range at 95% CL is defined as the difference between the 97.5 and 2.5 percentiles. It provides a more accurate quantification of the amplitude of the non-seasonal TWS variations than the RMS.

## 3 Results

### 3.1 Comparison of observed and predicted TWS anomalies

TWS anomalies (Fig. 1) are globally lower in hydrological models (Fig. 1c and d) than in GRACE solutions (Fig. 1a), leaving large residuals in GRACE satellite data (Fig. 1e and f). The underestimation of TWS anomalies is more acute with WGHM (Fig. 1d) than with ISBA (Fig. 1c). Significant (> 5 cm) residual TWS anomalies (Fig. 1e and f) are observed in South America (Amazon, Orinoco, Sao Francisco and Parana river basins), Africa (Congo and Zambezi basins), Australia (northern part of the continent), Eurasia (India, North European Plains, Ural Mountains, Siberian Plateau) and North America (Colorado Plateau, Rocky Mountains). Very large (≥ 30 cm) residual TWS anomalies are observed around glaciers (Alaska, Patagonia) due to ice-melting, which is not the concern of the present study. In most regions of the world, the residual TWS anomalies



are significantly larger (5th, 50th and 95th percentiles of the RMS of residual TWS anomalies at 4, 8 and 20 cm) than the uncertainty on GRACE data estimated by the dispersion among the 9 solutions (5th, 50th and 95th percentiles of the standard deviation between the 9 GRACE solutions at 1, 3 and 13 cm). The largest ($\geq 5$ cm) dispersion values are observed in coastal and mountainous regions, or in regions with very large ($\geq 20$ cm) residuals (Fig. 1b). Larger sources of errors are indeed expected near the coast in GRACE measurements due to leakage errors, making the interpretation of residual signals difficult in islands such as Madagascar or the Indonesian Archipelago. Similarly significant ice-melt from glaciers occurs in mountainous regions such as the Alaska or Tibetan plateau, which constitutes the most likely explanation for the large residuals observed in these regions. However, this does not constitute the topic of the current study. We therefore exclude islands and glaciers from the results and discussion. Larger dispersion values should however not prevent the discussion of the results in regions where very large residual TWS anomalies are observed, if the observed signals are several times larger than the estimated uncertainty.

To be able to differentiate a systematic underestimation of TWS anomalies from singular differences in the spatial and temporal variability, we computed the range ratio between the average GRACE solution and each hydrological model. For most regions of the world (Fig. 2a and 2b), the range of TWS anomalies is larger for GRACE than for ISBA-CTRIP or WGHM, except in East Canada (Ontario, Quebec, Newfoundland), North Asia (East Siberia, Ob River, Finland/Northwest Russia) and central Africa (Cameroun, Gabon, Congo). In these regions, the coefficient of determination ($R^2$) between the GRACE and the hydrological models is typically negative (Fig. 2c and d), indicating that the variance of the residuals is larger than the variance of GRACE data. The global hydrological models ISBA-CTRIP and WGHM are therefore not able to predict the TWS variability estimated from GRACE satellite data in these regions.

The large residuals observed with ISBA-CTRIP in the North-West of South America (Fig. 1e) are due to differences in the spatial and temporal variability of observed and predicted TWS changes. The range of TWS variations is indeed larger for ISBA-CTRIP than for GRACE in this region. $R^2$ values are relatively high (0.5-0.9) at the North of the Amazon, indicating important similarities between GRACE and ISBA-CTRIP. To the contrary, $R^2$ values are very low ($< 0.3$) at the South of the Amazon, indicating significant differences between GRACE and ISBA-CTRIP.

The range of TWS anomalies is smaller for hydrological models than for GRACE over most of the study area (76% for ISBA-CTRIP and 83% for WGHM). TWS anomalies predicted by hydrological models are underestimated by at least 50% over almost half of the study area (40% for ISBA-CTRIP and 49% for WGHM). TWS anomalies are at least two times smaller than GRACE for 22% of the study area for ISBA-CTRIP and 25% for WGHM. The largest range ratios ($> 5$) are reached across deserts (Sahara, Arabian Peninsula, Gobi Desert) and glaciers (Alaska, Patagonia, Himalaya). Such differences are due to numerical artefacts (denominator near zero) and non-hydrological signals (ice melting) observed by GRACE. Very large range ratios (2-4) are also observed for ISBA-CTRIP across the United States (Great Plains aquifer) and the North of India, because





of significant anthropogenic influences in these regions. Large range ratios (from 2 to 5) are reached in tropical and subtropical regions of the Southern hemisphere (Africa, South-America, Australia) for WGHM.

Over more than half of the study area (61% for ISBA-CTRIP and 53% for WGHM), global hydrological models explain a minor part ($R^2<0.5$) of the variance of the TWS anomalies estimated with the average GRACE solution (Fig. 2c and 2d). By comparison with GRACE, WGHM is more performant in the Northern than Southern hemisphere. Relatively large $R^2$ values (> 0.5) are reached in the United States of America, central and North Europe, West and central Siberia, Eastern Asia, North of India, Caspian Sea and Arabian Peninsula (Fig. 2d). Large $R^2$ values are also reached over most of South America (Fig. 2d).

Lower $R^2$ values (< 0.5) are reached over most of the African and Australian continents, and parts of the Northern (North Canada, central Asia, Eastern Siberia, South India) hemisphere (Fig. 2d). By comparison (Fig. 2c), ISBA-CTRIP is more performant ($R^2>0.5$) in the Southern hemisphere (North, Central and East Australia, South and East Africa, South-America except Peru, Bolivia and Patagonia) and parts of the Northern hemisphere (Eastern US, South Canada, central and North Europe, South of Siberia, Caspian Sea, South of India, East China). Lower $R^2$ values (< 0.5) are reached for ISBA-CTRIP in

North Canada, West and Central Africa, Arabian Peninsula, South and central Asia and West Australia (Fig. 2c). Both models exhibit negative $R^2$ values in central and Sahelian Africa, as well as in Quebec and Ontario (Fig. 2c and 2d). For ISBA-CTRIP, negative $R^2$ coefficients are also reached in North Bolivia, Alaska, North of India and Siberia (south of Lena River). For WGHM, negative $R^2$ coefficients are reached in the central US and South India. These metrics indicate that for some regions of the world (not necessarily the same for both models), hydrological models are able to capture a large part of the TWS

variability estimated from GRACE, but that, overall, significant differences exist between global hydrological models and GRACE satellite data.

## 3.2 Characteristic time scales of residual TWS anomalies

The differences in TWS anomalies estimated from GRACE and global hydrological models (or residual TWS anomalies) are largely dominated by pluri-annual and decadal signals (Fig. 3). Residual TWS anomalies have been separated into sub-annual,

pluri-annual and decadal contributions using a high-pass (cut-off period at 1.5 years), band-pass (cut-off periods at 1.5 and 10 years) and low-pass (cut-off period at 10 years) filters respectively. The percentage of variance explained by each contribution has been calculated as $R^2$ values and reported in Maxwell's colour triangle (Fig. 3). Residual TWS anomalies are dominated by decadal signals over a large part of the study area (51% with ISBA-CTRIP and 40% with WGHM), including Alaska, West Canada, Brazilian highlands (Sao Francisco and Parana river basins), Patagonia, West (Niger and Volta river basins) and South

Africa (Okavango and Zambezi river basins), parts of West (Arabic Peninsula, Caspian Sea drainage area, Tigris/Euphrates, Dnieper, Volga and Don river basins), central (Tibetan Plateau, and Tarim, Ganges and Brahmaputra river basins) and North (Yenisei and Lena river basins) Asia, and East Australia. When calculating the residuals with ISBA-CTRIP, large decadal





signals are also observed across North-West America (Sierra Madre, Sierra Nevada, Great Basin, Rocky Mountains) and the North of India (Indus River basin).


Pluriannual signals are prevalent in residual TWS anomalies across central Africa, West Australia, Siberia (Ob and Yenisei), Eastern Europe, North-East America (Great Lakes) and the Southwest of the Amazon basin. Subannual signals are prevalent in regions with tenuous TWS variability (i.e., Sahara, South Africa, Southwest Australia), likely pointing out the remaining level of noise in GRACE data (Fig. 1b). Regions with large (≥ 10 cm) residual TWS anomalies (Fig. 1e), are systematically

dominated by pluri-annual to decadal contributions (Fig. 3).

Residual TWS anomalies are dominated by slow changes in the TWS, including linear trends and non-linear signals (Fig. 4). Though significant linear trends are detected (+/- 1 cm/yr), residual TWS anomalies are mainly due to non-linear variability in the TWS (Fig. 4). Apart from glaciers, significant trends in TWS residuals are observed in West (Niger) and South

(Okavango and Zambezi) Africa, North-East Australia, South Asia (mostly the North of India, especially when using ISBA-CTRIP), Northwest America (ISBA-CTRIP only) and central US (mainly WGHM). Part of the residual TWS trends observed with ISBA-CTRIP in Northwest America and South Asia are likely due to anthropogenic influences. In other regions of the world, residual trends in TWS are likely related to climate variability (South Africa, Northeast Australia) or land-use changes (West Africa). In most regions of the world (72% of the study area for ISBA-CTRIP and 83% for WGHM), the residual

variability in TWS cannot be explained by a linear trend and involves significant variability at interannual and decadal time scales (Fig. 4c to 4f).

## 4 Discussion

To better characterise and understand the nature of residual TWS anomalies, TWS anomalies estimated from GRACE and global hydrological models have been averaged over large regions of the world and compared to in-situ and satellite

precipitation. In the following, we discuss regional TWS anomalies where the largest residuals are observed around the central Amazon corridor, the upper Sao Francisco River, the Zambezi and Okavango rivers, the Congo River, the North of Australia, the Ogallala aquifer in central USA, the North of the Black Sea and the Northern Plains in India (see map in Fig B1 - Appendix B). For each of these regions, all the solutions of the GRACE ensemble (3 mascon and 6 spherical harmonic solutions) detect slow changes in TWS, which indicates high confidence in these observations. Larger differences occur between ISBA-CTRIP

and WGHM, and both models systematically underestimate the pluri-annual and decadal changes in TWS captured by GRACE. Part of these differences may be attributed to common sources of errors in GRACE-based TWS estimates, including errors in background models (for example, the atmospheric circulation model) and post-processing choices (for example, the GIA model). However, errors in the atmospheric model (GAA from AOD1B, based on ERA5) would be associated with fast changes in TWS, while errors in the GIA model (ICE6G-D) would be characterised by linear trends over the GRACE period.



Here, the largest differences between GRACE and global hydrological models occur at pluri-annual and decadal time scales, and are generally well correlated with precipitation. A large part of the differences between GRACE and global hydrological models are therefore likely to be climate-related and at least partially due to inaccurate predictions of global hydrological models. Similar regional analyses have been done for the 40 largest river basins of the world with comparable results (Appendix C).

**4.1 Central Amazon Corridor**

The central Amazon corridor (1°N-7°S and 75°W-50°W) surrounds the Solimões-Amazon mainstream river, and the downstream parts of its main tributaries, including the Japura, Jurua, Purus, Negro, Madeira, Trombetas, Tapajos and Xingu rivers. Those large rivers exhibit a monomodal flood pulse lasting several months, flooding an extensive lowland area, largely covered by forests, called varzea or igapo depending on the river water colour (respectively white waters rich in sediments or

black waters rich in organic matter) (e.g., Junk et al., 1997; Melack and Coe, 2021). The extension of the flooded area varies from 100 000 to 600 000 km² in the Amazon basin (e.g., Fleishmann et al., 2022), in phase with water level variations in rivers that can reach up to 15 m annually (e.g., Birkett et al., 2002; Alsdorf et al., 2007; Frappart et al., 2012; Da Silva et al., 2012), with significant interannual variability (e.g., Fassoni-Andrade et al., 2021). Heterogeneous soils distributions, including ferralsols, plinthosols and gleysols (e.g., Quesada et al., 2011), lie over unconsolidated sedimentary rocks, alluvial deposits

and consolidated sedimentary rocks with relatively homogeneous hydraulic properties (e.g., Gleeson et al., 2011; Fan et al., 2013). Across the central Amazon lowlands, the groundwater table fluctuates by several metres (Pfeffer et al., 2014), corresponding to groundwater storage changes of several tens of centimetres (Frappart et al., 2019), which constitutes a large part of the TWS changes observed by GRACE (Frappart et al., 2019).

Over the central Amazon region (Fig. 5), TWS anomalies predicted by global hydrological models agree well with GRACE observations, with very large Pearson coefficients reached both for ISBA-CTRIP (R=0.90) and WGHM (R=0.86). The amplitudes of TWS anomalies predicted with ISBA-CTRIP match closely GRACE solutions, while WGHM tends to underestimate the TWS variability at interannual and decadal time scales, which is likely due to a more accurate representation of the floodplains and their interactions with the atmosphere, soil and aquifer with ISBA-CTRIP than WGHM (Fig. 5d).

Interannual variability occurs in the precipitation as well (Fig 5a and b), with significant correlation with GRACE (R=0.54), ISBA (R=0.59) and WGHM (R=0.64) and a phase lag of 1 month. Despite good performances for both models (especially ISBA-CTRIP), significant residual signals remain in TWS anomalies after correction of hydrological effects, consisting mostly of an increasing trend with ISBA-CTRIP, with significant interannual variability superimposed for WGHM. The residual TWS changes corrected with WGHM are still significantly correlated with precipitation (R=0.48) with a phase lag of 4 months. No

significant correlation can be found between the residual TWS anomalies calculated with ISBA and precipitation anomalies



(maximum R value of 0.22 with a time lag of 14 months), though significant decadal and pluri-decadal variability can be observed in GPCC precipitation records, that may explain a residual trend in TWS (~ 5 mm/yr).

Residual TWS anomalies may be due to inaccurately modelled water storage variations in any reservoir from the surface to the aquifer. The largest residual TWS variations are observed along the downstream part of the Solimoes, at the confluences with the Purus and the Rio Negro, which is a region that is largely covered by floodplains (e.g., Fleishmann et al., 2022) and dominated by changes in surface water storage (Frappart et al., 2019). The long time-scales associated with the residuals and increasing time-lags with precipitation suggest however a significant contribution from groundwater storage fluctuations, that are insufficiently constrained in global hydrological models (e.g., Decharme et al., 2019, Scanlon et al., 2018 and 2019). Large floodplains may indeed delay the water transport for several months (e.g., Prigent et al., 2020), through storage and percolation from the surface towards the aquifer (e.g., Lesack & Melack, 1995; Bonnet et al., 2008; Frappart et al., 2019). Groundwater stores excess water during wet periods and sustains rivers and floodplains during low-water periods (e.g., Lesack, 1993). Groundwater systems have also been shown to convey seasonal anomalies (for example, droughts) for several years at local (e.g., Tomasella et al., 2008) and regional (Pfeffer et al., 2014) scales. Such memory effects may be underestimated by global hydrological models, which would result in much faster variations of the TWS.

## 4.2 Upper Sao Francisco

The Sao Francisco River, located in North-East Brazil, is 3200 km long and drains an area of about 630 000 km$^2$. Hydroelectric dams located along the Sao Francisco provide about 70% of Northeast Brazil electricity, including the Três Marias, Sobradinho and Luíz Gonzaga (Itaparica) reservoirs with respective volumes of 15,278 hm$^3$, 28,669 hm$^3$ and 3,549 hm$^3$. Significant decreases in the river flow during the 1980–2015 period have been attributed to increased groundwater withdrawals sustaining irrigated agriculture and decreasing the groundwater contributions to streamflow (i.e., baseflow) (Lucas et al., 2020). As a result of a prolonged drought lasting from 2002 to 2017 (Freitas et al., 2021), the Sao Francisco hydroelectric plants only provided a minor part (from 18 to 42% depending on the year) of the total electricity demand, which was sustained by increased fossil fuel consumption (de Jong et al., 2018). A decrease in TWS was also observed from 2012 to the end of the GRACE mission (mid-2017) across the Sao Francisco coincident with the observed rainfall deficit (Ndehedehe and Ferreira, 2020), allowing to better quantify the impact of prolonged droughts on the water supply in a vulnerable region (Paredes-Trejo et al., 2021).

Over the upper Sao Francisco region (Fig. 6), TWS anomalies predicted with global hydrological models are well correlated with GRACE data on a year-to-year basis (R=0.79 for ISBA and R=0.81 for WGHM). The maxima and minima in TWS associated with wet and dry years are well picked up by satellite observations and models, though the amplitude of TWS anomalies is underestimated by hydrological models. All 9 GRACE solutions exhibit an interannual signal, with a peak at a





period ~ 6 years, and a decadal oscillation, with a drop in terrestrial water storage from 2012 to 2016 (Fig. 6b), not predicted

by ISBA-CTRIP or WGHM, but corresponding to 4 years of consecutive deficit in precipitation (Fig. 6a). As a consequence,

residual TWS anomalies (Fig. 6e), characterised by prominent interannual and decadal signals (Fig 6f), reach 10-20 cm in the

Sao Francisco region. TWS anomalies predicted by hydrological models are relatively well correlated with precipitation

(R=0.6 for ISBA and 0.52 for WGHM) with a time lag of 1 month, while the correlation with GRACE TWS anomalies is more

marginal (R=0.39 with a time lag of 1 month). Residual TWS anomalies are also only marginally correlated with precipitation

(R=0.29 for GRACE-WGHM and 0.33 for GRACE-ISBA), with a time lag of 3 months.


These results tend to show that global hydrological models reproduce quite well the year-to-year variability of TWS anomalies

across the Sao Francisco (especially in term of occurrence of a wet/dry anomaly, as the amplitudes of the anomalies may be

underestimated), but struggle to predict slower hydrological processes characterised by interannual and decadal time scales.

### 4.3 Zambezi - Okavango

The Zambezi River basin, located in South tropical Africa, drains an area of 1 400 000 km² connecting Angola (18.3 %),

Namibia (1.2 %), Botswana (2.8 %), Zambia (40.7 %), Zimbabwe (15.9 %), Malawi (7.7 %), Tanzania (2.0 %) and

Mozambique (11.4 %) (Vörösmarty and Moore III, 1991). It encompasses humid, semi-arid and arid regions dominated by

seasonal rainfall patterns associated with the Inter-Tropical Convergence Zone (ITCZ), with a wet season spanning from

October to April and a dry season spanning from May to September (Lowmann et al., 2018). The Zambezi basin harbours very

large wetland areas and lakes, whose extension considerably varies with precipitation at seasonal and interannual time scales

(Hugues et al., 2020). Significant interannual variability in the precipitation and TWS have been detected over the Zambezi

and Okavango regions, and attributed to several climate modes, including the Pacific Decadal Oscillation, Atlantic

Multidecennal Oscillation and El Niño Southern Oscillation (Pfeffer et al., 2021).

Across the Zambezi and Okavango region (Fig. 7), TWS anomalies are well correlated with precipitation (R=0.62 and 0.49

with a time lag of 1 month for ISBA-CTRIP and WGHM), as years with a positive (respectively negative) precipitation

anomaly correspond to a local maximum (respectively minimum) in TWS. This year-to-year variability is consistent with all

9 GRACE solutions, as evidenced by a Pearson correlation coefficient of 0.60 and 0.63 between the average GRACE solution

and both hydrological models. However, the TWS anomalies estimated from all 9 GRACE solutions exhibit a strong decadal

oscillation with a minimum in 2005/2006 and a maximum in 2011/2012, that is not picked up by hydrological models, leaving

a strong (20 cm in amplitude) decadal anomaly in the residuals TWS estimated with the difference between GRACE and global

hydrological models. Though the residual TWS anomalies are poorly correlated with the precipitation anomaly (R=0.23 and

0.25 with a phase lag of 28 and 40 months for GRACE - ISBA and GRACE - WGHM respectively), they are strongly related

to the accumulated precipitation anomalies, also exhibiting a strong decadal anomaly with a minimum in 2005/2006 and a





maximum in 2011/2012. The TWS residuals can be reduced locally by up to 50% in the Zambezi region by applying an

empirical model based on climate modes, as formulated by Pfeffer et al., (2021). The main modes of variability found in the

TWS residuals are the Pacific Decadal Oscillation and the Atlantic Multidecennal Oscillation.

## 4.4 Congo

The Congo basin is the second largest river basin in the world, with a drainage area of ~ 3.7 $10^6$ km² and an average annual

discharge of ~ 40 500 m³s$^{-1}$ (Laraque et al., 2020). Despite its importance, the Congo River basin is scarcely studied (Alsdorf

et al., 2016), though a growing interest arose over the past decade, substantially due to advances in satellite hydrology (e.g.,

Papa et al., 2022, Paris et al., 2022, Schumann et al., 2022). With an average rainfall around 1500 mm$^{-1}$, the Congo basin

benefits from a humid tropical climate with a complex seasonal migration of rainfall across the basin with a first maximum in

November-December and a second peak in April-May (Alsdorf et al., 2016) leading to a bimodal river discharge (Kitambo et

al., 2022). The "Cuvette centrale" is a topographic depression located at the centre of the basin, harbouring wetlands covered

by rainforests permanently or periodically flooded (Becker et al., 2018). The Congo floodplain hydrodynamics are

disconnected from the main river, with much less variability observed throughout the year (Alsdorf et al., 2016). The Congo

River basin hosts a large complex fractured sedimentary aquifer, with relatively low storage but high recharge rates (Scanlon

et al., 2022). Very little is known about the groundwater storage variability, though comparisons of satellite estimations

(including satellite imagery and satellite altimetry) of the surface water storage with terrestrial water storage changes from

GRACE, suggest that most (~ 90% at annual time scales) of the variability in water storage occurs under the surface (Becker

et al., 2018).

Non-seasonal TWS anomalies are very different over the Congo basin depending on the method of estimation considered. All

9 GRACE solutions are consistent one with another, but differ from both global hydrological models that also exhibit large

discrepancies one with another. The correlations of TWS anomalies with precipitation are also marginal (maximum correlation

of 0.5 with WGHM). All 9 GRACE solutions exhibit a 6-year cycle, in phase with accumulated precipitation with local minima

in 2006 and 2012 and a local maxima in 2003, 2009 and 2015. Slow changes in TWS observed with GRACE are not predicted

by hydrological models, leaving large residuals in TWS characterised by a ~6-year cycle. Significant power is found in multi-

decennal precipitation time series at similar periods, ranging from 5 to 8 years (Laraque et al., 2020), as well as in discharge

times series at 7.5 and 13.5 years (Labat et al., 2005). The variability of the TWS cannot be explained by major climate modes

over the Congo River basin, except for the PDO, which may slightly influence the TWS variability at the North of the Congo

River (Pfeffer et al., 2022). The variability in river discharge has been found to be temporarily consistent with NAO at 7.5

years (from the 1970s to the 1990s) and 35 years (from the 1940s to the 1990s) (Labat et al., 2005). Part of the inaccuracies in

global hydrological models may be due to the scarcity of in-situ data available to constrain forcing such as the precipitation

(Figure 2 in Laraque et al., 2020), errors in flows such as runoff or evapotranspiration, or unresolved underground processes, including for example preferential flow along faults (Figure 1 in Garzanti et al., 2019).

### 4.5 North Australia

The climate of Northern Australia is characterised by a wet season lasting from November to April, subject to intense
thunderstorms and cyclones, with virtually no precipitation during the remainder of the year (Smith et al., 2008). Annual streamflow is highly dominated by monsoon rainfall, with dry season flows fed by groundwater discharge, that may stop for several months for a large number of rivers (Petheram et al., 2008; Smerdon et al., 2012). Groundwater plays an essential role in Northern Australia as it sustains rivers and vegetation, through baseflow and water uptake for plant transpiration (Lamontagne et al., 2005; O Grady et al., 2006). Significant interannual variability, principally related to ENSO in the North
of the continent, has been observed in rainfall (Cai et al., 2011; Sharmila et al., 2020), river discharge (Chiew et al., 1998; Ward et al., 2010) and terrestrial water storage (Xie et al., 2019).

During the GRACE era, Australia encountered a prolonged drought from 2002 to 2009, sometimes referred to as the 'millennium drought' or 'big dry', immediately followed by intensely wet conditions in 2010-2011 (the 'big wet' associated
with La Nina) and a sustained drought, leading to another dry El Nino event in 2015 (Figure 3 in Xie et al., 2019 and Figure 9 in the present manuscript). Three major climate modes (ENSO, IOD and SAM) are necessary to explain the water storage variability across Australia, but the Northern part of the country is dominated by ENSO (Xie et al., 2019).

Across North Australia, TWS anomalies predicted by global hydrological models are well correlated with precipitation
(R=0.73 and 0.67 with a phase lag of 1 month for ISBA and WGHM) and TWS anomalies estimated with GRACE (R=0.76 and 0.71 with ISBA and WGHM respectively). The amplitude of annual extreme events (for example La Niña in 2011) from ISBA matches GRACE estimates, while WGHM tends to underestimate the response of TWS to both dry (2005) and wet (2011) events. The main difference between TWS estimations from global hydrological models and GRACE solutions is the pace at which TWS return to average conditions after a wet/dry event. For example, after the flooding events associated with
La Niña 2011, all 9 GRACE solutions estimate a slow decrease of the TWS returning to average conditions in about two years. On the other hand, both global hydrological models predict a sharp decrease of the TWS returning to average conditions in about 6 months. As a consequence, a positive TWS anomaly remains in the residuals after La Niña event to account for the difference in the velocity of changes in TWS.

These results are consistent with the findings of Yang et al., (2020), who found that except for the CLM-4.5 model, hydrological models underestimated the GRACE-derived TWS trends across Australia, due to inaccurately modelled contributions from soil moisture and groundwater storage. Similarly, TWS anomalies from GRACE were found to be a better




link between vegetation change and climate variability than precipitation (Xie et al., 2019), because they convey more information about water availability in the soils and aquifers, especially when associated with SMOS measurements (Tian et al., 2019).

## 4.6 Central USA: Ogallala aquifer

The Ogallala, or High Plains, aquifer covers a surface area of about 450 000 km$^2$ across 8 states in the central USA, including parts of Colorado, Kansas, Nebraska, New Mexico, Oklahoma, South Dakota, Texas, and Wyoming. The Ogallala aquifer region supports about 20% of the wheat, corn and cotton production in the USA (Houston et al., 2013). Groundwater abstractions for irrigation began in Texas in the 1930s (Luckey et al., 1981) and exceeded recharge over much of the central and southern parts of aquifer in the 1950s (Luckey and Becker, 1999), resulting in substantial decline of the groundwater table in the Southern and Central High Plains, while the Northern High Plains stayed in balance or replenished (Haacker et al., 2016). At current depletion rates, a large part of irrigation (about 30%) may not be supported in the coming decades (Scanlon et al., 2012, Haacker et al., 2016, Steward et al., 2016, Deines et al., 2020).

In the central USA region, centred on the Ogallala aquifer, all GRACE solutions exhibit a series of upwards and downwards trends in TWS with a regular increase from mid-2006 to mid-2011, a sharp decrease in TWS from mid-2011 to 2013, followed by another increase in TWS from early 2013 to 2016. This pattern is linked with precipitation anomalies that were mainly in excess over 2006-2011, in deficit over 2011/2013 and oscillated around average values over 2013-2016, with a remarkably rainy year in 2014. This succession of opposite trends is not predicted by global hydrological models, though WGHM does predict a sharp decrease in TWS from mid-2011 to 2013. The TWS anomalies during 2006-2011 are much more constant in spite of abundant precipitation. Such differences might be explained by an overestimation of water abstractions by WGHM, which would result in almost constant TWS changes, while precipitation, and subsequent aquifer recharge, is increasing. This assumption is supported by the work of Rateb et al. (2020), showing that global hydrological models such as WGHM or PCR-GLOBWB tend to overestimate groundwater depletion due to human intervention in the region. Good agreement is found with in-situ observations of the groundwater table, though large uncertainties affect (i) the decomposition of the GRACE-based TWS anomalies into individual water reservoirs due to the vadose zone (Brookfield et al., 2018) and (ii) hydraulic parameters of the aquifer such as the conductivity or specific yield (Seyoum and Milewski, 2016). In that case, GRACE data may help to characterise the model parameters, such as irrigation efficiencies. In its current stage, the ISBA-CTRIP model is not adapted to estimate TWS changes in heavily managed regions, because it does not take irrigation into account.

## 4.7 North of India

The North of India hosts the Indus, Ganges and Brahmaputra river basins, with an average annual rainfall of 545, 1088, 2323 mm/yr respectively (e.g., Bhanja et al., 2016). The average population density ranges from 26-250 persons/km$^2$ in the





Northwest of India to over 1000 persons/km$^2$ in the Northeast of India (Dangar et al., 2021). India is the largest groundwater
user in the world, with an annual withdrawal of 230 -km$^3$ for irrigation, used essentially for rice, wheat, sugarcane, cotton and
maize cultures (Mishra et al. 2018, Xie et al., 2019). High abstraction rates largely exceeding precipitation rates have been
reported in Northwest India, in particular in the Punjab region, leading to an aquifer depletion rate of about 1m/yr (Mishra et
al. 2018; Dangar et al., 2021).

Because WGHM takes into account irrigation, predicted TWS anomalies match closely GRACE observations (R=0.96),
leaving residuals of about +/- 2.5 cm, which is about 4 to 6 times less than across the central Amazon or Zambezi regions. As
expected in strongly anthropized regions, ISBA-CTRIP fails to recover the TWS changes estimated with GRACE,
characterised by a clear decreasing trend (-7.71 +/- 0.71 mm/yr) over 2002-2016, clearly due to groundwater abstractions for
irrigation. Numerous studies have reported a good agreement between in situ groundwater level measurements and GRACE
TWS measurements in the North of India (e.g., Bhanja et al., 2016; Dangar et al., 2021). Detailed studies indicated that better
model performances could be gained by adjustment of several model parameters (water percolation rate, crop water stress,
irrigation efficiency, soil evaporation compensation and groundwater recession) against GRACE data (Xie et al., 2019). Such
information is critical to ensure the reliability of hydrological models across several regions. For example, the ISBA-CTRIP
model exhibit better performances than WGHM when compared to GRACE across the Indian Southern Peninsular Plateau
(Figure 1), because of an overestimation of groundwater abstractions in WGHM, leading to spurious decreasing trends, not
observed by satellite gravity measurements (Appendix D). An increase in TWS and replenishment of groundwater resources
has indeed been reported in South India from the analysis of GRACE and wells data (e.g., Asoka et al., 2017; Bhanja et al,
2017).

### 4.8 North of the Back Sea

The Black Sea Catchment hosts a population of 160 million people in 23 countries drained by major rivers including the
Danube, Dniester, Dnieper, Don, Kuban, Sakarya, and Kizirmak. The annual precipitation varies from less than 190 mm/yr at
the Northeast of the catchment (Russia) to more than 3000 mm/yr at the West (South Austria, Slovenia, Croatia)
(Rouholahnejad et al., 2014 and 2017). The annual average temperature varies from 2 to 7°C at the North of the catchment
(East European Plains at the border of Ukraine, Belarus and Russia), with a local minimum (<-3°C) in the Krasnodar region
(Southwest Russia) to over 15°C at the South of the Catchment (North of Turkey) (Rouholahnejad et al., 2014 and 2017). Land
use in the Black Sea Catchment is dominated by agriculture (Rouholahnejad et al., 2014 and 2017).

Large TWS residuals are observed in the Northeast of the Black Sea Catchment, in the East European plains crossing Ukraine,
Belarus and Russia. Large (~ 20 cm) TWS changes are observed by GRACE satellites in this region, characterised by a
decreasing trend at decadal time scales conjugated with significant interannual variability, with a peak at 6-7 years. Such TWS



changes are not predicted by hydrological models, leaving large (~15 cm) TWS residuals, dominated by decadal and interannual variability.

Due to rising temperatures, a generalised drop (10-15%) in solid precipitation has been observed across the East European Plain, partially offset by liquid precipitation, except along the Northern coast of the Black and Azov Sea (drop ~ 10%), the lower Volga River Basin (drop ~ 20%) and the Dvina River Basin further North (drop ~ 25%) (Kharmalov et al., 2020). A drop in summer precipitation, together with an increase in temperature, was observed at the North of the Black, Azov and Caspian Sea, generating severe drought conditions in the region (Kharmalov et al., 2020). Water scarcity has indeed become a critical concern, with increased water stress and decreased water availability, observed today and predicted to increase in the future (Rouholahnejad et al., 2014 and 2017).

## 5 Conclusion

Over most (> 75%) of continental areas, non-seasonal TWS anomalies are underestimated by the global hydrological models ISBA-CTRIP and WGHM when compared to GRACE solutions. While both hydrological models agree relatively well with GRACE observations on short time scales (i.e., good detection of abnormally dry or wet years), they systematically underestimate slower changes in TWS observed by GRACE satellites occurring on pluri-annual to decadal time-scales. Particularly large (15 - 20 cm) residual TWS anomalies are observed across the North-East of South America (Orinoco, Amazon and Sao Francisco basins), tropical Africa (Zambezi and Congo rivers basin) and North Australia. In such remote areas, better performances are reached with ISBA-CTRIP than WGHM, owing to the detailed representation of hydrological processes in a natural environment. However, the TWS predicted with ISBA-CTRIP still lack amplitude at pluri-annual and decadal time-scales leaving large linear (Amazon) and nonlinear (Sao Francisco, Zambezi, Congo, North Australia) trends in the residuals. The comparison of global hydrological models against GRACE data does not allow the identification of the processes responsible for these discrepancies, that could originate from any reservoir from the surface to deep aquifers. However, long time-scales associated with the residuals, combined with increasing time-lags and decreasing correlations with precipitation, suggest at least some mismodelled contributions from the groundwater cycle. Aquifers constitute the natural accumulation of runoff and precipitation, and mis-estimated parameters (hydraulic properties such as the conductivity or storage capacity) and flows (e.g., recharge, discharge, deep inflow, preferential flow along faults and fractures) may lead to significant errors in predicted groundwater storage changes. An overestimation of runoff and/or evapotranspiration may also lead to an excessively quick return of the water to the atmosphere and ocean. Evapotranspiration may in particular be difficult to estimate in regions with temporary surface water bodies (for example related to the variation of the floodplain extension, or to the formation of temporary rivers flowing during the wet season and dried up during the dry season). If ISBA-CTRIP leads to TWS predictions in better agreement with GRACE than WGHM over remote areas, the situation is inverted for strongly anthropized regions such as the Northern Indian Plain, Central Valley (California, USA) or Great Plains (Ogallala, USA)





aquifer regions. Unlike WGHM, ISBA-CTRIP does not account for human induced changes in the TWS, and is therefore not able to reproduce TWS changes in highly anthropized regions. However, important differences between GRACE and WGHM are still observed in some highly anthropized regions, such as the Ogallala aquifer, which may be due to locally mis-estimated parameters. Large uncertainties may indeed affect the parameterisation of the water use model. For example, an overestimation of the irrigation efficiency may lead to an overestimation of evapotranspiration and underestimation of deep percolation. Errors in such parameterisation may have a strong effect on the predicted TWS changes, that could eventually be more accurately estimated using GRACE estimates to constrain unknown parameters. The evaluation of global hydrological models would therefore benefit the consideration of a broader range of datasets, including traditional discharge and evapotranspiration data, but also including terrestrial water storage anomalies from GRACE and GRACE-FO satellites. GRACE-based observations have for example been proven useful to quantify the impact of irrigation on groundwater resources in Northern India and improve groundwater forecasts under different Representative Concentration Pathways (RCP) in the region (Xie et al., 2020). Significant advances would be expected from the generalisation of such approaches in a dedicated framework (e.g., Condon et al., 2021, Gleeson et al., 2021).

**Appendix A Comparison of TWS anomalies from GRACE and global hydrological models over large lakes**

Residual TWS anomalies (Fig. A1) are compared for ISBA-CTRIP and WGHM with and without including the lake correction from the hydroweb database based on satellite altimetry and satellite imagery measurements. The TWS residuals are reduced for both models when applying the lake correction, especially around the Caspian Sea (-30 cm), North American Great Lakes (-7 cm), African Great lakes (-15 cm) and Volta Lake (-5 cm). A marginal increase (+2 cm) in TWS residuals can be observed for high altitude lakes of the Tibetan plateau (e.g., Pu Moyongcuo, Yamzho Yumco, Namu Cuo, Qinghai). Slight increases in the TWS residuals (at most +1 cm) are observed in a few anthropized regions when applying the lake correction to ISBA-CTRIP, especially near the Zeya Reservoir (Russia) and the Roraima region (North Brazil). Overall, the prediction of TWS anomalies due to hydrology is improved when using the lake correction and the residual TWS anomalies are reduced.

**Appendix B Location of eight regions with significant residual TWS anomalies**

Residual TWS anomalies are calculated as the difference between the TWS anomalies estimated from GRACE and global hydrological models. The ensemble of residual TWS anomalies counts 18 solutions, pertaining to 9 GRACE solutions (3 mascon and 6 spherical harmonic solutions) and 2 global hydrological models (ISBA-CTRIP and WGHM). The range of average residual TWS anomalies shown in Fig. B1a depends on the systematic biases between the TWS estimates from GRACE and global hydrological models. These differences are significant if they exceed the dispersion among the 18 solutions, calculated as the difference between the 97.5 and 2.5 percentiles of the range of residual TWS anomalies (see Fig. B1b). The significance ratio of residual TWS anomalies (Fig. B1c) has been calculated to identify where the differences





between GRACE solutions and hydrological models are significant, regardless of the solution or model considered. The dispersion of residual TWS solutions (Fig. B1b) is much larger than the dispersion of GRACE-based TWS solutions (Fig 1b), showing that the differences between the two models may have a large impact on the residuals and their significance.


To explore a large variety of scenarios, we selected 8 regions with large residuals (>10 cm) and high significance ratio (>2), including the central Amazon corridor (region A), the upper Sao Francisco River (region B), the Zambezi and Okavango rivers (region C), the Congo River (region D), the North of Australia (region E), the Ogallala aquifer in central USA (region F), the North of the Black Sea (region H) and the Northern Plains in India (region G). It may be noted that the significance ratio is not

extremely high across the North of India, because of the differences in the predictions of ISBA-CTRIP and WGHM. The region G was included to discuss the differences between models with respect to GRACE-based TWS anomalies. Glaciers and coastal regions have been excluded from the analyses (see section 3.1).

**Appendix C Comparison of TWS anomalies from GRACE and global hydrological models over large river basins**


Non-seasonal precipitation, TWS and residual TWS anomalies have been calculated and plotted for the 40 largest river basins of the world (Fig C1) according to the Global Runoff Data Centre (GRDC) Major River Basins (MRB) database (GRDC, 2020). The main conclusions drawn from global (section 3, main text) and regional (section 4, main text) analyses remain valid at basin scale. In particular, large residual TWS anomalies are observed at pluri-annual and decadal timescales, due to an

underestimation of slow TWS anomalies by the two global hydrological models considered in this study (ISBA-CTRIP and WGHM) when compared to GRACE. The amplitude of ISBA-CTRIP TWS predictions is closer to GRACE in remote river basins such as the Amazon, Lake Eyre, Murray Darling, Nelson, Okavango, Orinoco, Orange and Zambezi basins. WGHM better predicts TWS anomalies observed by GRACE in anthropized basins such as the Aral Sea, Colorado, Columbia, Ganges, Indus, Rio Grande or Yellow River basins. The difference of behaviour between both hydrological models is however not

systematic. For example, the TWS predictions from ISBA-CTRIP are closer to GRACE than WGHM across the Mississippi, Parana, Saint Lawrence or Yangtze basins, which are significantly affected by human interventions. Adversely, WGHM predictions fit better GRACE-based TWS anomalies than ISBA-CTRIP across the remote Yenisei and Kolyma river basins. However, it must be noted that large discrepancies are observed for both models when compared to GRACE for the Yenisei and Kolyma basins. Indeed, for a majority of basins (Dnieper, Danube, Amur, Brahmaputra, Congo, Chad, Jubba, Lena,

Mackenzie, Mekong, Niger, Nile, Ob, Sao Francisco, Shatt Al Arab, Tarim He, Tocantins, Volga, Yukon), both models struggle to reproduce non-seasonal TWS anomalies at pluri-annual and decadal time-scales.



## Appendix D Comparison of TWS anomalies from GRACE and global hydrological models over Southern India

TWS anomalies estimated from GRACE and global hydrological models have been averaged over Southern India and compared to in-situ and satellite precipitation (Fig. D1). The TWS anomalies captured with GRACE are well correlated with ISBA-CTRP (R=0.77) and mildly correlated (R=0.47) with WGHM predictions and precipitation (R=0.41 with a lag of 1 month). A spurious negative trend is observed in WGHM prediction over 2006-2016 (Fig. D1c), likely due to overestimated groundwater abstractions. Better performances are reached with ISBA-CTRIP, although anthropogenic contributions are neglected (Decharme et al., 2019).

## Code and data availability

All code and data necessary to validate the research findings have been place in a public repository at: https://doi.org/10.5281/zenodo.7142392

## Author contribution

All authors contributed to the conceptualization of ideas presented in the manuscript. JP, AB, BD and SM provided
resources necessary to conduct the research findings. JP carried out the formal analysis. AC provided research supervision and funding acquisition. All authors contributed to the investigation of research findings. JP wrote the original draft. All authors contributed to the review and editing of the manuscript.

## Competing interests

The authors declare that they have no conflict of interest.
## Acknowledgements

This project has received funding from the European Research Council (ERC) under the European Union's Horizon 2020 research and innovation program (GRACEFUL Synergy Grant agreement No 855677).

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

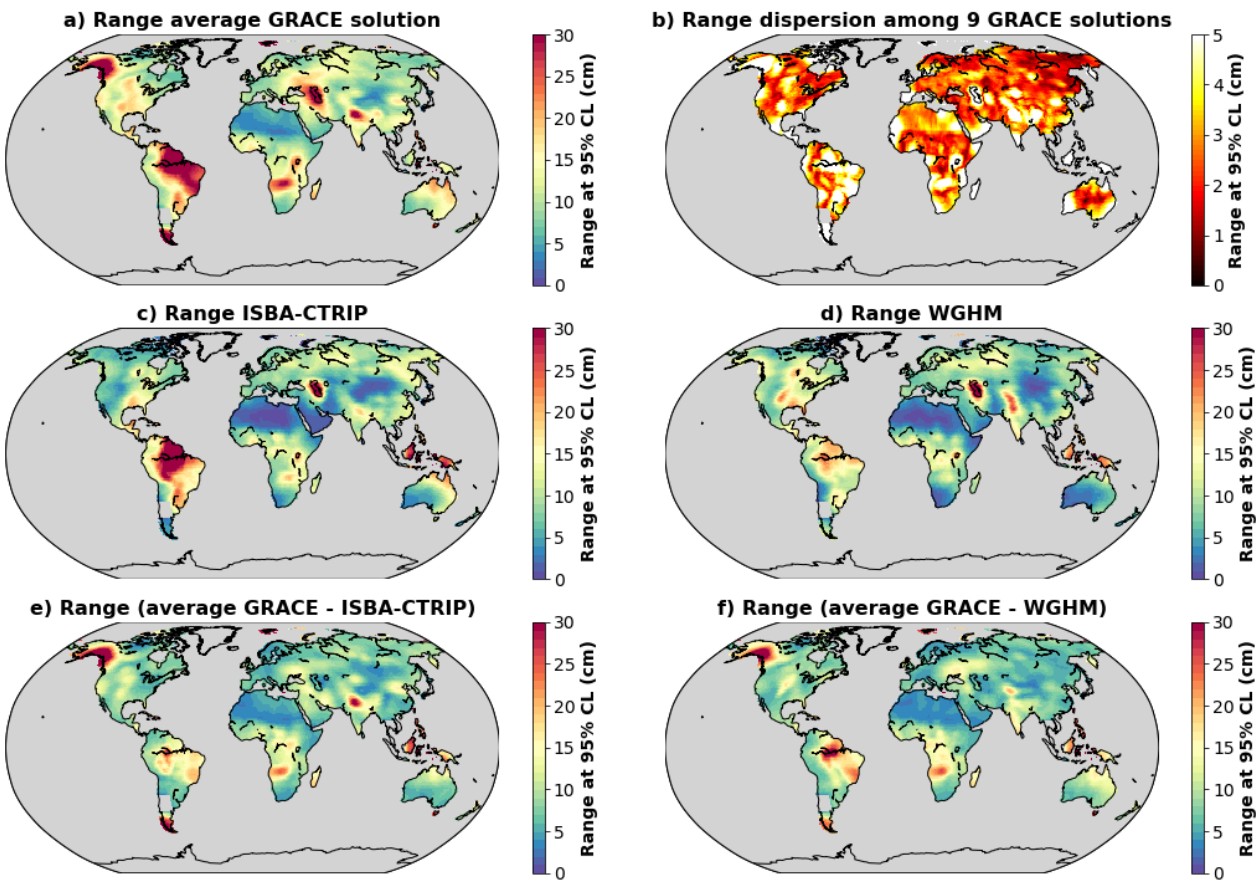

**Figure 1: Comparison of TWS anomalies estimated from an ensemble of nine GRACE solutions and two global hydrological models. a) Range of TWS anomalies estimated as the average of nine GRACE solutions. b) Dispersion of the range of TWS anomalies among nine GRACE solutions. Range of TWS anomalies estimated with ISBA-CTRIP (c)**



**and WGHM (d). Range of residual TWS anomalies estimated as the difference between the average of 9 GRACE solutions and ISBA-CTRIP (e) or WGHM (f).**

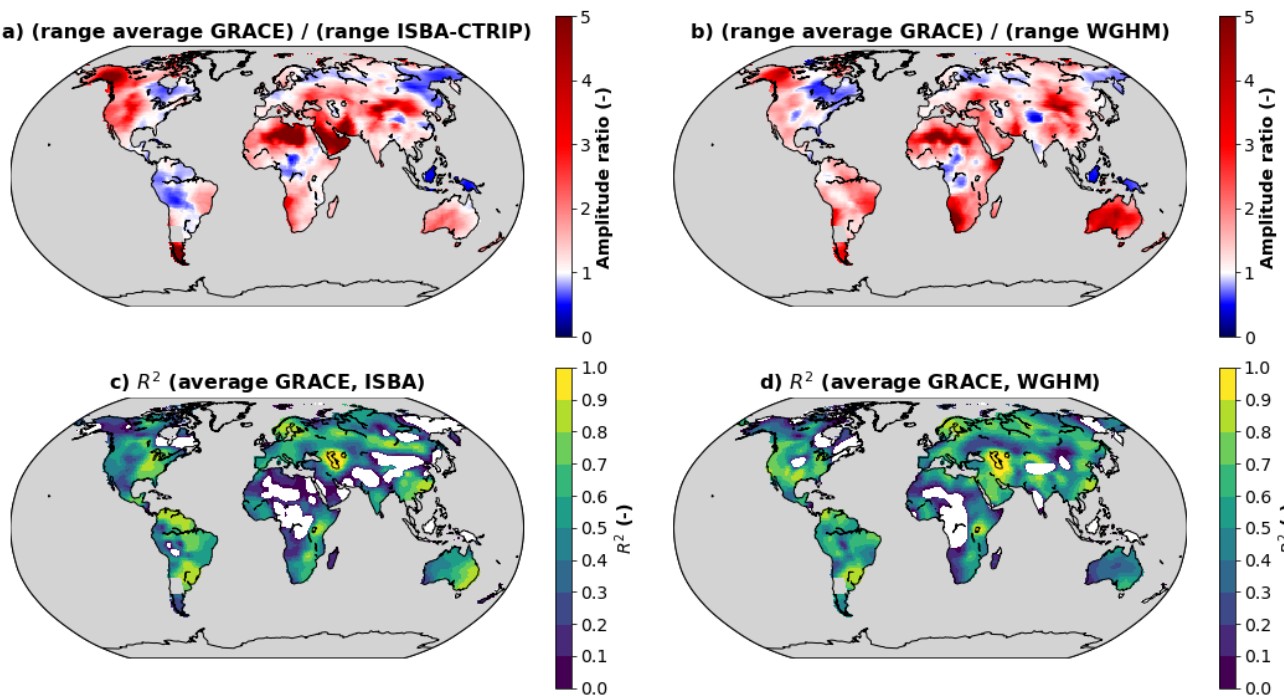

**Figure 2: Range ratios between the average GRACE solution and the hydrological models ISBA-CTRIP (a) and WGHM (b). Determination coefficients between the average GRACE solution and the hydrological models ISBA-CTRIP (c) and WGHM (d). Regions, where the coefficient of determination is negative, are shown in white**



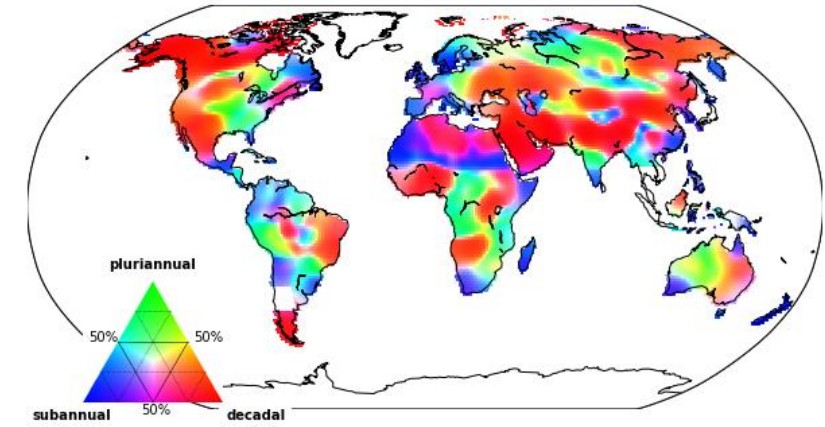

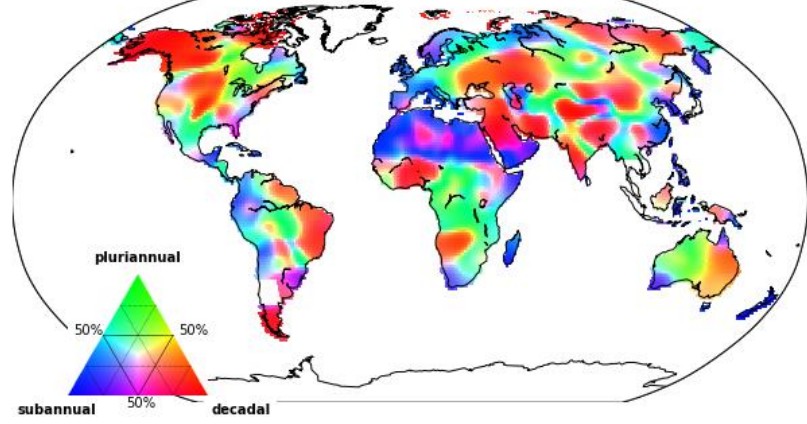

**Figure 3: Characteristic time scales in residual TWS anomalies calculated as the differences between the average GRACE solution and ISBA-CTRIP (a) or WGHM (b). Subannual, pluriannual and decadal contributions have been computed with high-pass (cut-off period at 1.5 years), band-pass (cut-off periods at 1.5 and 10 years) and low-pass (cut-off period at 10 years) filters respectively. The percentage of variance explained by one contribution has been calculated as the coefficient of determination with respect to the full residual signal.**







**Figure 4: a) Linear trends in residual TWS anomalies calculated as the difference between the average GRACE solution and ISBA-CTRIP. b) Same as (a) with WGHM. c) Amplitude of non-linear signals in residual TWS anomalies calculated as the difference between the average GRACE solution and ISBA-CTRIP. The amplitude is calculated as the difference between the 97.5 and 2.5 percentiles. d) Same as (c) with WGHM. e) Coefficient of determination calculated for non-linear signals with respect to TWS anomalies calculated as the difference between the average GRACE solution and ISBA-CTRIP. f) Same as (e) with WGHM.**



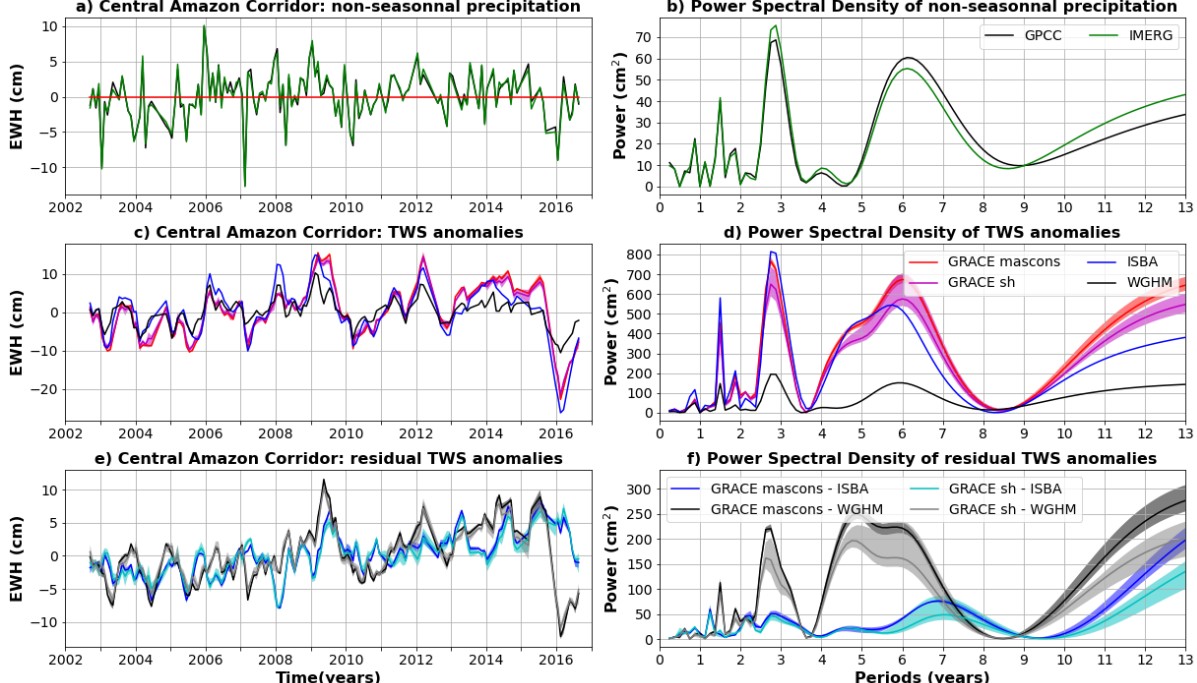

**Figure 5: Comparison of TWS and precipitation anomalies averaged over the Central Amazon Corridor (box A in Fig. B1 - Appendix B). a) Average precipitation anomalies for the GPCC (gauge-based) and IMERG (satellite-based) products. b) Power Spectral Density (PSD) of average precipitation anomalies. c) TWS anomalies average over the central Amazon for two global hydrological models (ISBA-CTRIP in blue and WGHM in black) and 9 GRACE solutions (mascons in red, spherical harmonic in magenta). The solid line corresponds to the average of the sub-ensemble, the shaded area to the minimum to maximum envelope. d) PSD of the averaged TWS anomalies shown in (c). e) Residual TWS anomalies averaged over the central Amazon corridor and calculated as the difference between GRACE and ISBA-CTRIP (blue when the difference is calculated with mascons, cyan with spherical harmonics) or WGHM (black when the difference is calculated with mascons, grey with spherical harmonics).**





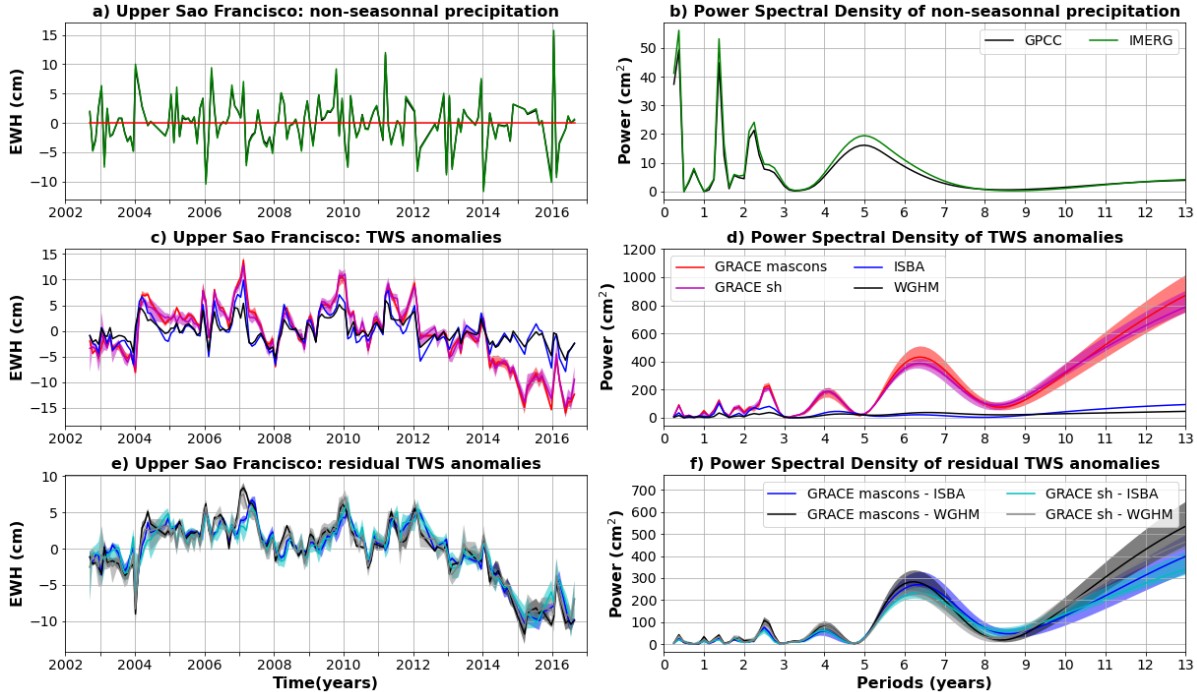

**Figure 6: Same as Fig. 5 but for the Upper Sao Francisco (box B in Fig. B1 - Appendix B).**




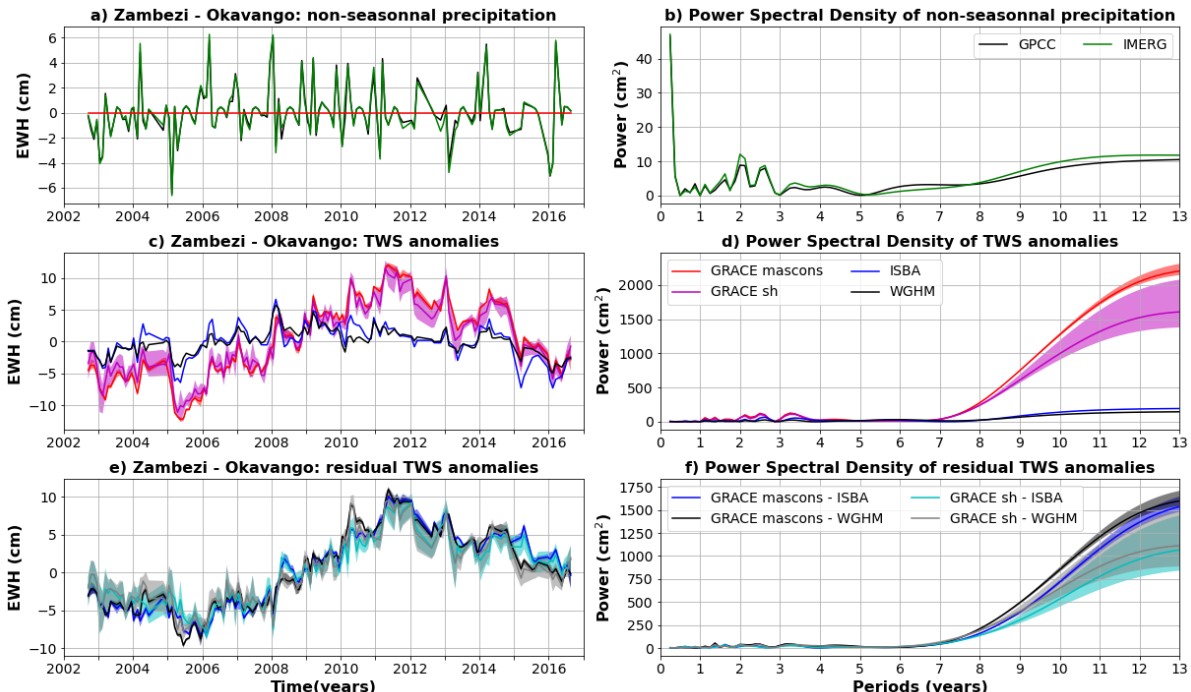

**Figure 7: Same as Fig. 5 but for the Zambezi and Okavango rivers (box C in Fig. B1 - Appendix B).**





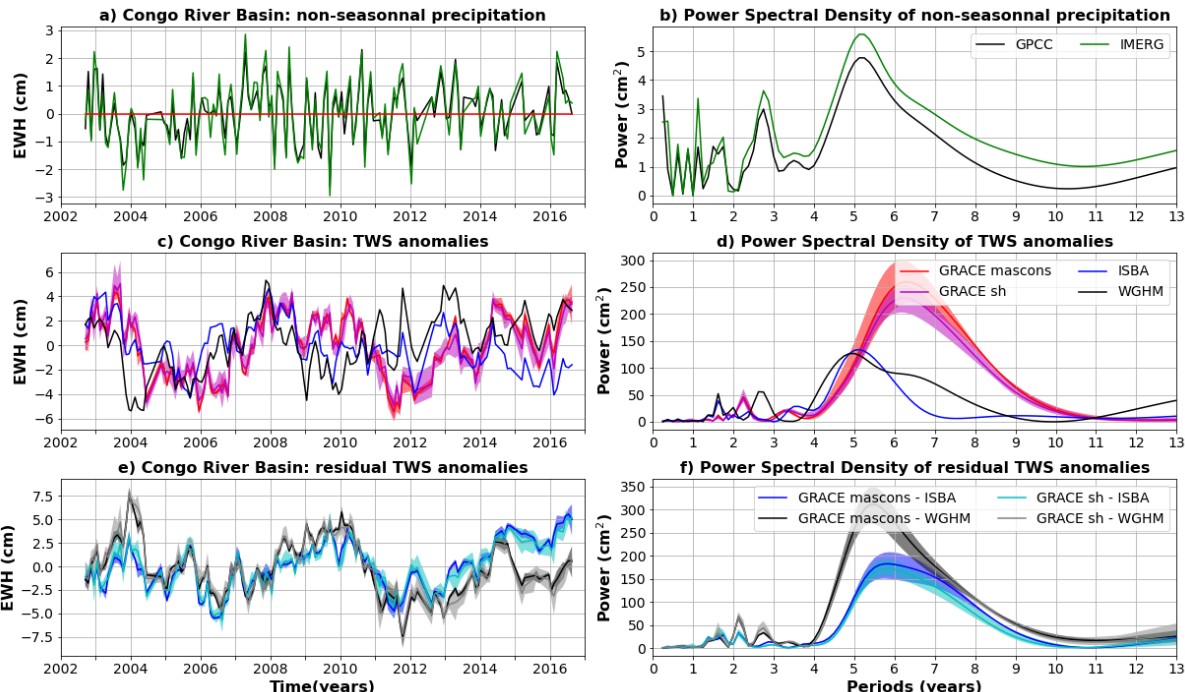

**Figure 8: Same as Fig. 5 but for the Congo River (box D in Fig. B1 - Appendix B).**



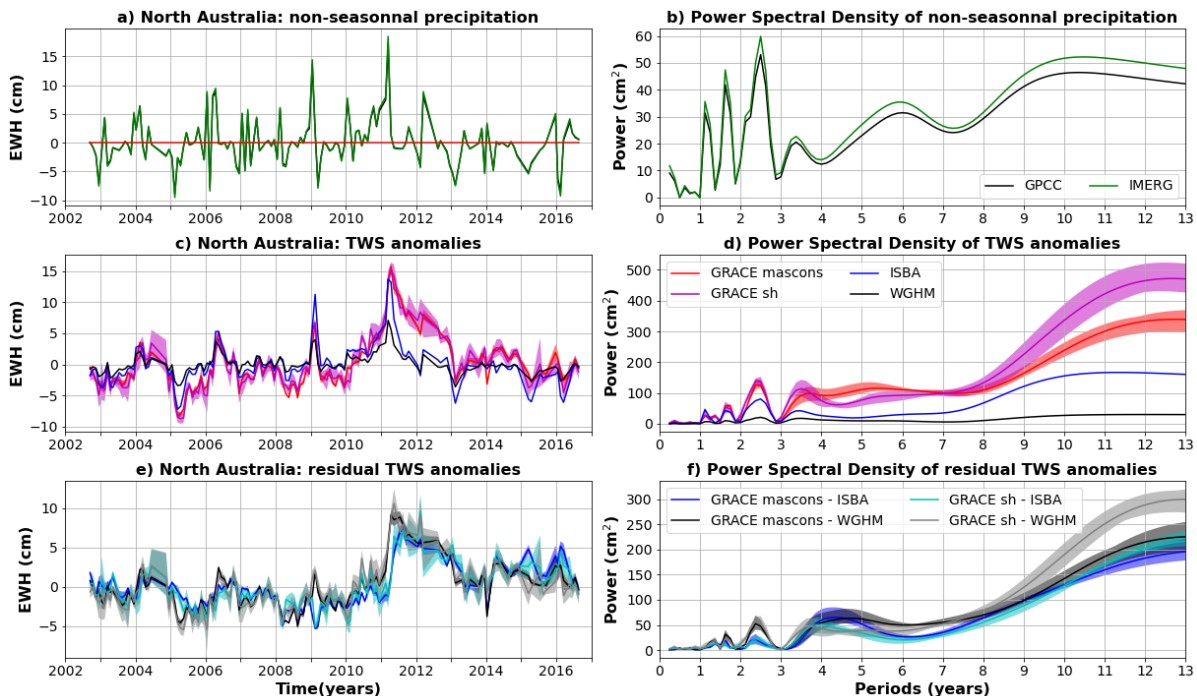

**Figure 9: Same as Fig. 5 but for North Australia (box E in Fig. B1 - Appendix B).**





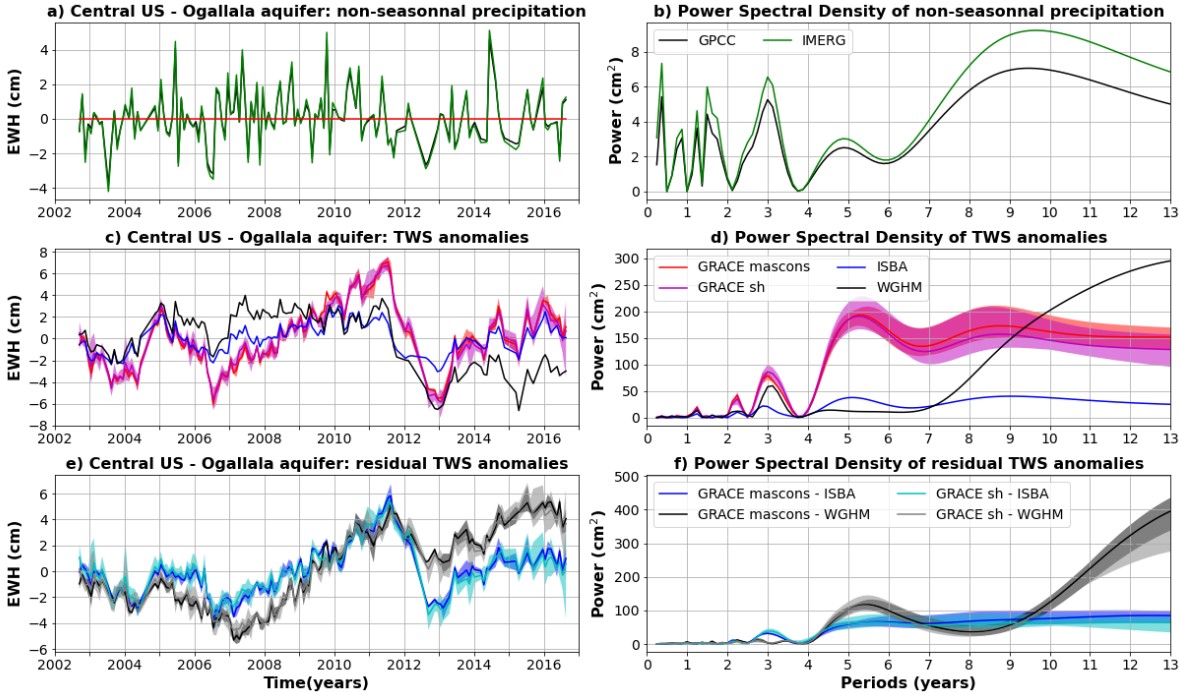

**Figure 10: Same as Fig. 5 but for the Central USA - Ogallala aquifer region (box F in Fig. B1 - Appendix B).**



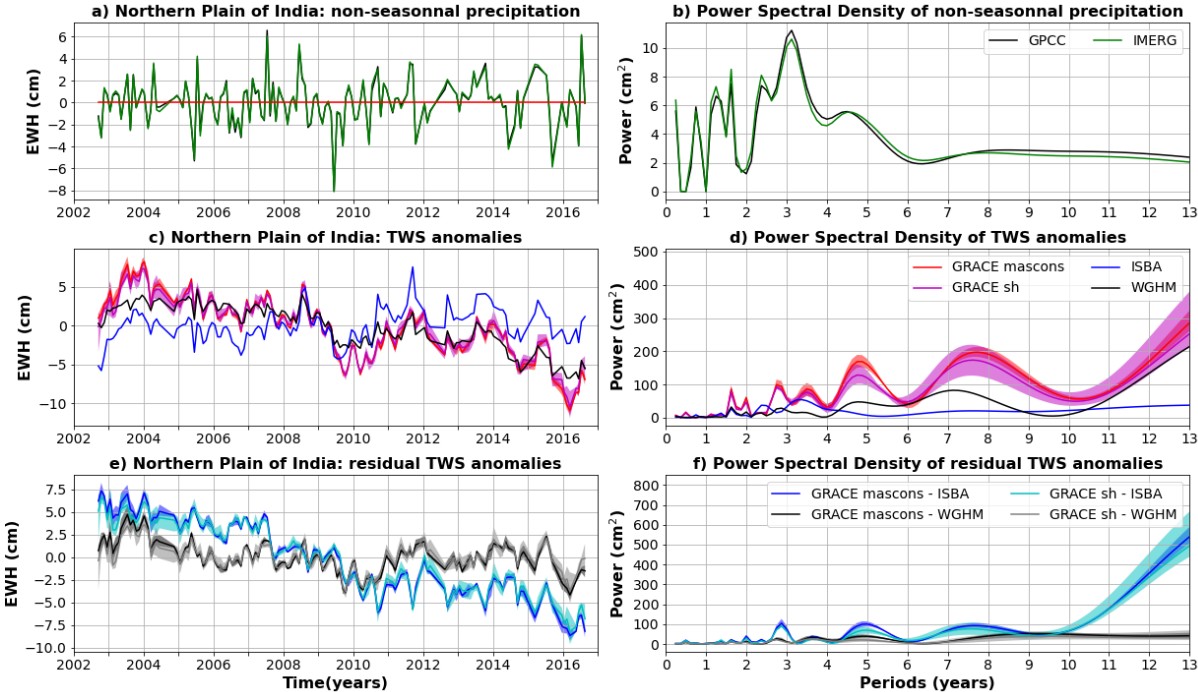

**Figure 11: Same as Fig. 5 but for the Indian Northern Plains (box G in Fig. B1 - Appendix B).**



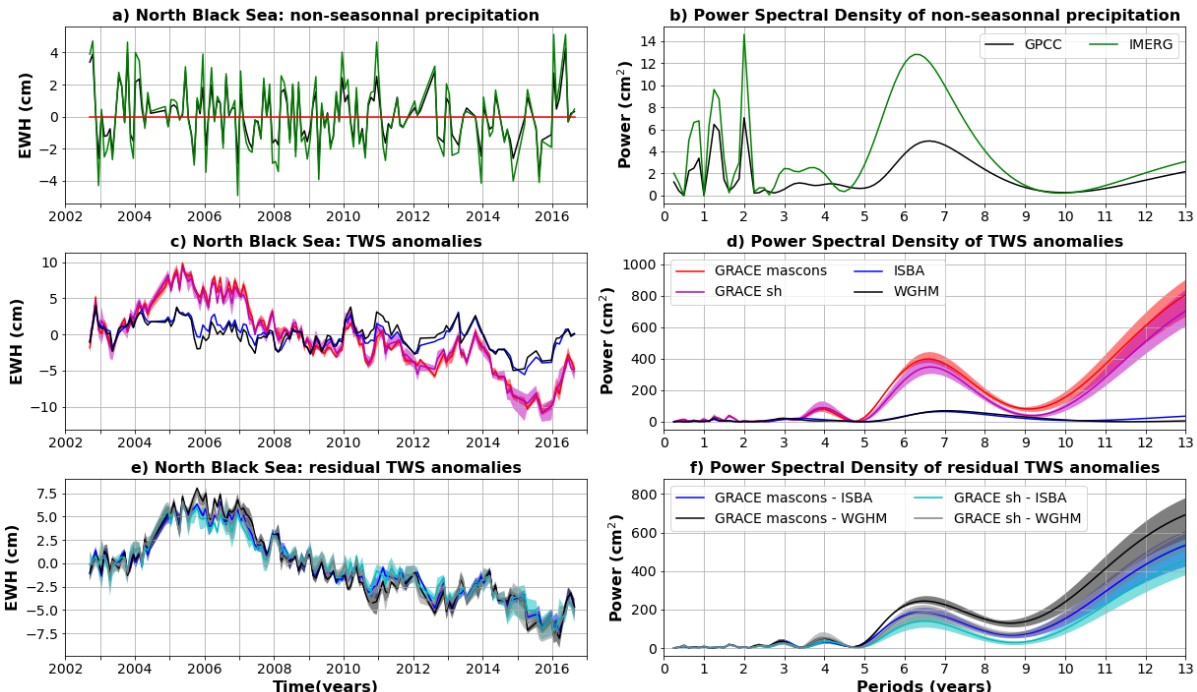

**Figure 12: Same as Fig. 5 but for the North of the Black Sea (box H in Fig. B1 - Appendix B).**





**Figure A1: a)** Range of residual TWS anomalies calculated with ISBA-CTRIP. **b)** Range of residual TWS anomalies calculated with WGHM. **c)** Range of residual TWS anomalies calculated with ISBA-CTRIP without including the lake correction. **d)** Range of residual TWS anomalies calculated with WGHM without including the lake correction. **d)** Difference between a and c due to the lake correction. **e)** Difference between b and d due to the lake correction.





**Figure B1: a) Average range of 18 residual TWS anomalies. b) Dispersion of the range of residual TWS anomalies. The dispersion is calculated as the difference between the 97.5 and 2.5 percentiles of the range of 18 residual TWS anomalies. c) Significance ratio of the averaged residual TWS anomalies calculated as the average range of residual TWS anomalies (a) divided by the dispersion of the range among the 18 solutions (b).**



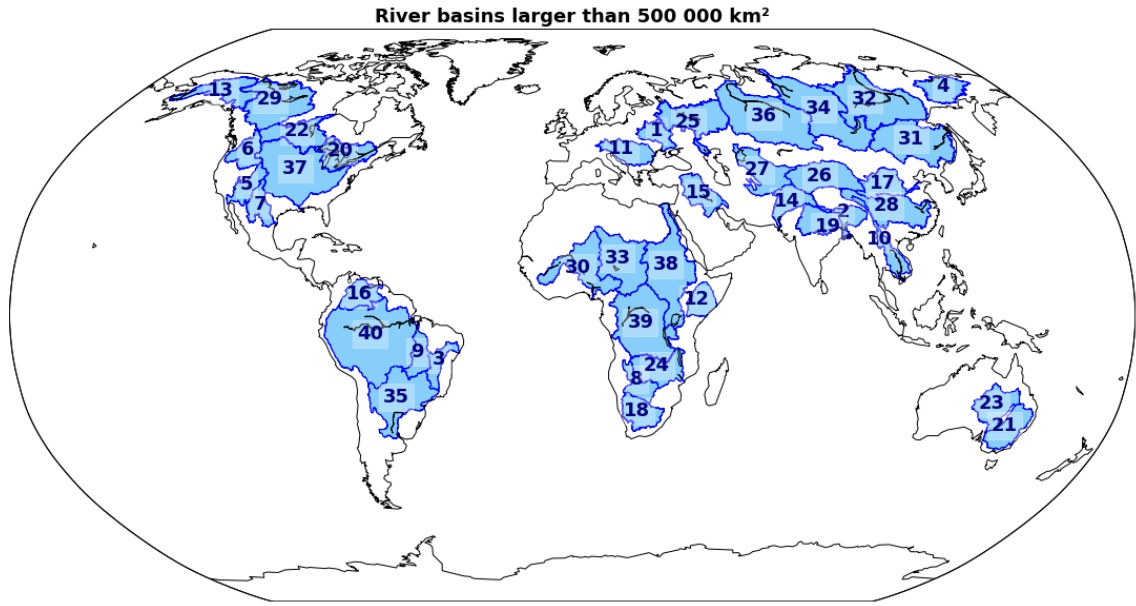

**Figure C1: Map of the 40 largest river basins considered in this study: 1) Dnieper, 2) Brahmaputra, 3) Sao Francisco, 4) Kolyma, 5) Colorado, 6) Columbia, 7) Rio Grande, 8) Okavango, 9) Tocantins, 10) Mekong, 11) Danube, 12) Jubba, 13) Yukon, 14) Indus, 15) Shatt Al Arab, 16) Orinoco, 17) Yellow River, 18) Orange, 19) Ganges, 20) Saint Lawrence, 21) Murray, 22) Nelson, 23) Lake Eyre, 24) Zambezi, 25) Volga, 26) Tarim He, 27) Aral Sea, 28) Yangtze, 29) Mackenzie, 30) Niger, 31) Amur, 32) Lena, 33) Chad, 34) Yenisei, 35) Parana, 36) Ob, 37) Mississippi, 38) Nile, 39) Congo, 40) Amazon.**



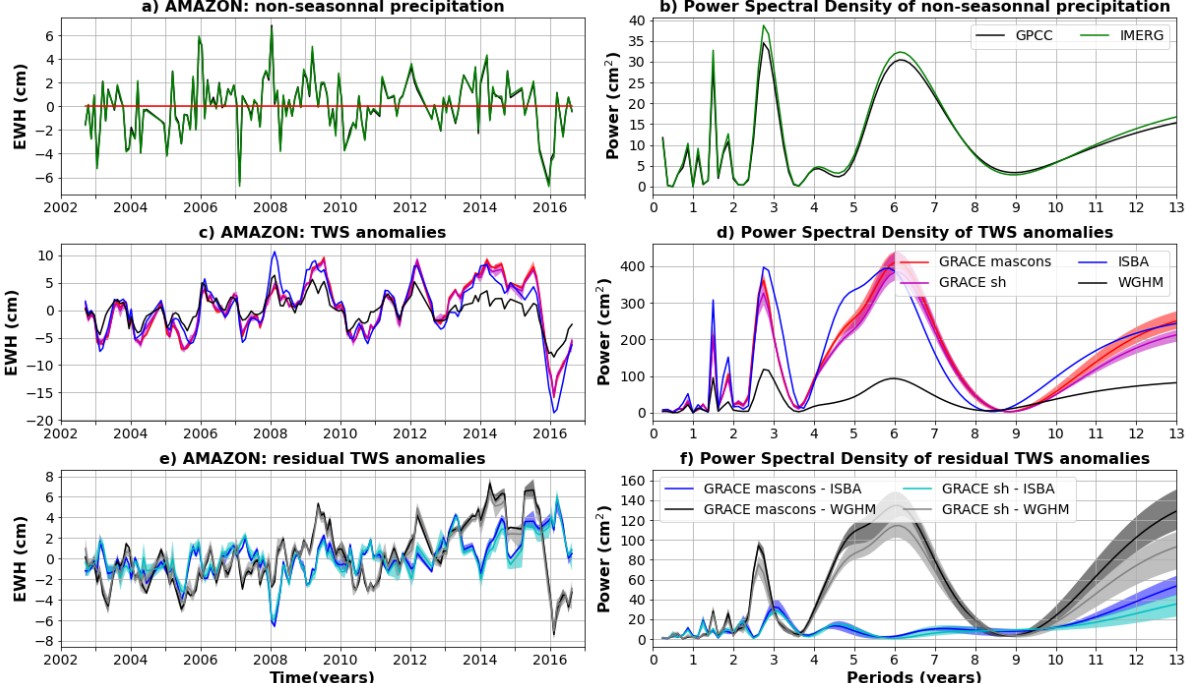

**Figure C2: Comparison of TWS and precipitation anomalies averaged over Amazon basin. a) Average precipitation anomalies for the GPCC (gauge-based) and IMERG (satellite-based) products. b) Power Spectral Density (PSD) of average precipitation anomalies. c) TWS anomalies average over the central Amazon for two global hydrological models (ISBA-CTRIP in blue and WGHM in black) and 9 GRACE solutions (mascons in red, spherical harmonic in magenta). The solid line corresponds to the average of the sub-ensemble, the shaded area to the minimum to maximum envelope. d) PSD of the averaged TWS anomalies shown in (c). e) Residual TWS anomalies averaged over the central Amazon corridor and calculated as the difference between GRACE and ISBA-CTRIP (blue when the difference is calculated with mascons, cyan with spherical harmonics) or WGHM (black when the difference is calculated with mascons, grey with spherical harmonics).**





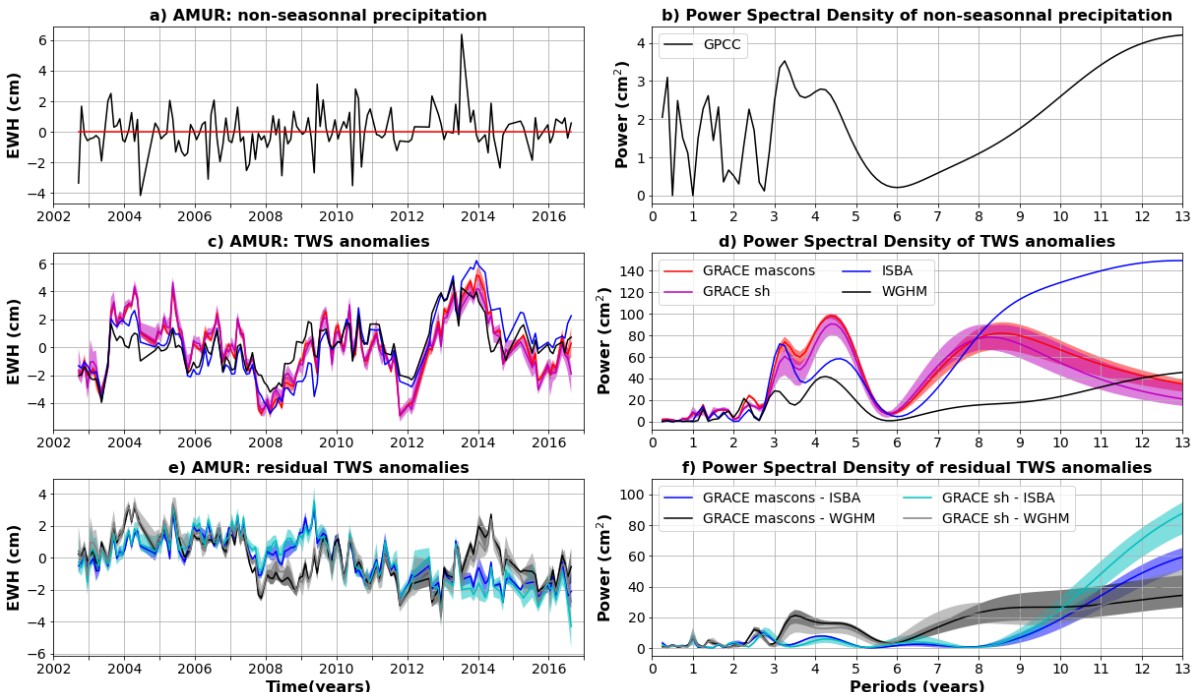

**Figure C3: Same as C2 for the Amur Basin. Non-seasonal precipitation anomalies are only estimated with GPCC, as a significant part of the basin is not covered by IMERG satellites due to the high latitude of the Amur basin.**






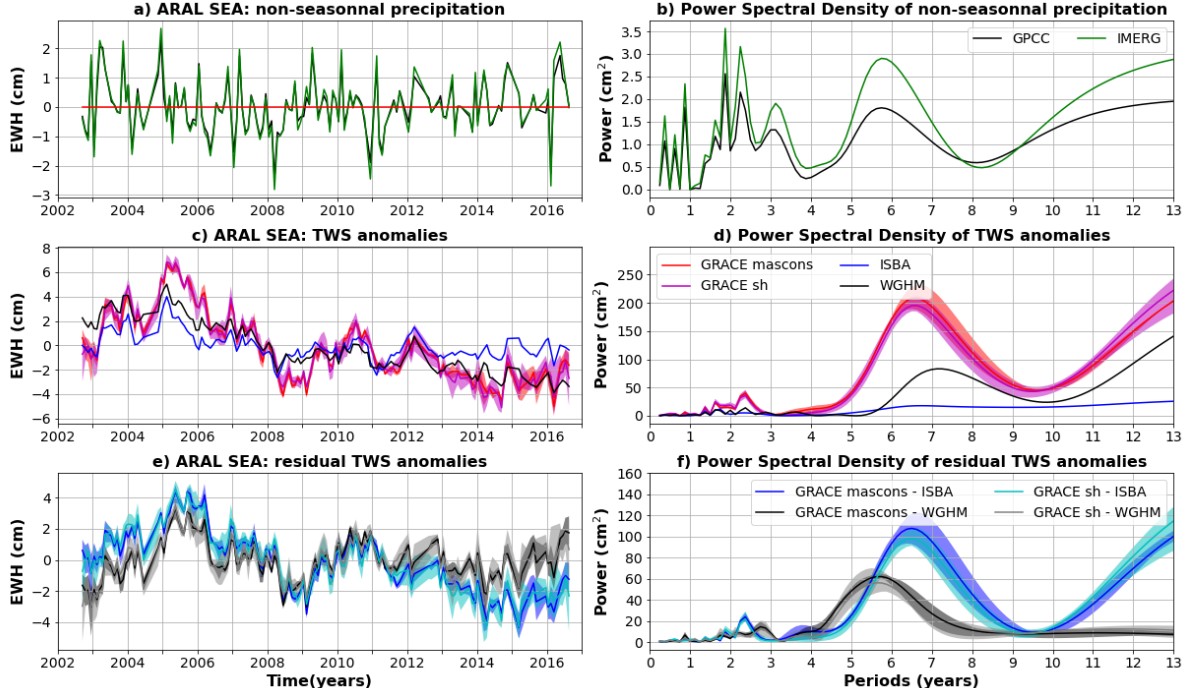

**Figure C4: Same as C2 for the Aral Sea basin.**



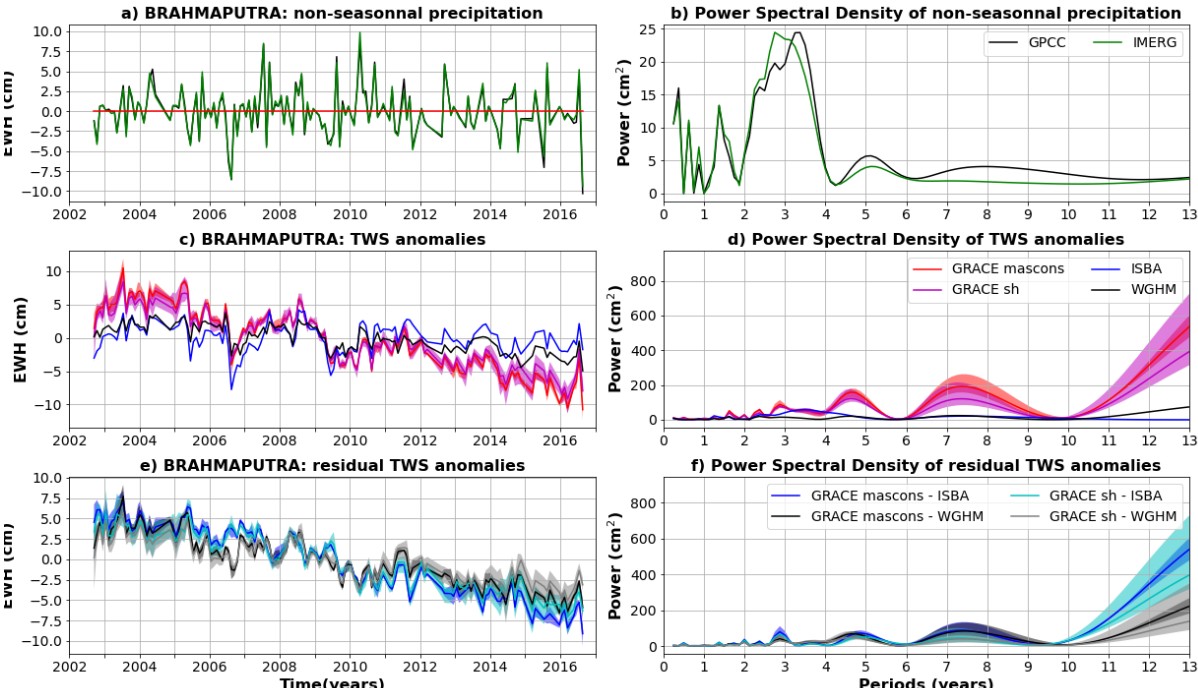

**Figure C5: Same as C2 for the Brahmaputra basin.**



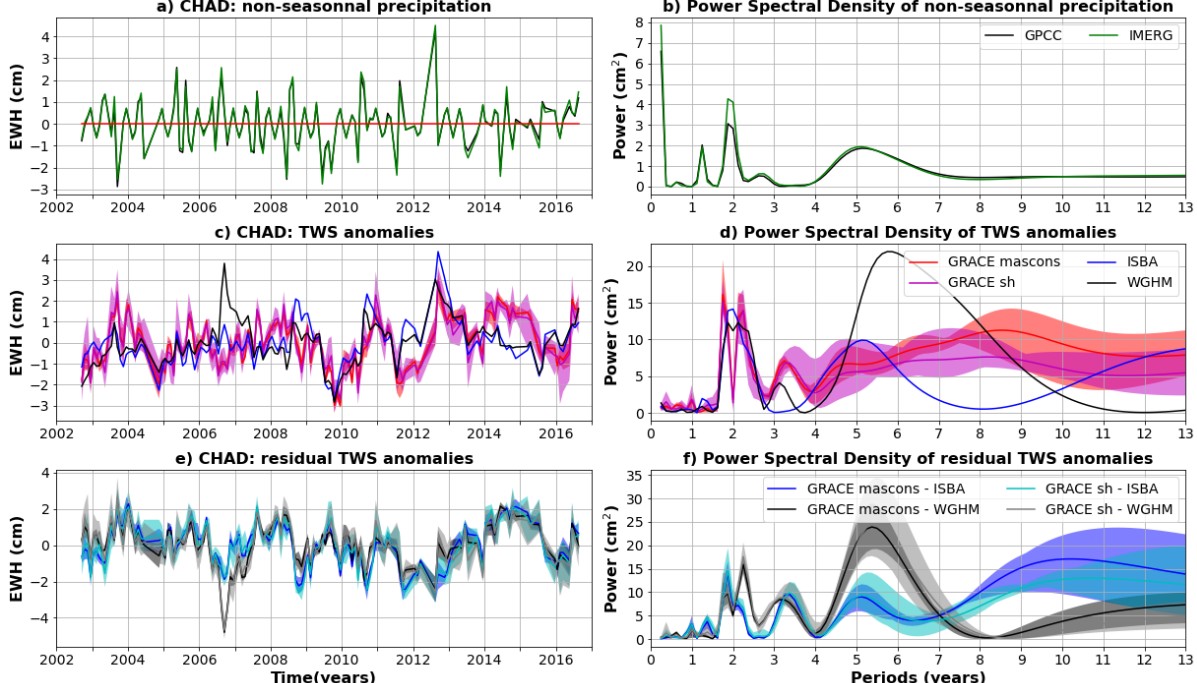

**Figure C6: Same as C2 for the Chad basin.**




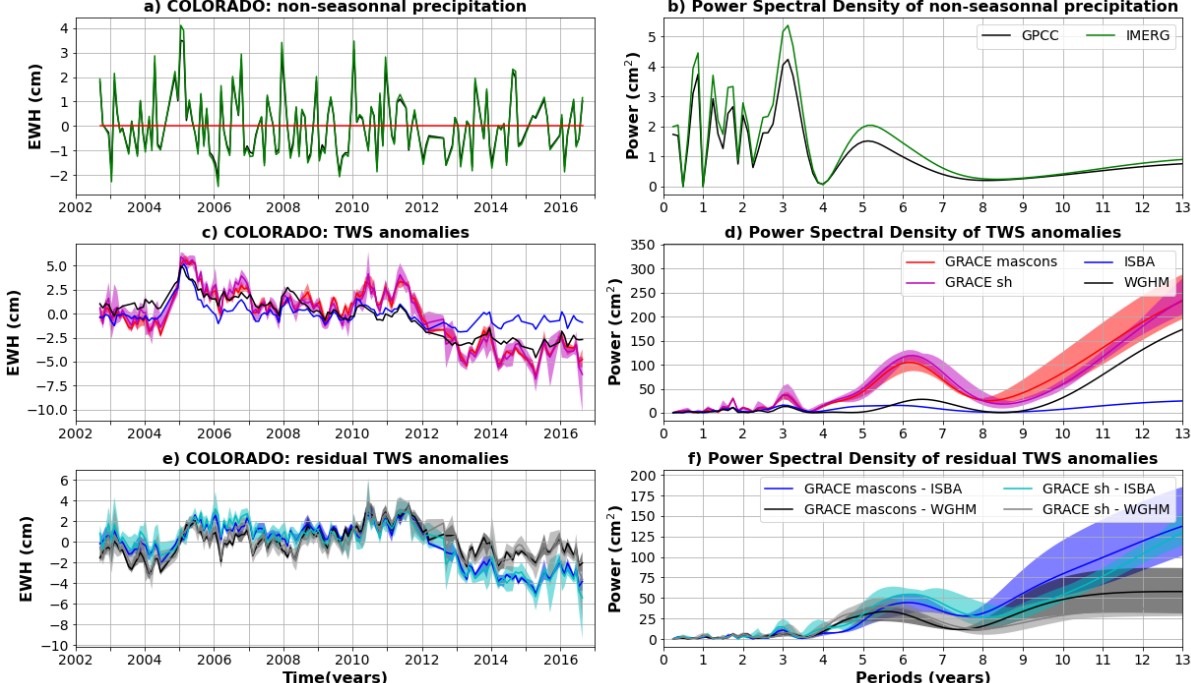

**Figure C7: Same as C2 for the Colorado basin.**



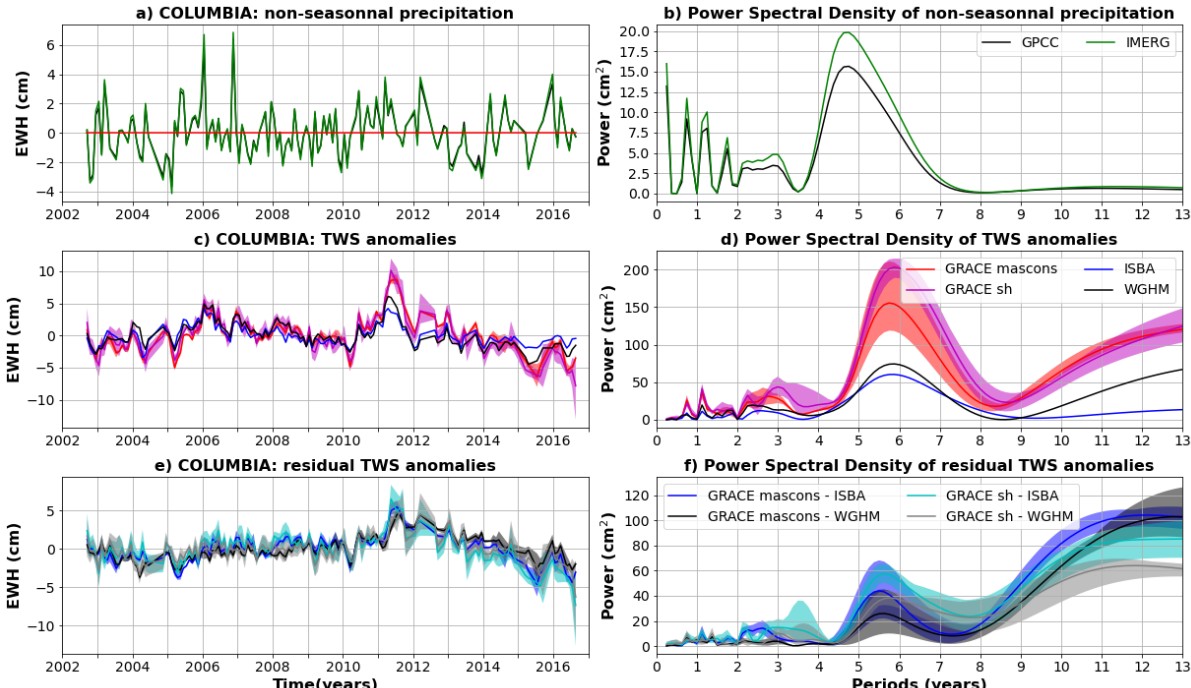

**Figure C8: Same as C2 for the Columbia basin.**



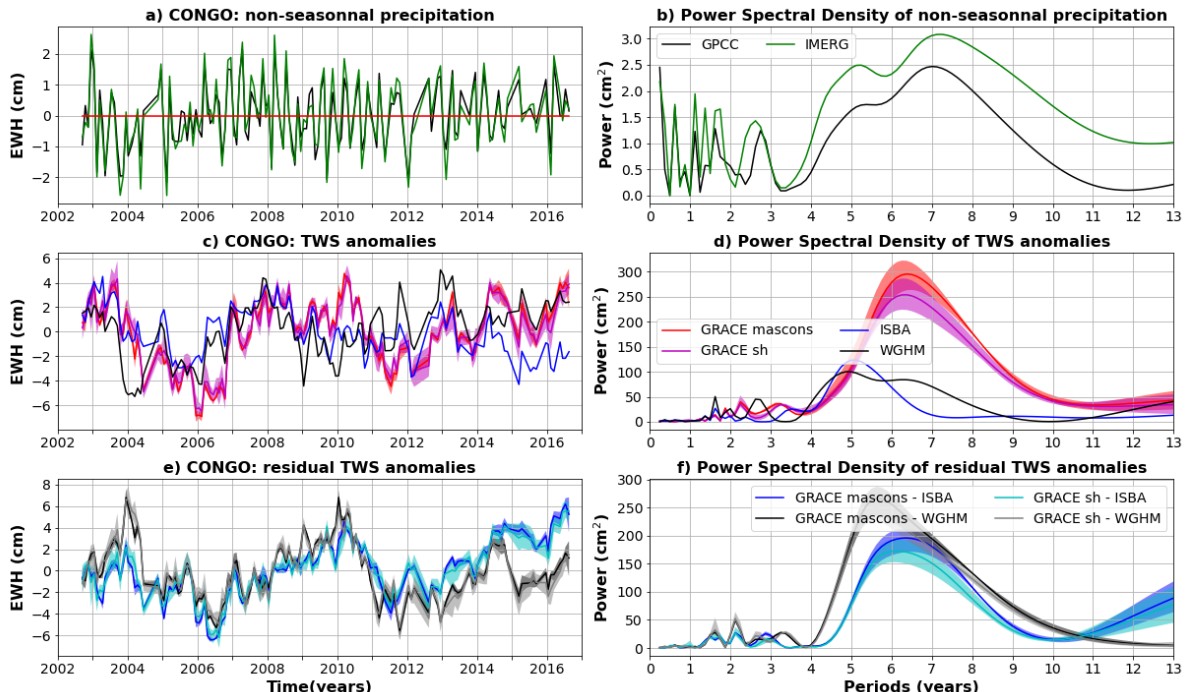

**Figure C9: Same as C2 for the Congo basin.**





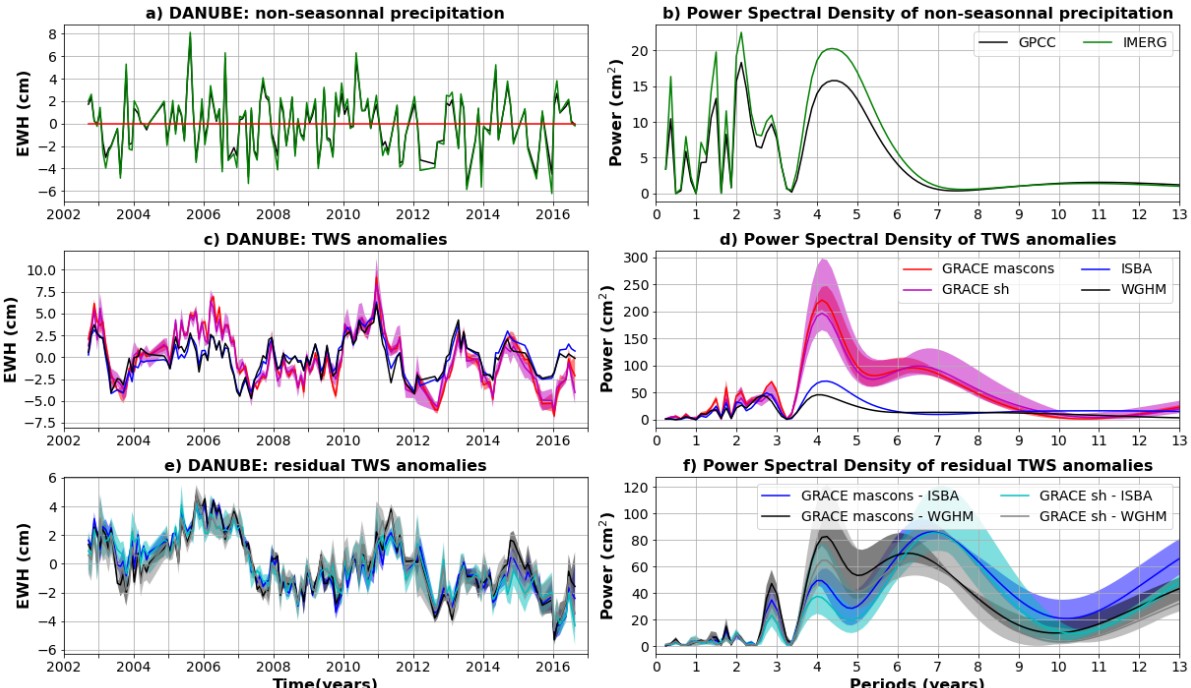


**Figure C10: Same as C2 for the Danube basin.**



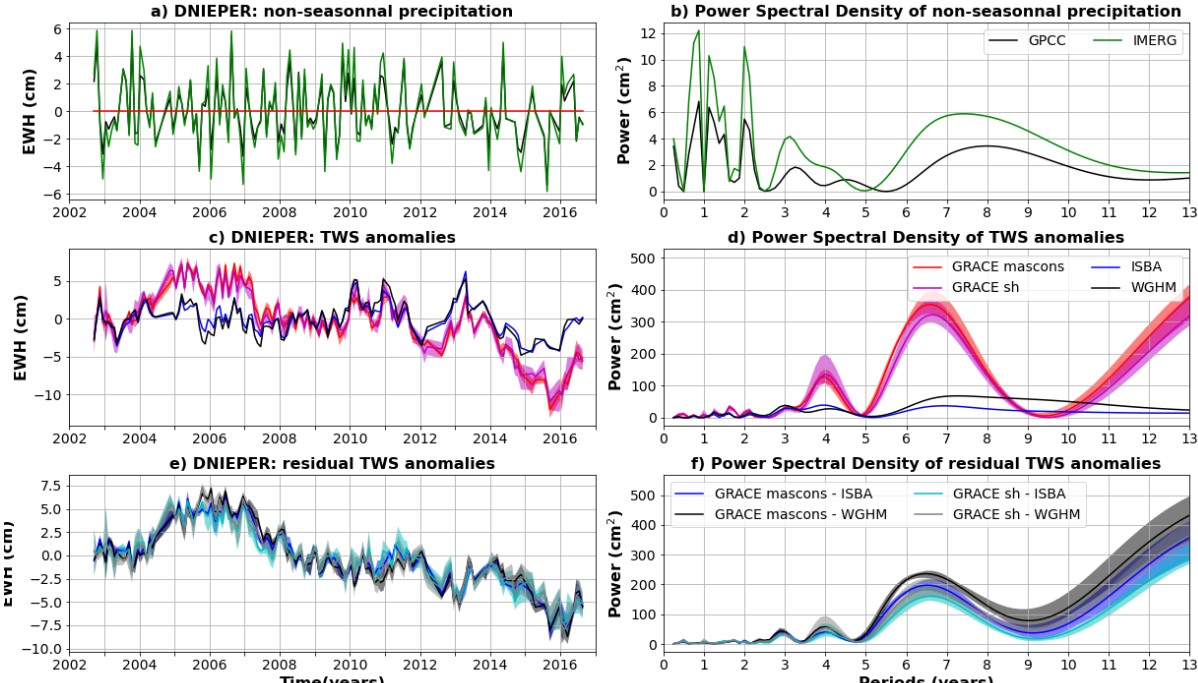

**Figure C11: Same as C2 for the Dnieper basin.**




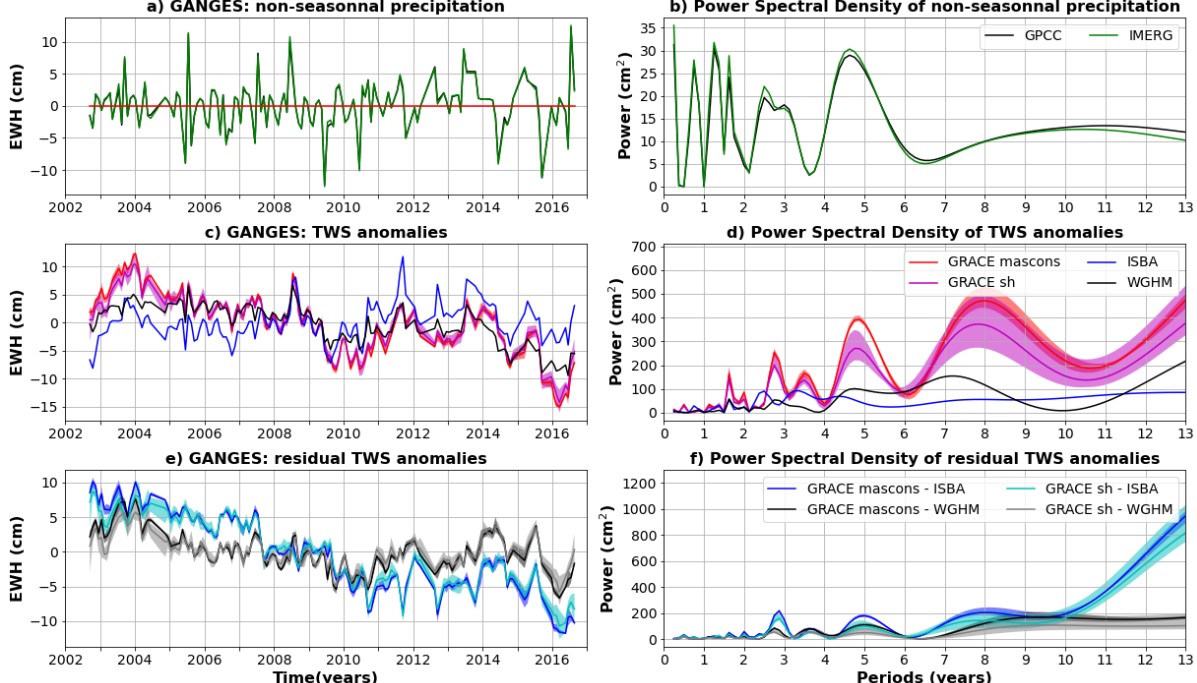

**Figure C12: Same as C2 for the Ganges basin.**





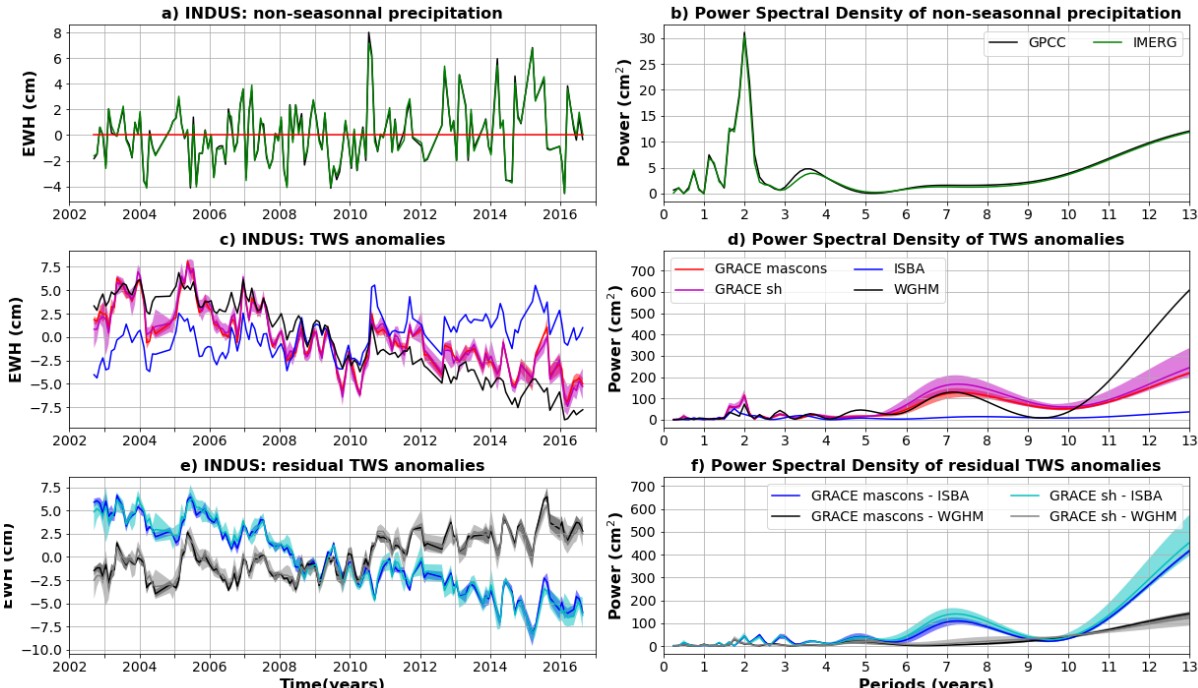


**Figure C13: Same as C2 for the Indus basin.**





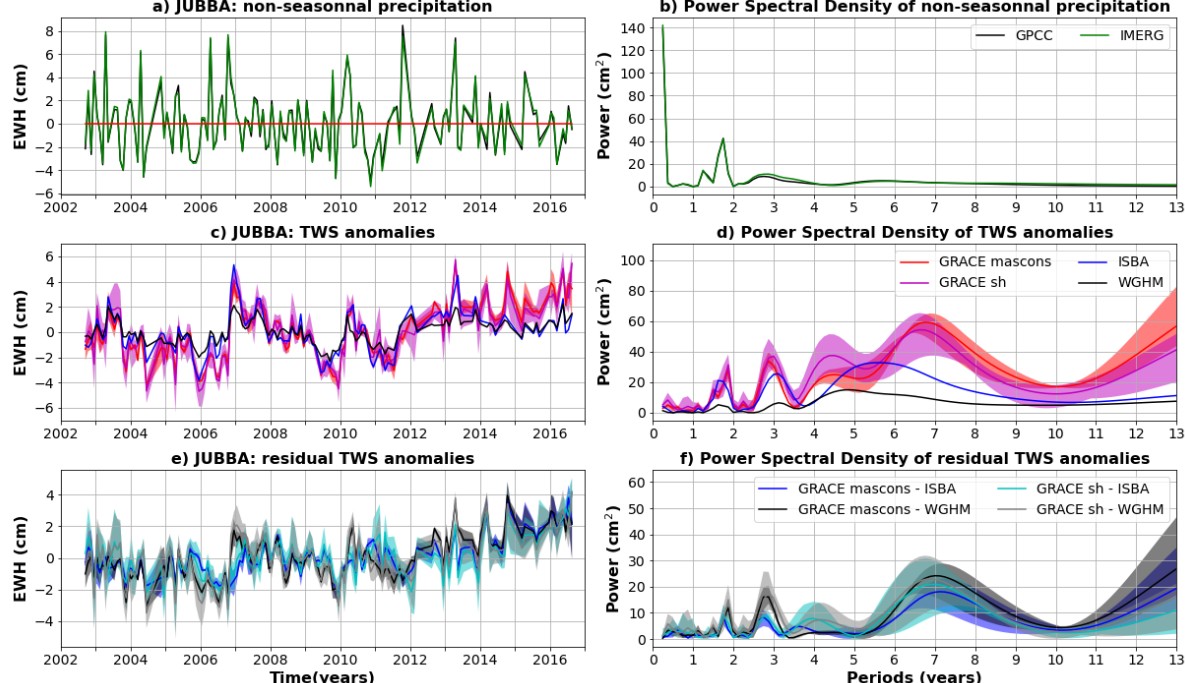

**Figure C14: Same as C2 for the Jubba basin.**



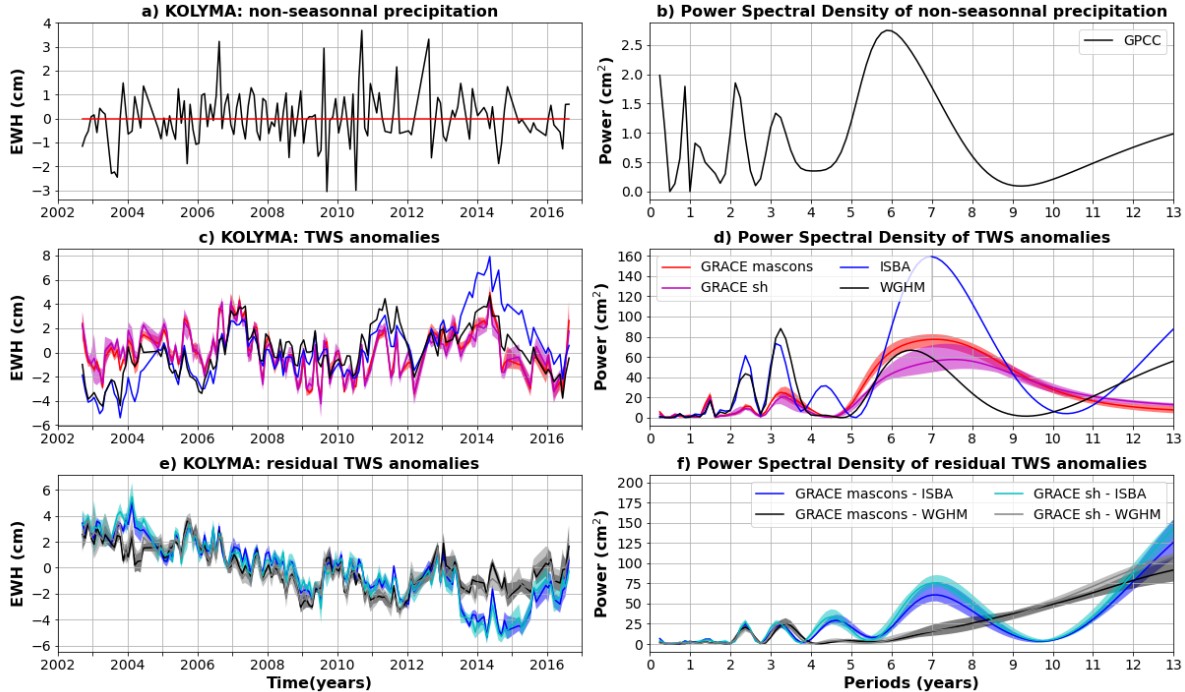

**Figure C15: Same as C2 for the Kolyma basin. Non-seasonal precipitation anomalies are only estimated with GPCC, as a significant part of the river basin is not covered by IMERG satellites due to its high latitude.**




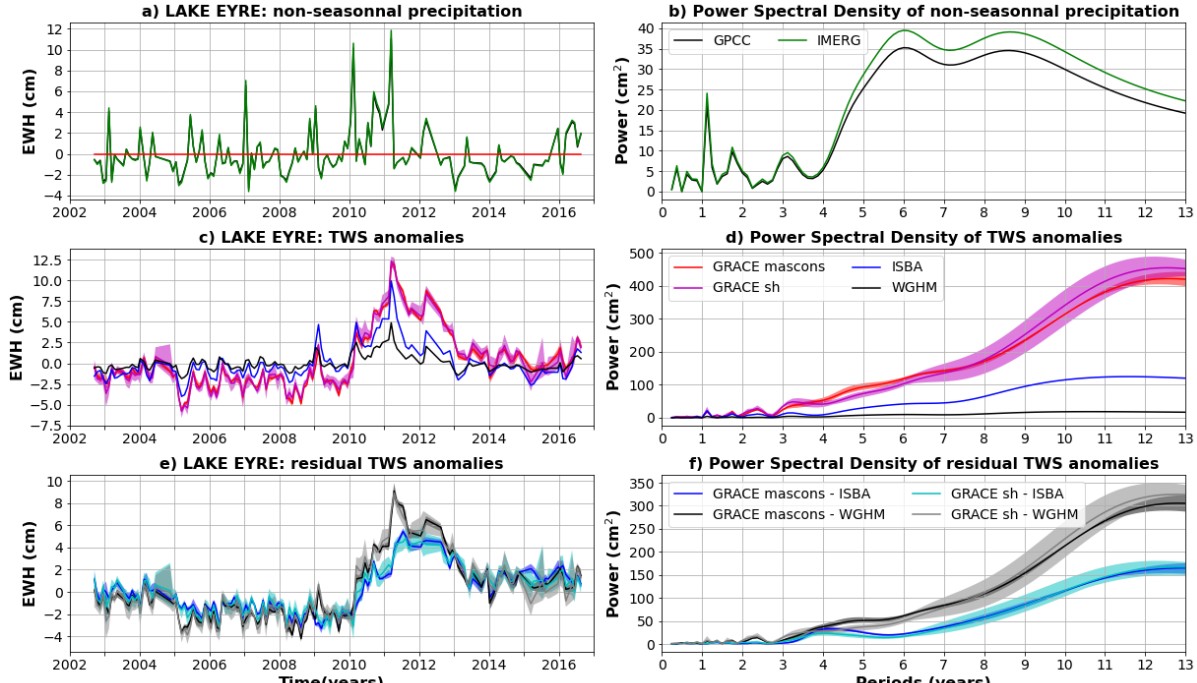

**Figure C16: Same as C2 for the Lake Eyre basin.**



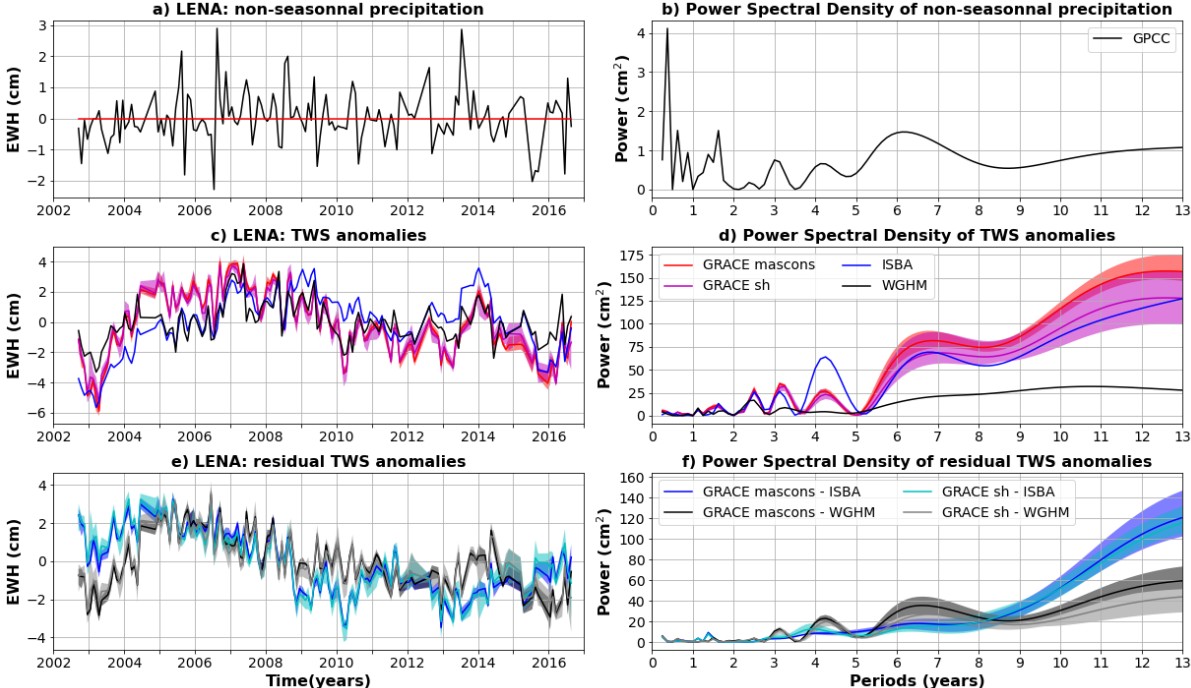

**Figure C17: Same as C2 for the Lena basin. Non-seasonal precipitation anomalies are only estimated with GPCC, as a significant part of the river basin is not covered by IMERG satellites due to its high latitude.**



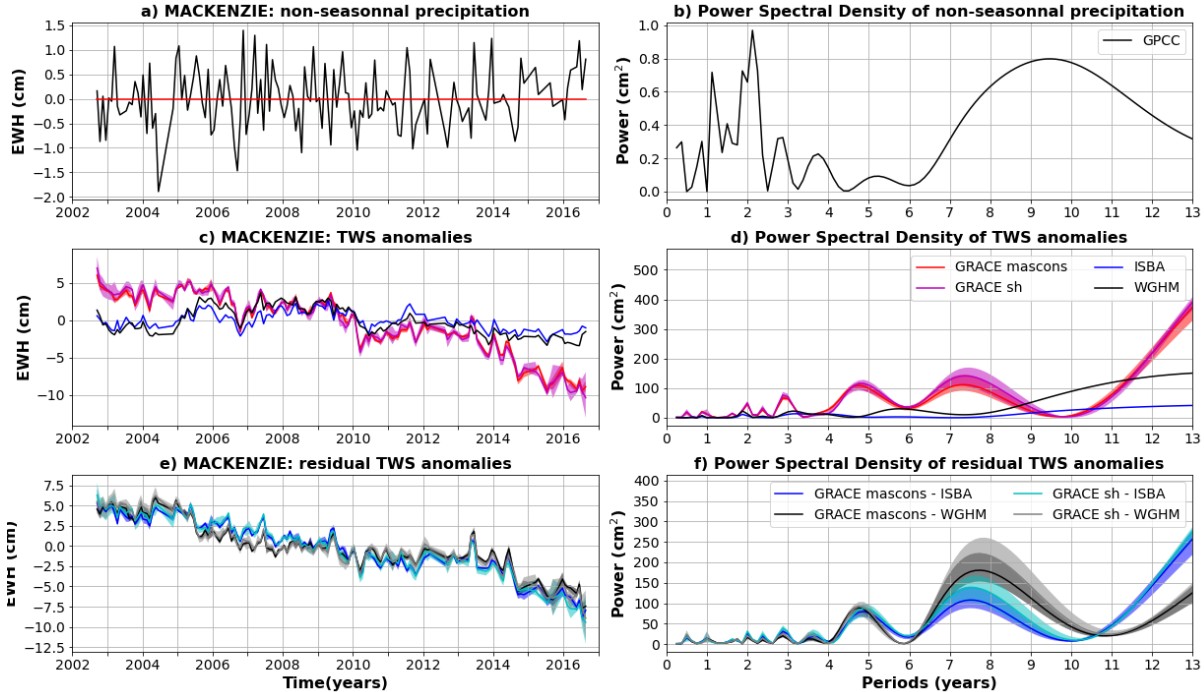


**Figure C18: Same as C2 for the Mackenzie basin. Non-seasonal precipitation anomalies are only estimated with GPCC, as a significant part of the river basin is not covered by IMERG satellites due to its high latitude.**



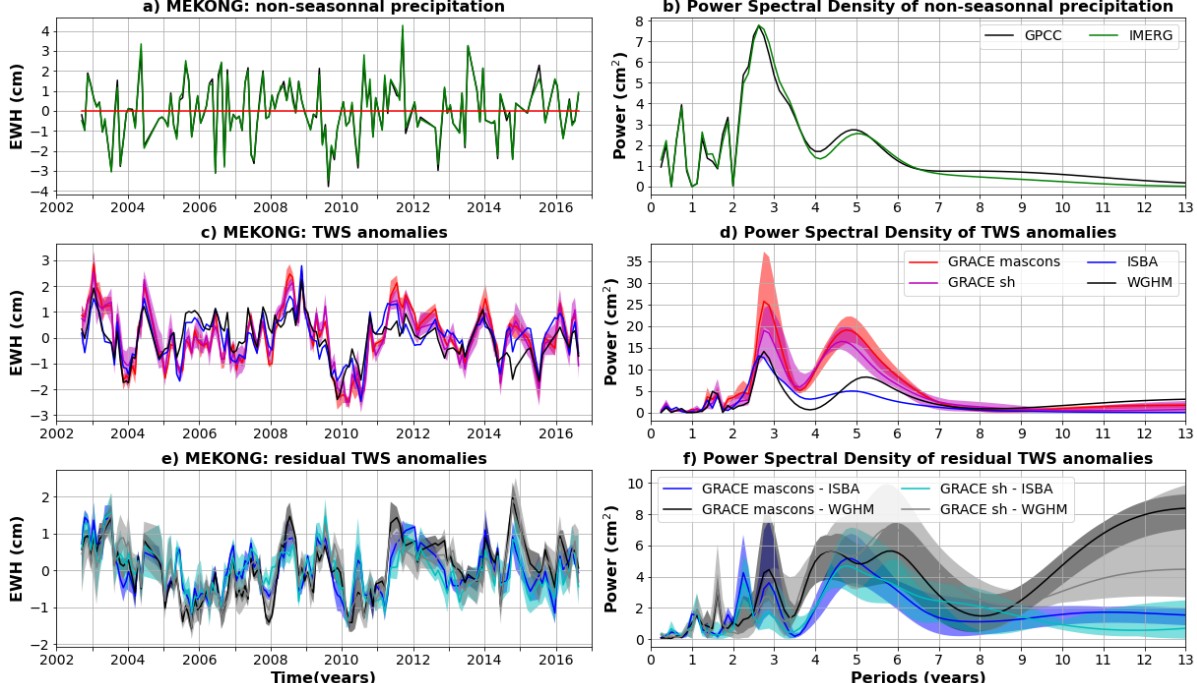

**Figure C19: Same as C2 for the Mekong basin.**





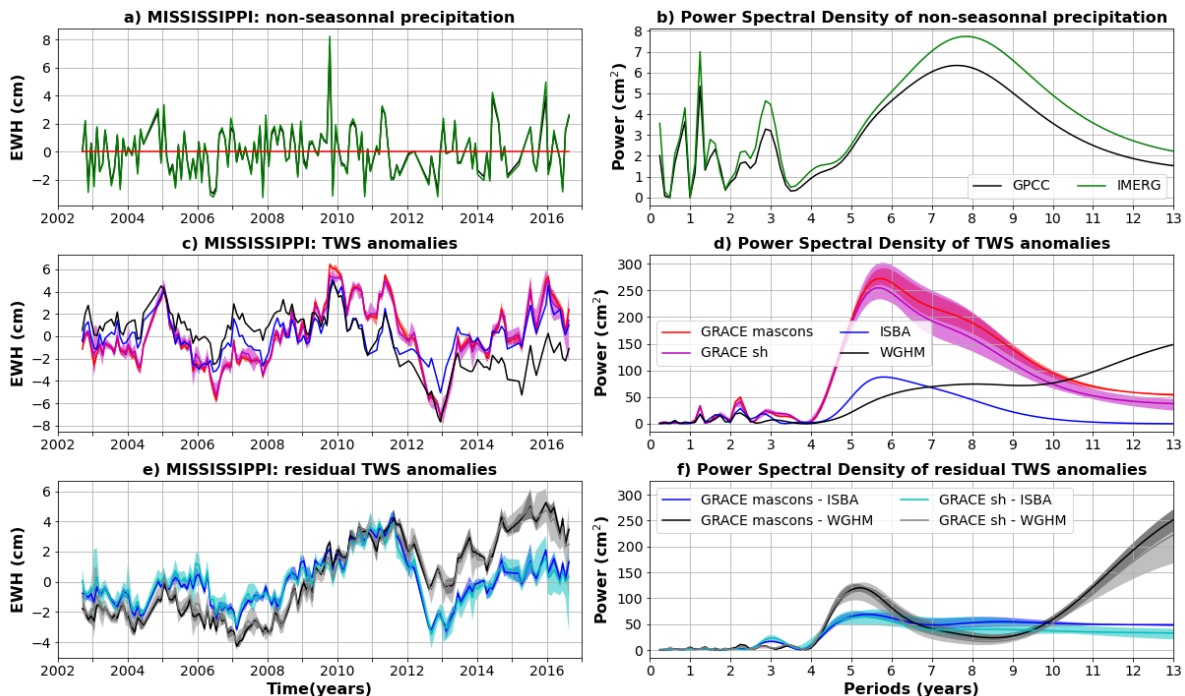

**Figure C20: Same as C2 for the Mississippi basin.**



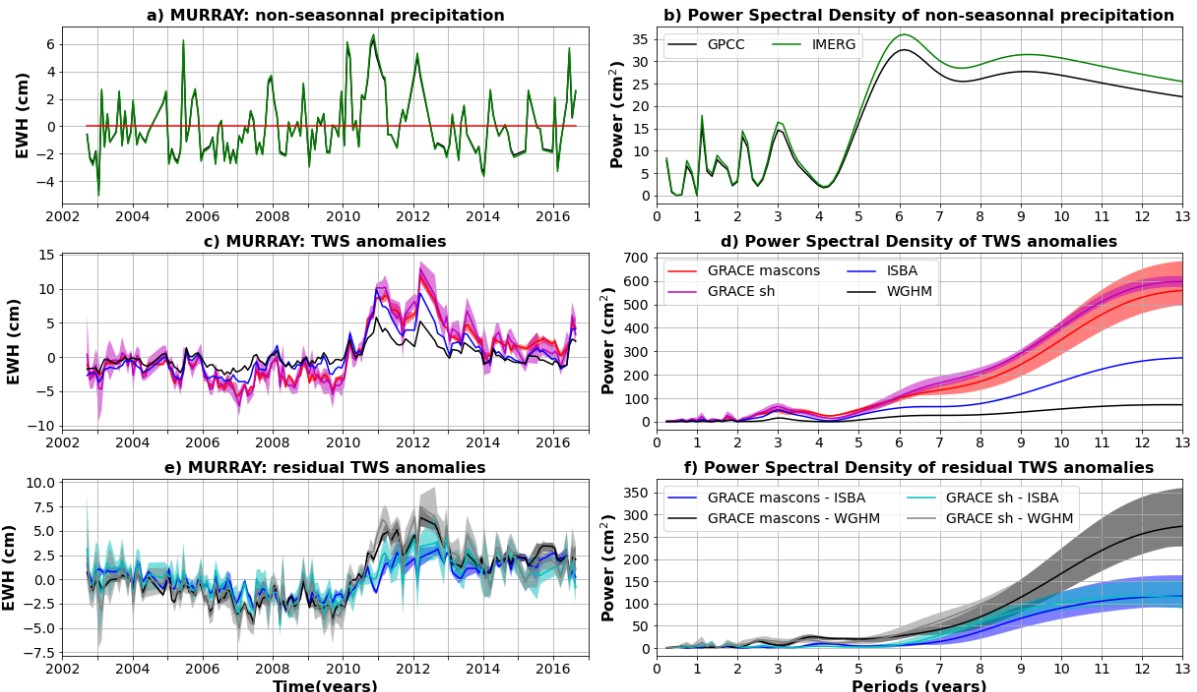


**Figure C21: Same as C2 for the Murray basin.**




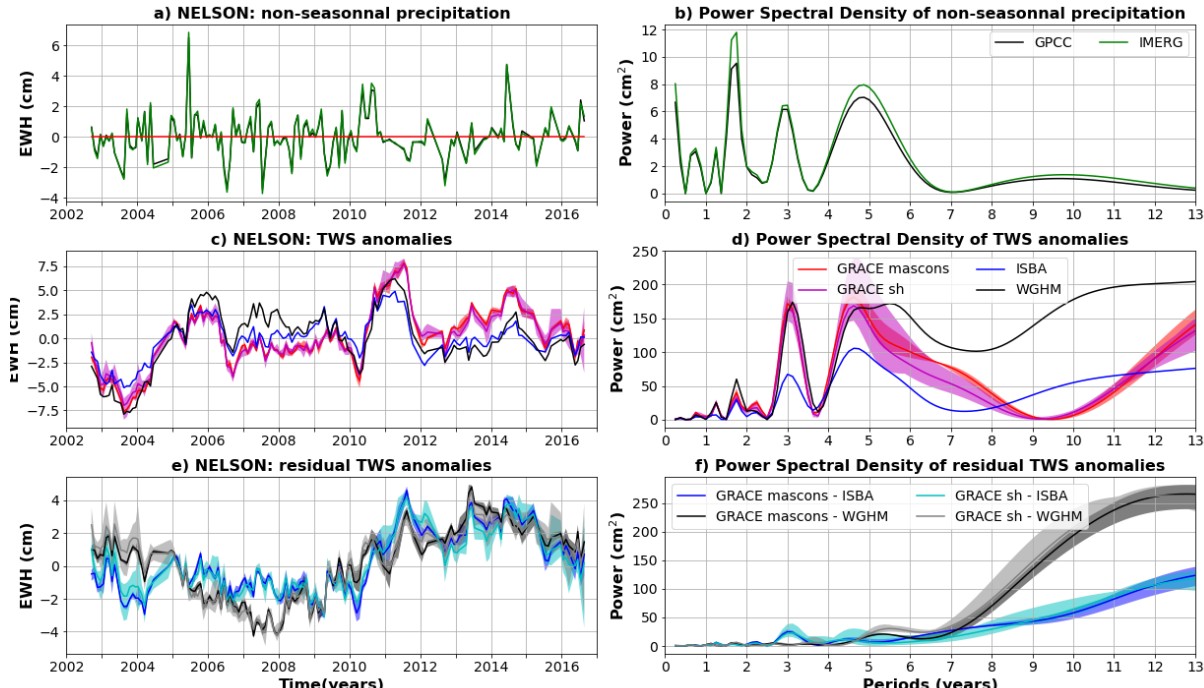

**Figure C22: Same as C2 for the Nelson basin.**






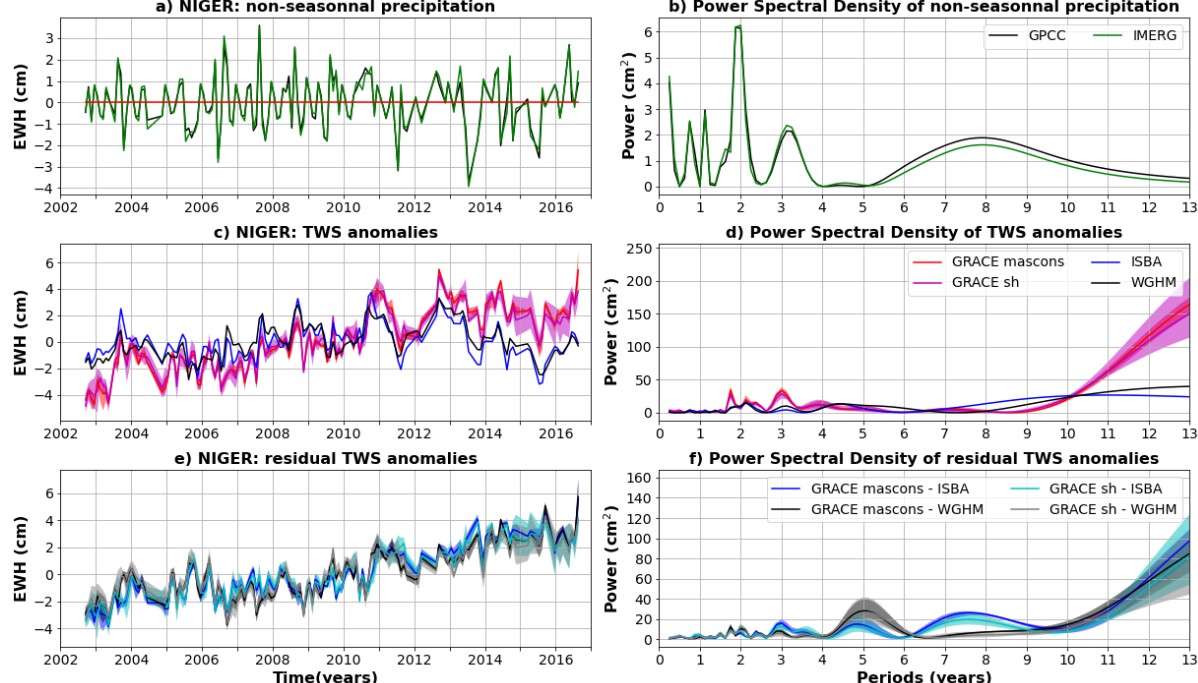

**Figure C23: Same as C2 for the Niger basin.**



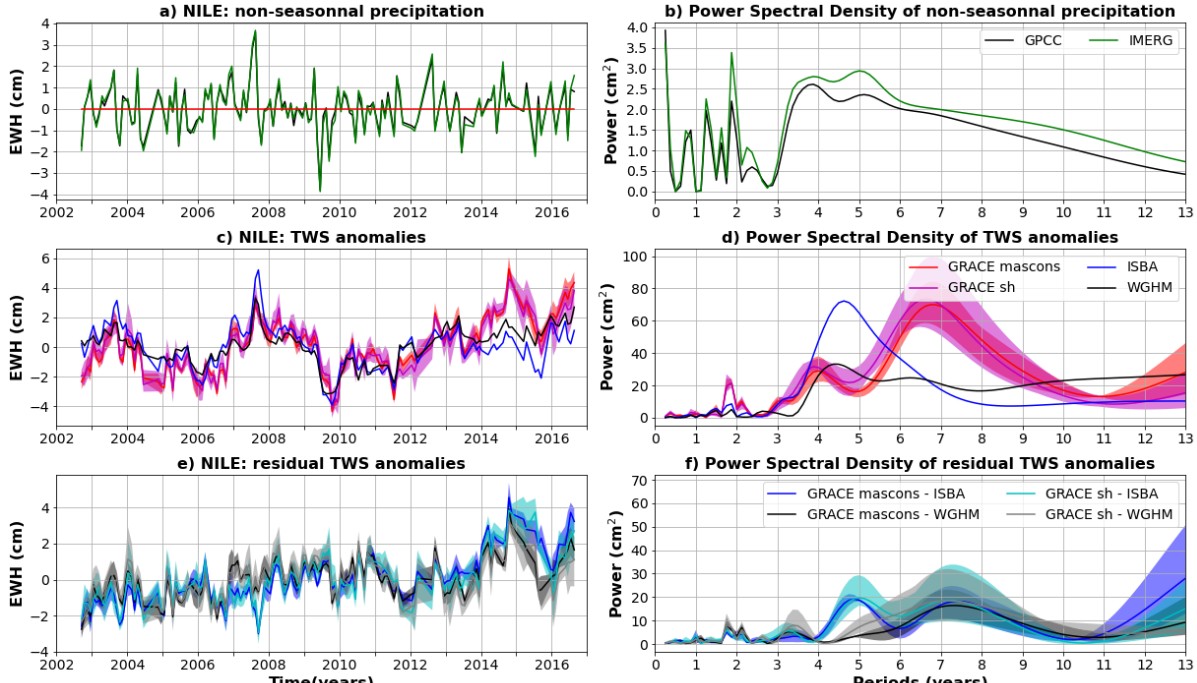

Figure C24: Same as C2 for the Nile basin.





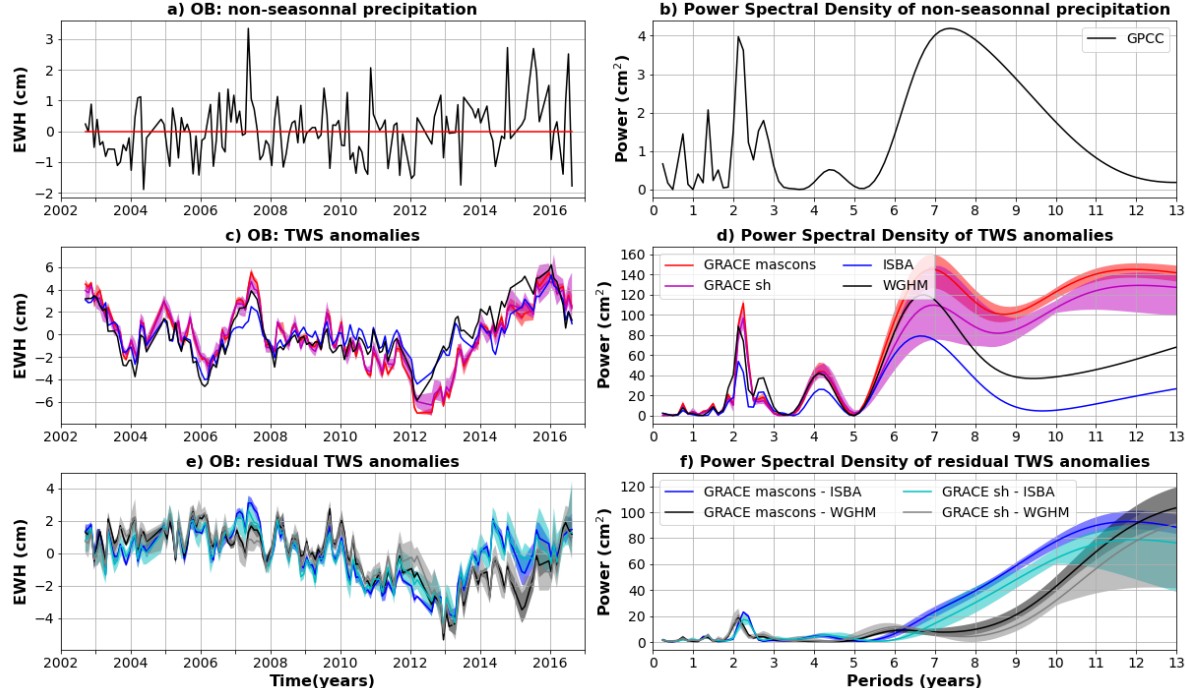

**Figure C25: Same as C2 for the Ob basin. Non-seasonal precipitation anomalies are only estimated with GPCC, as a significant part of the river basin is not covered by IMERG satellites due to its high latitude.**




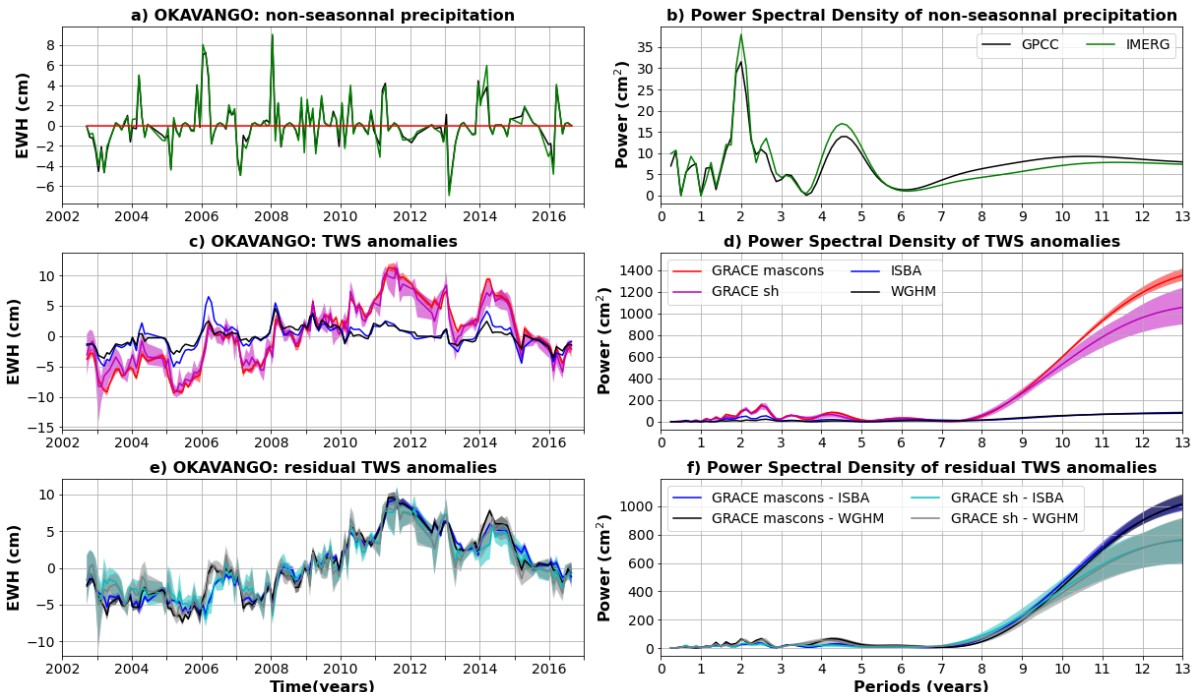

**Figure C26: Same as C2 for the Okavango basin.**



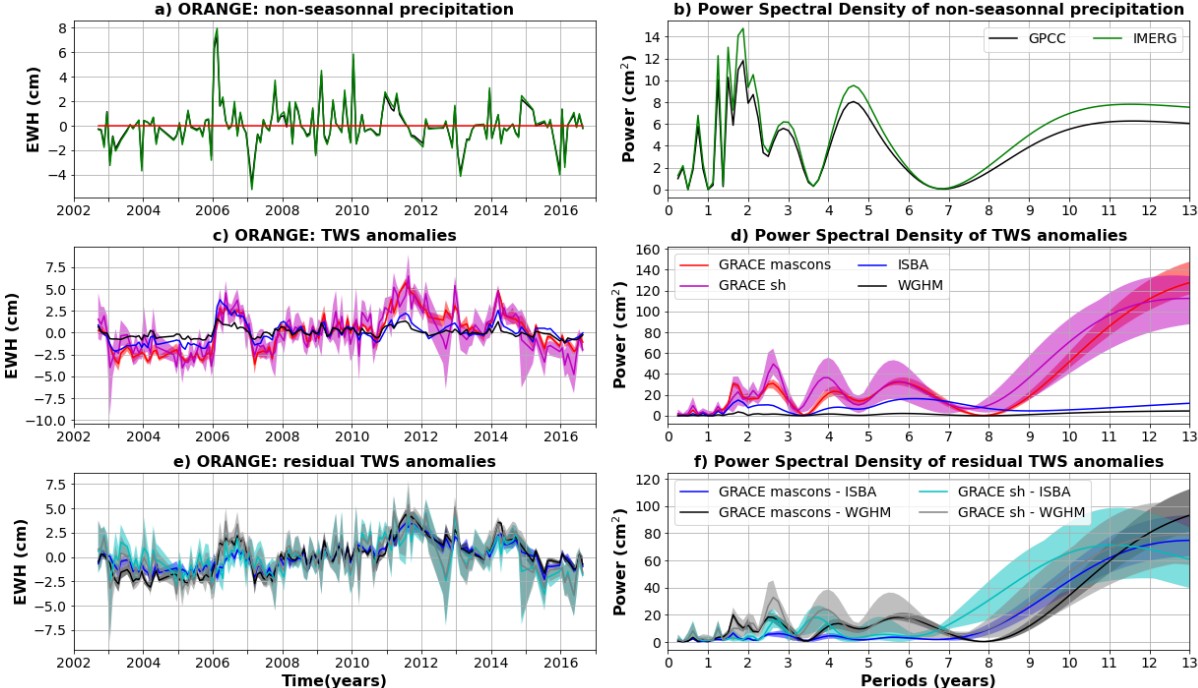

**Figure C27: Same as C2 for the Orange basin.**





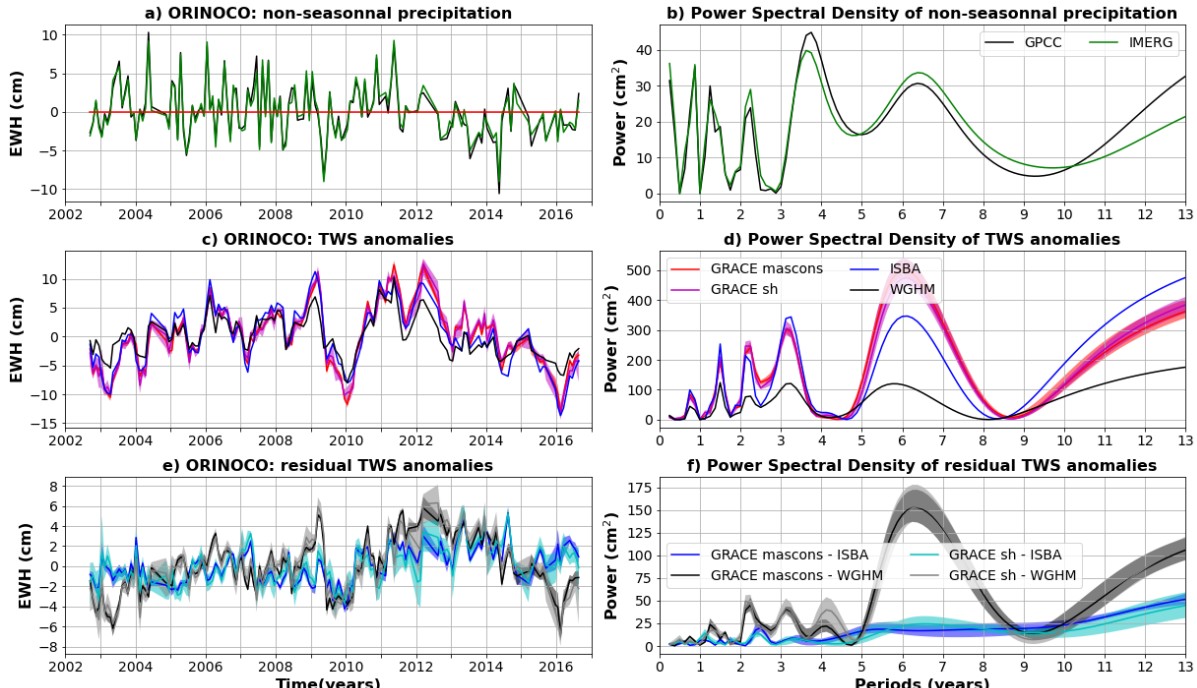

**Figure C28: Same as C2 for the Orinoco basin.**



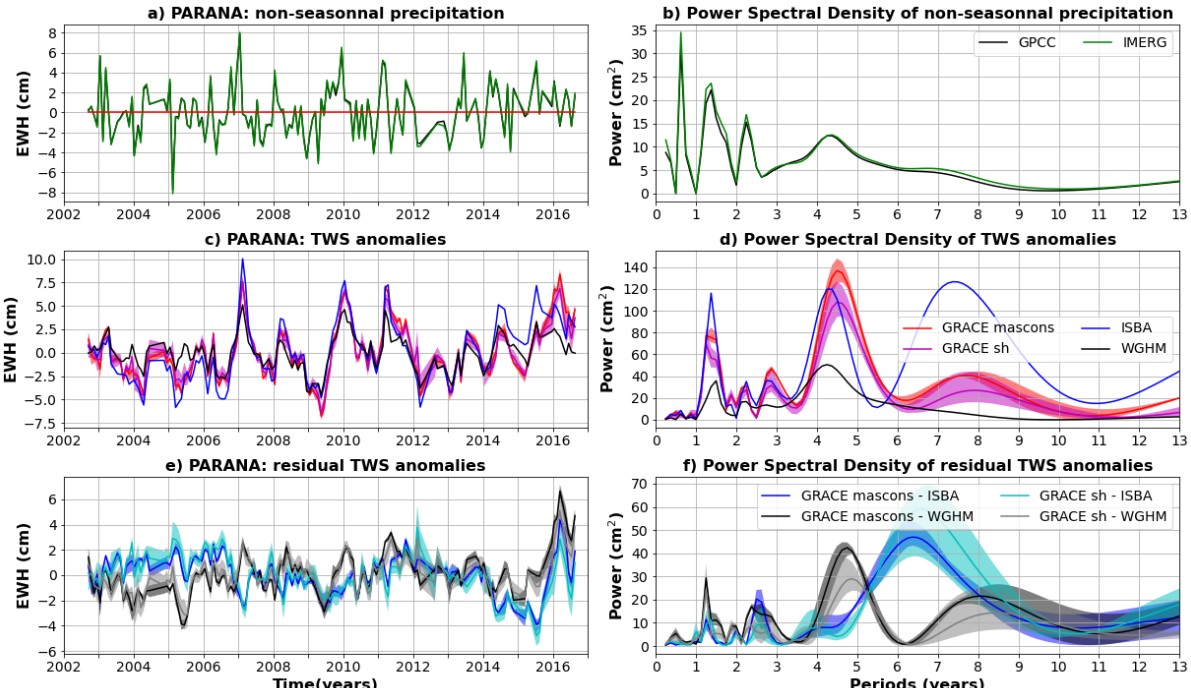


**Figure C29: Same as C2 for the Parana basin**.



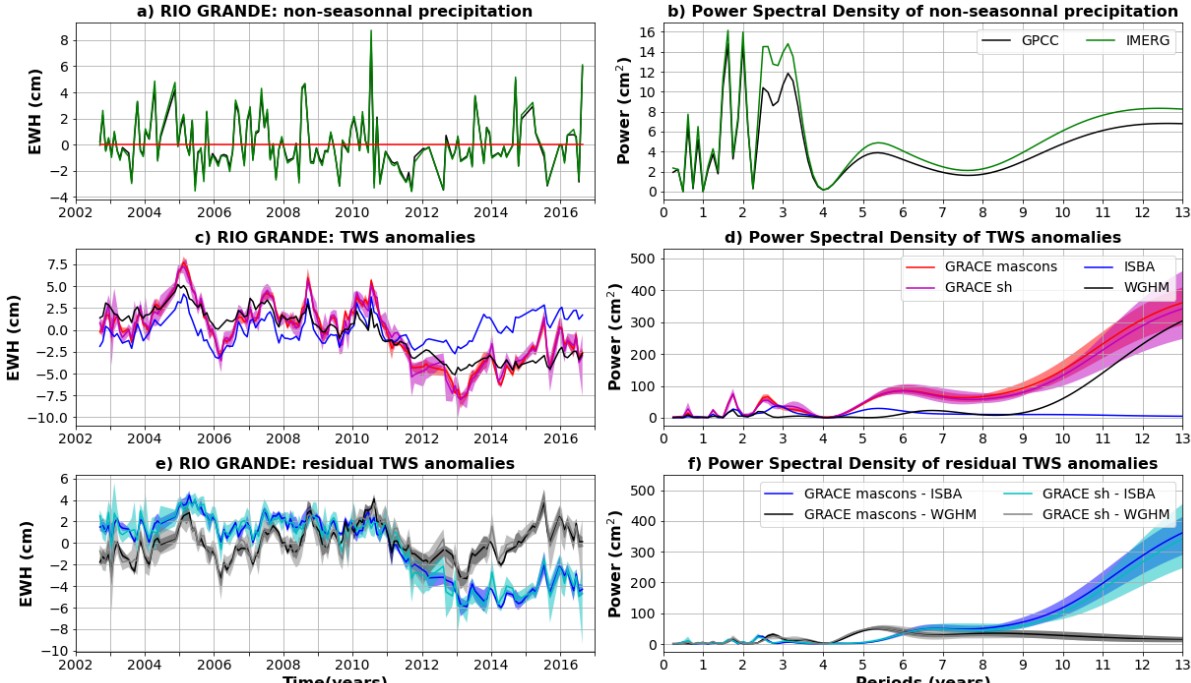

**Figure C30: Same as C2 for the Rio Grande basin.**




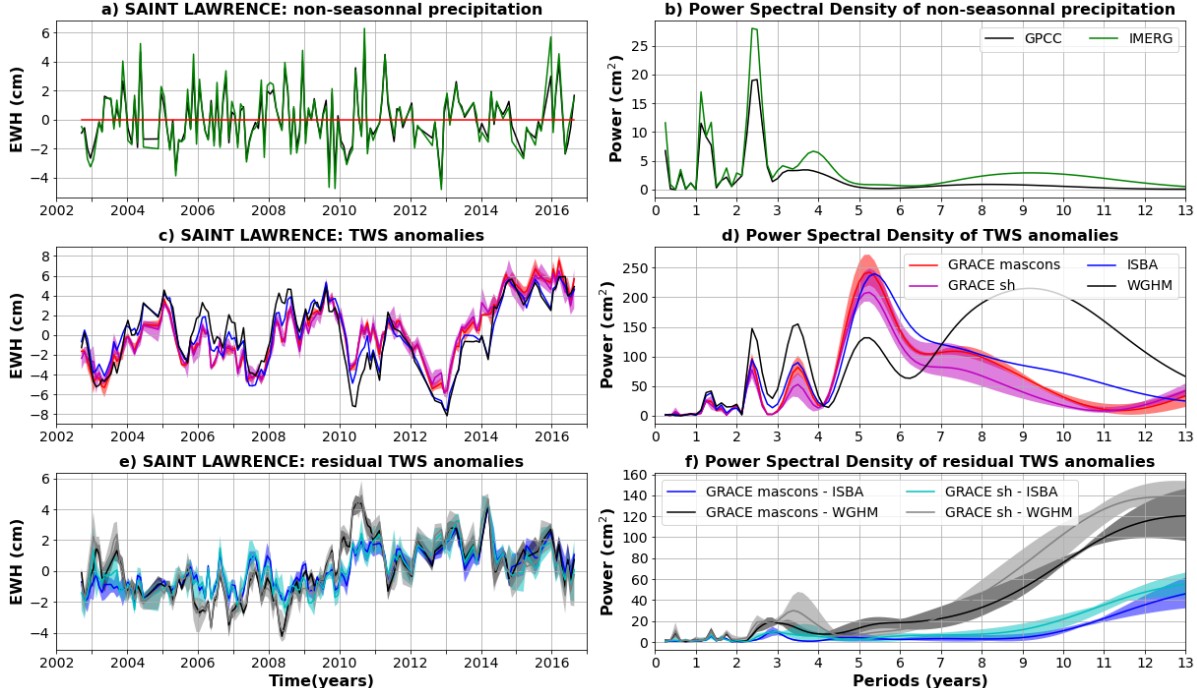

**Figure C31: Same as C2 for the Saint Lawrence basin.**



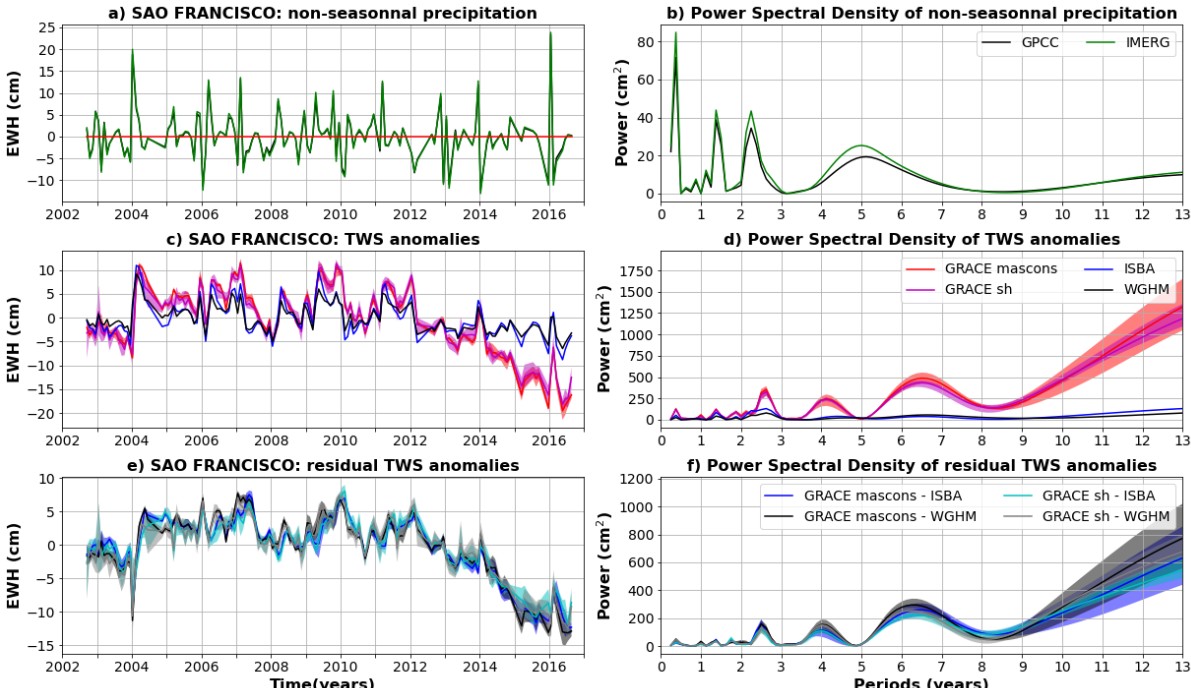

**Figure C32: Same as C2 for the Sao Francisco basin.**



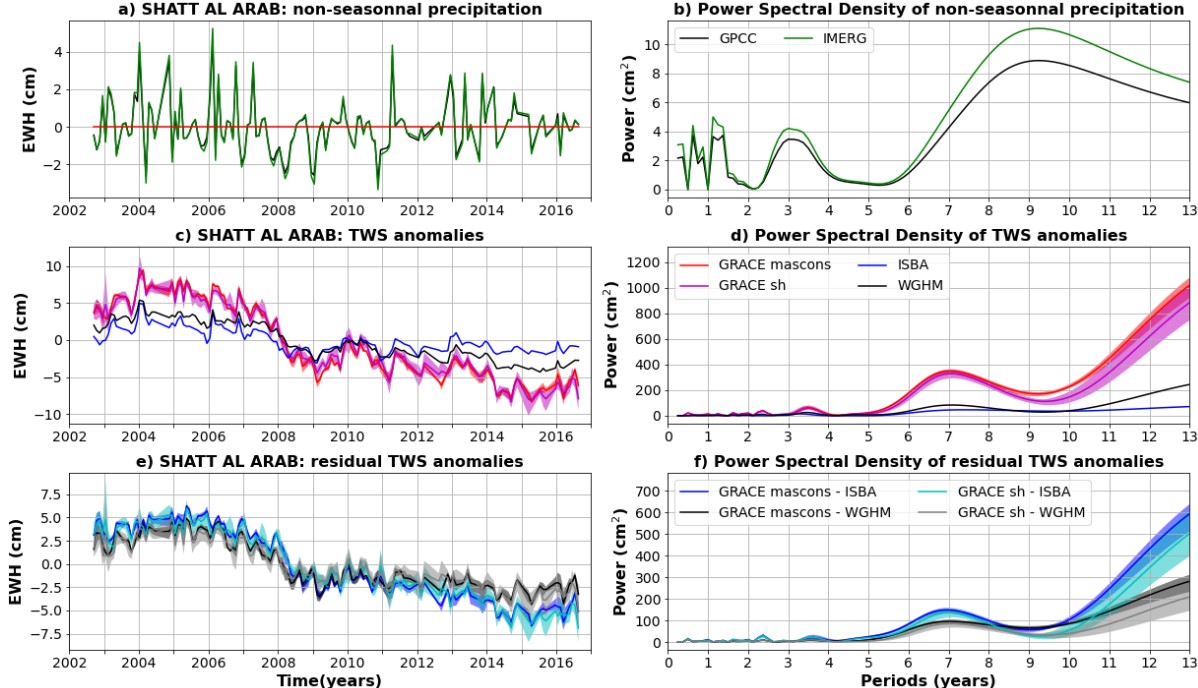

**Figure C33: Same as C2 for the Shatt al Arab basin.**






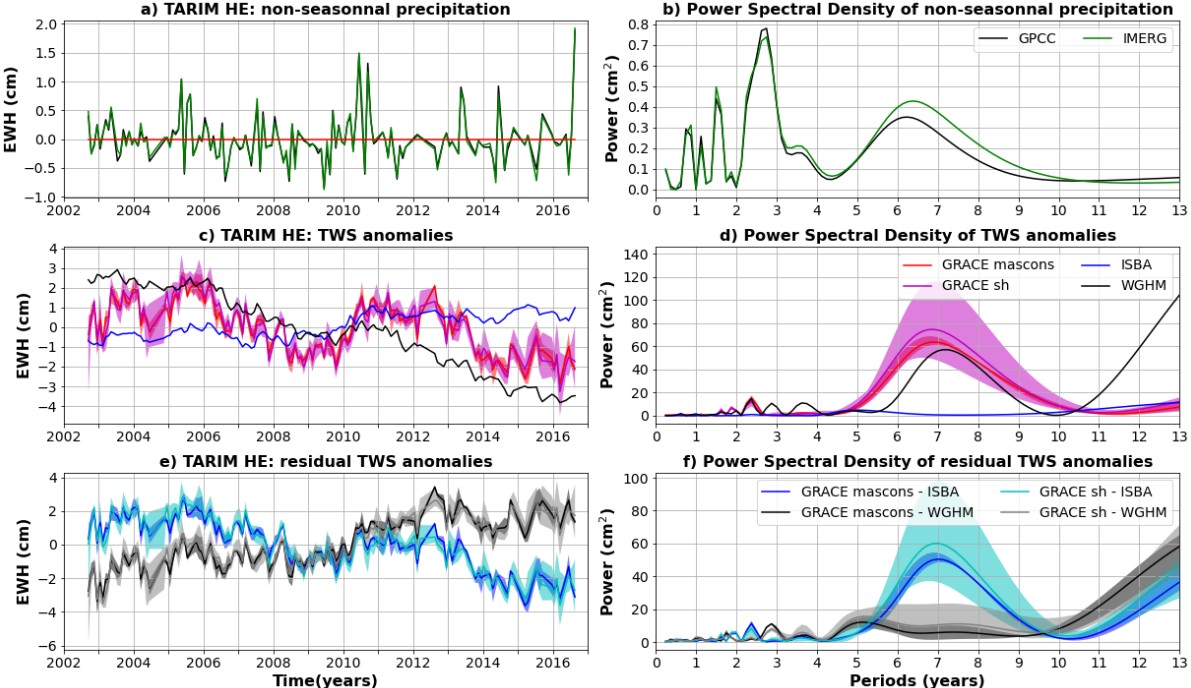

**Figure C34: Same as C2 for the Tarim He basin.**





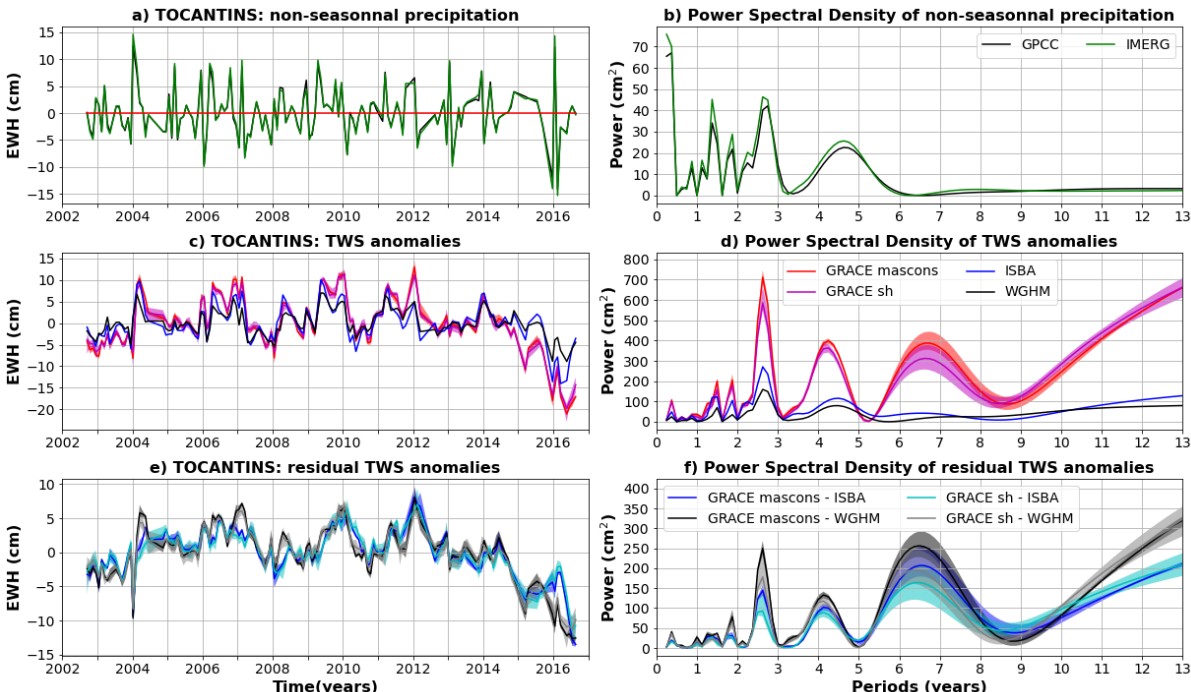

**Figure C35: Same as C2 for the Tocantins basin.**





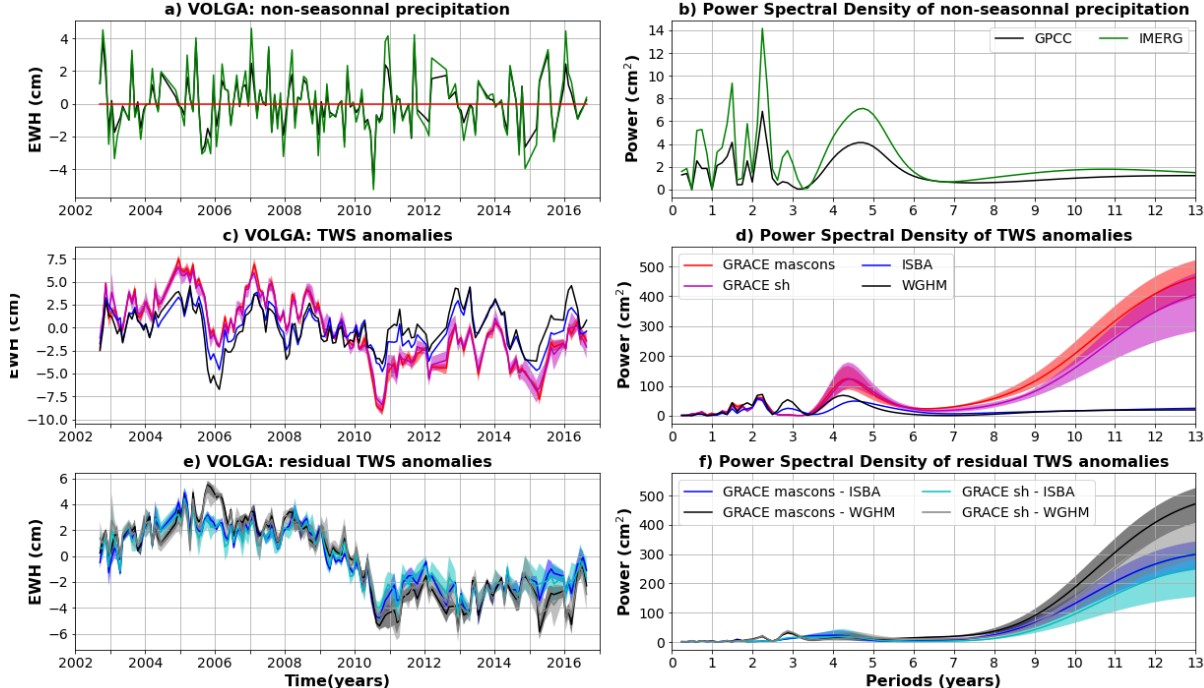

**Figure C36: Same as C2 for the Volga basin**.






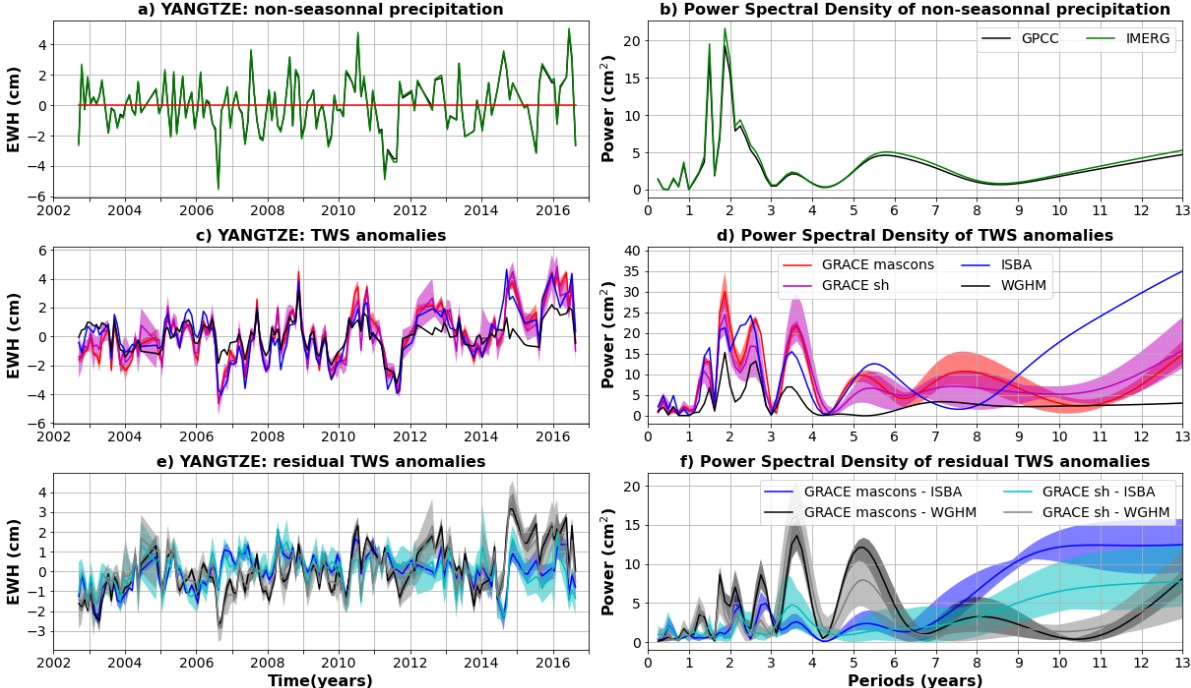

**Figure C37:** Same as C2 for the Yangtze basin.





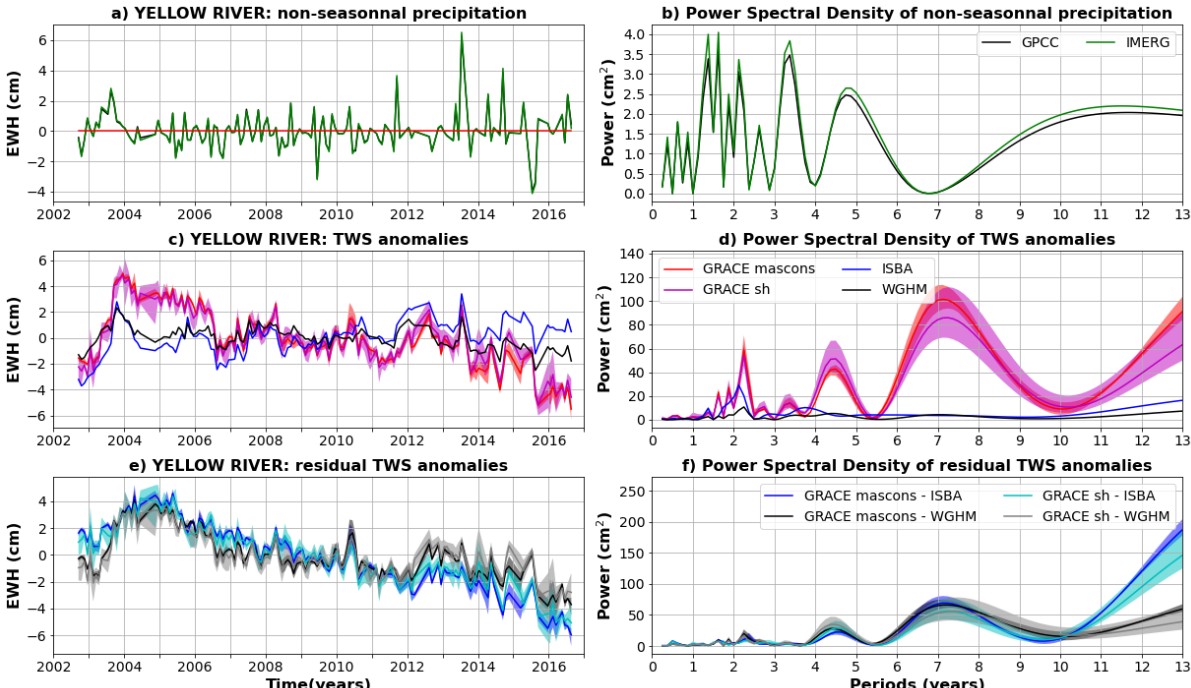

**Figure C38: Same as C2 for the Yellow River basin.**



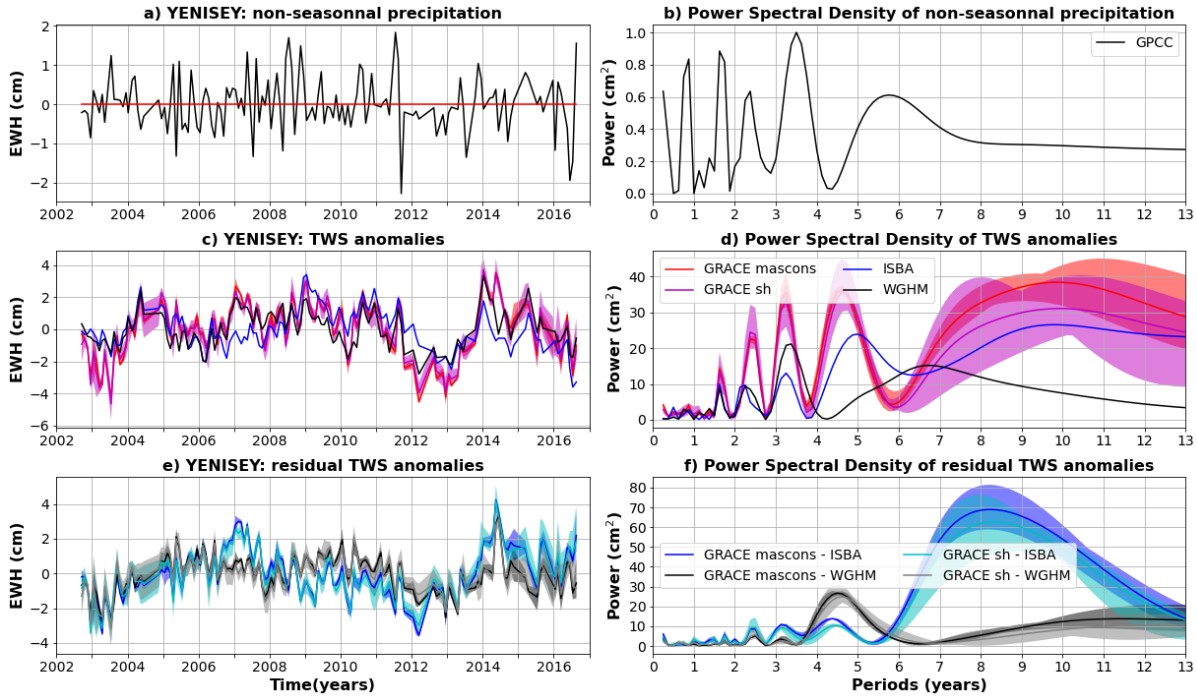

**Figure C39: Same as C2 for the Yenisei basin. Non-seasonal precipitation anomalies are only estimated with GPCC, as a significant part of the river basin is not covered by IMERG satellites due to its high latitude.**




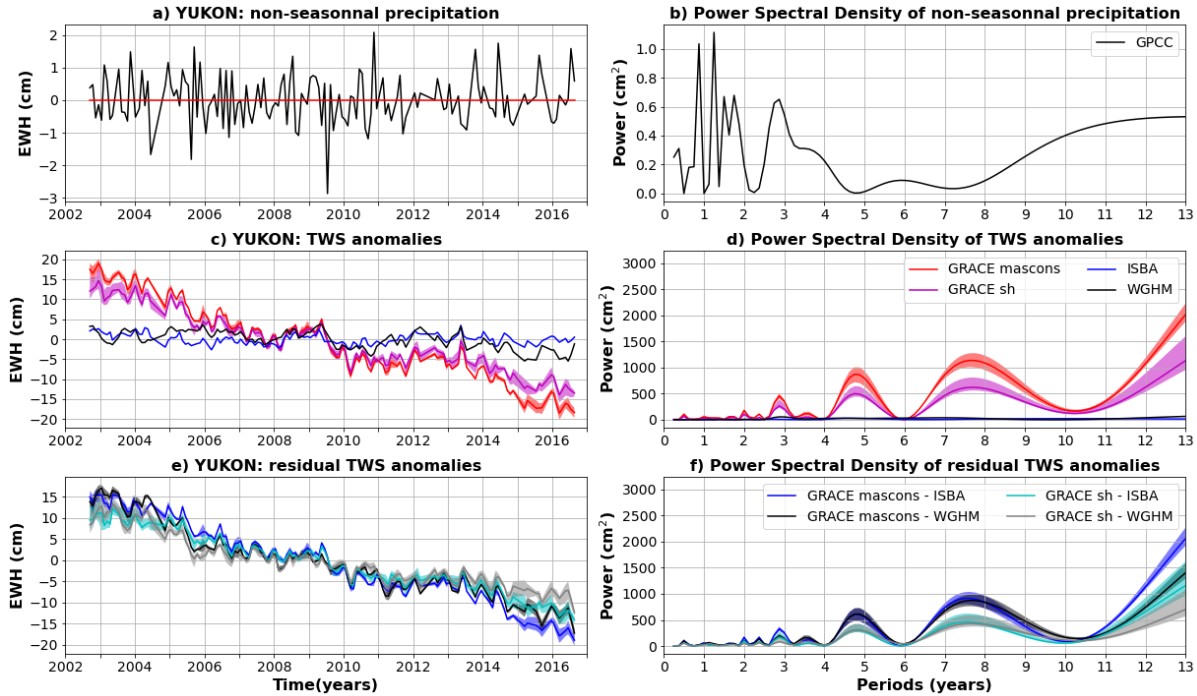

**Figure C40: Same as C2 for the Yukon basin. Non-seasonal precipitation anomalies are only estimated with GPCC, as a significant part of the river basin is not covered by IMERG satellites due to its high latitude.**



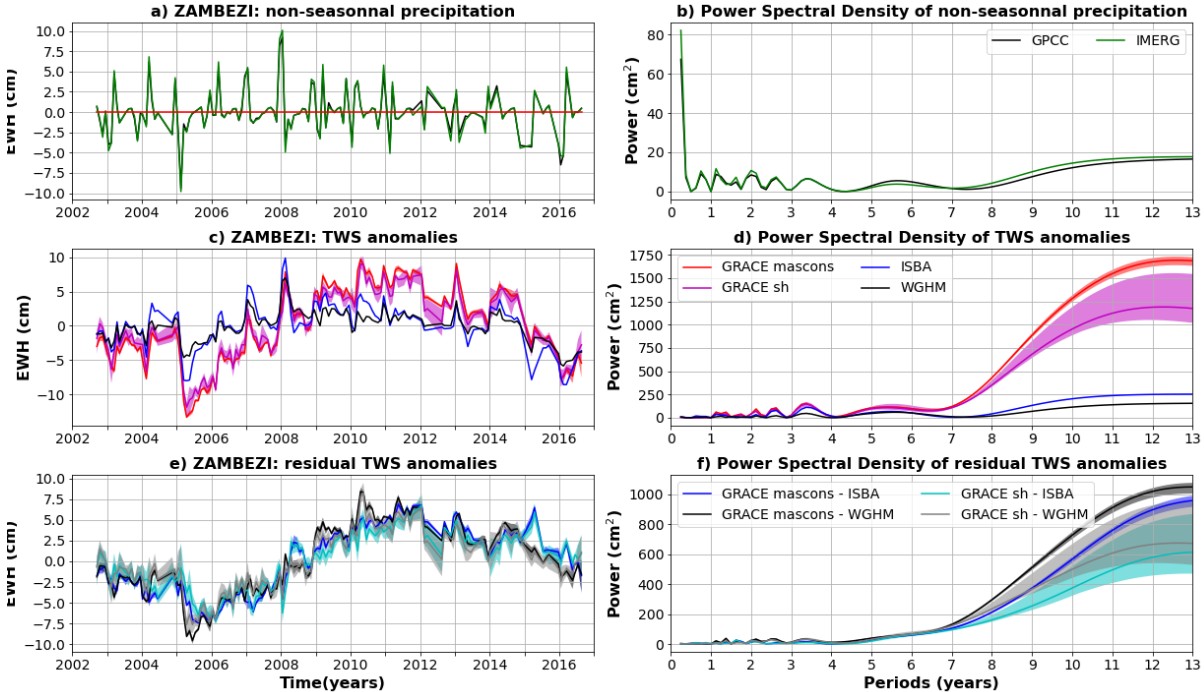

**Figure C41: Same as C2 for the Zambezi basin**



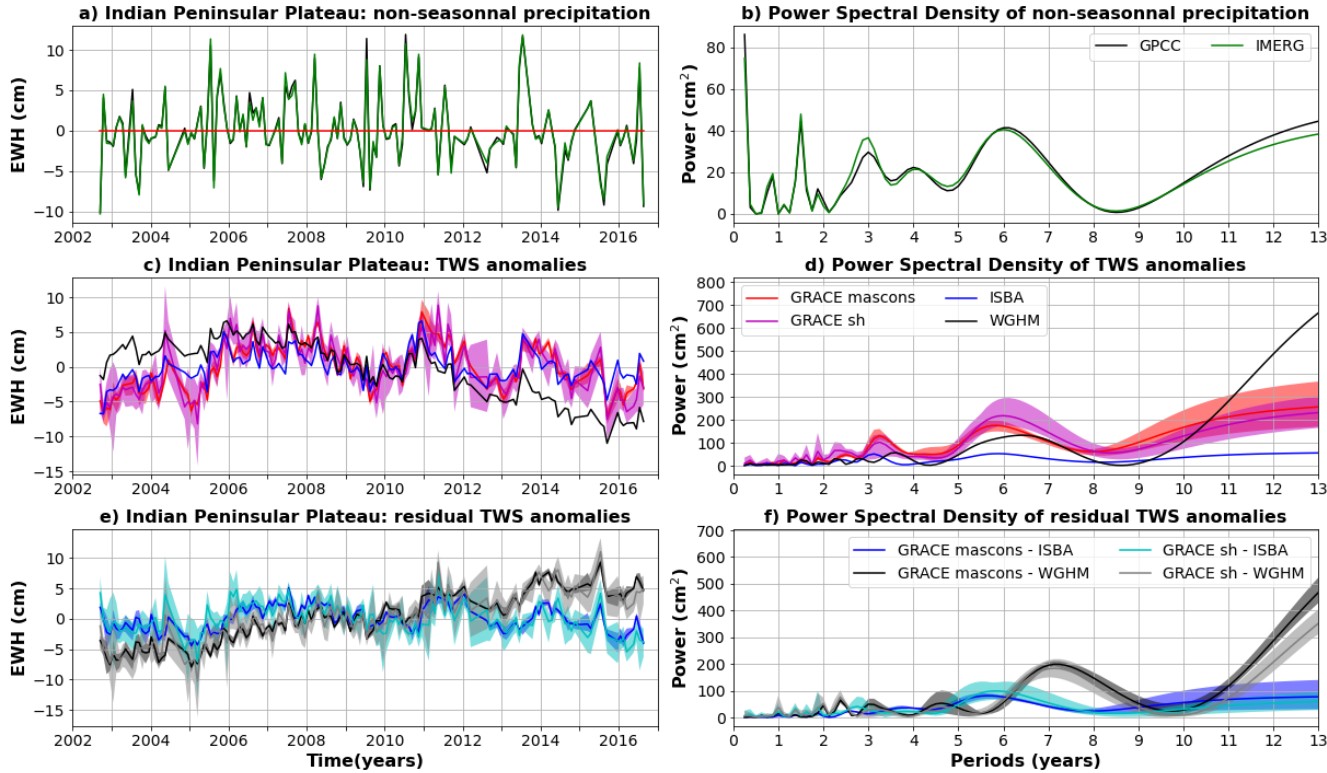

**Figure D1 Comparison of TWS and precipitation anomalies averaged across the Indian Peninsular Plateau (latitudes**
**7 -23°N; longitudes 70-80°E). a) Average precipitation anomalies for the GPCC (gauge-based) and IMERG (satellite-based) products. b) Power Spectral Density (PSD) of average precipitation anomalies. c) TWS anomalies average over the central Amazon for two global hydrological models (ISBA-CTRIP in blue and WGHM in black) and 9 GRACE solutions (mascons in red, spherical harmonic in magenta). The solid line corresponds to the average of the sub-ensemble, the shaded area to the minimum to maximum envelope. d) PSD of the averaged TWS anomalies shown in**
**(c). e) Residual TWS anomalies averaged over the central Amazon corridor and calculated as the difference between GRACE and ISBA-CTRIP (blue when the difference is calculated with mascons, cyan with spherical harmonics) or WGHM (black when the difference is calculated with mascons, grey with spherical harmonics).**