# Peer review of "Assessment of pluriannual and decadal changes in terrestrial water"

_EGUsphere, 2022_

## Author Comment (AC3)

[Figure]

**Figure S1: Comparison of TWS anomalies estimated from an ensemble of three GRACE mascon solutions and two global hydrological models. The amplitude of the non-seasonal TWS variability is expressed as the range at 95% CL, calculated as the difference between the 97.5 and 2.5 percentiles of the TWS anomalies obtained in each grid cell over the entire study period. a) Range of TWS anomalies estimated as the average of three GRACE mascons solutions. b) Dispersion of the range of TWS anomalies among three GRACE mascons solutions. Range of TWS anomalies estimated with ISBA-CTRIP (c) and WGHM (d). Range of residual TWS anomalies estimated as the difference between the average of three GRACE mascon solutions and ISBA-CTRIP (e) or WGHM (f).**

[Figure]

**Figure S2: Comparison of TWS anomalies estimated from an ensemble of six GRACE spherical harmonic solutions and two global hydrological models. The amplitude of the non-seasonal TWS variability is expressed as the range at 95% CL, calculated as the difference between the 97.5 and 2.5 percentiles of the TWS anomalies obtained in each grid cell over the entire study period.  a) Range of TWS anomalies estimated as the average of six GRACE spherical harmonics solutions. b) Dispersion of the range of TWS anomalies among six GRACE spherical harmonics solutions. Range of TWS anomalies estimated with ISBA-CTRIP (c) and WGHM (d). Range of residual TWS anomalies estimated as the difference between the average of six GRACE spherical harmonic solutions and ISBA-CTRIP (e) or WGHM (f).**

[Figure]

**Figure S3: Range ratios between the average of three GRACE mascon solutions and the hydrological models ISBA-CTRIP (a) and WGHM (b). Determination coefficients between the average GRACE mascon solution and the hydrological models ISBA-CTRIP (c) and WGHM (d). Regions, where the coefficient of determination is negative, are shown in white**

[Figure]

**Figure S4: Range ratios between the average of six GRACE spherical harmonic solutions and the hydrological models ISBA-CTRIP (a) and WGHM (b). Determination coefficients between the average GRACE spherical harmonic solution and the hydrological models ISBA-CTRIP (c) and WGHM (d). Regions, where the coefficient of determination is negative, are shown in white**

**a) Contribution of subannual, pluri-annual and decadal signals in residual TWS anomalies calculated as the difference between mascons and ISBA**

[Figure]

**b) Contribution of subannual, pluri-annual and decadal signals in residual TWS anomalies calculated as the difference between mascons and WGHM**

[Figure]

**Figure S5: Characteristic time scales in residual TWS anomalies calculated as the differences between the average of three GRACE mascon solutions and ISBA-CTRIP (a) or WGHM (b). Subannual, pluriannual and decadal contributions have been computed with high-pass (cut-off period at 1.5 years), band-pass (cut-off periods at 1.5 and 10 years) and low-pass (cut-off period at 10 years) filters respectively. The percentage of variance explained by one contribution has been calculated as the coefficient of determination with respect to the full residual signal.**

**a) Contribution of subannual, pluri-annual and decadal signals in residual TWS anomalies calculated as the difference between sh and ISBA**

[Figure]

**b) Contribution of subannual, pluri-annual and decadal signals in residual TWS anomalies calculated as the difference between sh and WGHM**

[Figure]

**Figure S6: Characteristic time scales in residual TWS anomalies calculated as the differences between the average of six GRACE spherical harmonic solutions and ISBA-CTRIP (a) or WGHM (b). Subannual, pluriannual and decadal contributions have been computed with high-pass (cut-off period at 1.5 years), band-pass (cut-off periods at 1.5 and 10 years) and low-pass (cut-off period at 10 years) filters respectively. The percentage of variance explained by one contribution has been calculated as the coefficient of determination with respect to the full residual signal.**

[Figure]

**Figure S7: a)** Linear trends in residual TWS anomalies calculated as the difference between the average of three GRACE mascon solutions and ISBA-CTRIP. **b)** Same as (a) with WGHM. **c)** Amplitude of non-linear signals in residual TWS anomalies calculated as the difference between the average of three GRACE mascon solutions and ISBA-CTRIP. The amplitude is calculated as the difference between the 97.5 and 2.5 percentiles. **d)** Same as (c) with WGHM. **e)** Coefficient of determination calculated for non-linear signals with respect to TWS anomalies calculated as the difference between the average GRACE mascon solution and ISBA-CTRIP. **f)** Same as (e) with WGHM.

[Figure]

**Figure S8: a)** Linear trends in residual TWS anomalies calculated as the difference between the average of six GRACE spherical harmonic solutions and ISBA-CTRIP. **b)** Same as (a) with WGHM. **c)** Amplitude of non-linear signals in residual TWS anomalies calculated as the difference between the average of six GRACE spherical harmonic solutions and ISBA-CTRIP. The amplitude is calculated as the difference between the 97.5 and 2.5 percentiles. **d)** Same as (c) with WGHM. **e)** Coefficient of determination calculated for non-linear signals with respect to TWS anomalies calculated as the difference between the average GRACE mascon solution and ISBA-CTRIP. **f)** Same as (e) with WGHM.

---

## Author Response (AR1)

RC1

This study focuses on the analysis of interannual and decadal (slow changes) terrestrial water storage – TWS from GRACE solutions and two global hydrological models. The main contribution of this study is the analysis of slow changes in areas where interannual and decadal anomalies are dominating, and how are compared with the two global hydrological models.

I found an interesting read, very well written, however, I have some few observations below:

- Line 92-93: Since in the title and in the abstract they mention GRACE and GRACE-FO, I expected the analysis to include the new data returned by GRACE-FO, but I see that it only covers the time period of the first GRACE mission between 2002 and 2016. GRACE-FO was launched in 2018

**We were limited in our comparison study by the availability of WGHM data, only provided until December 2016. We only show results on the common data period (April 2002 - december 2016), therefore excluding the GRACE-FO mission. We removed any mention of GRACE-FO in the title and abstract.**

- Line 95: Do the two models used in the analysis belong to CMIP6 and ISI-MIP? If so, it should be clarified, since it is not very well understood why to compare GRACE with these two specific models.

**ISBA-CTRIP has been used in CMIP-6. WGHM has been used in ISI-MIP. This has been clarified in the text.**

- In figure 1, are the maps the average of the entire study period?

**In Fig. 1, we show the range of interannual TWS changes over the entire study period. The range at 95% CL (confidence limit) is calculated as the difference between the 97.5 and 2.5 percentiles of the TWS anomalies estimated in each grid cell over the entire study period. This allows removing outliers, and provides a good estimation of the amplitude of interannual variations of TWS anomalies in various regions of the globe. This is explained at the end of section 2.5. It has been added in the legend of Fig. 1, to avoid the reader going back and forth between the figures and the text.**

**We thank the reviewer for his/her comments, which helped improve the manuscript.**

RC2

This is an important study comparing the slow change in terrestrial water storage between GRACE solutions and two global hydrological models.

I am outlining below some major comments:

1. The term slow change is not defined clearly. Does it include the linear trend? I see from the results it includes the sub-annual component, and I don't think the sub-annual changes can be slow changes. I suggest formulating these definitions using equations. Also, it will be helpful to show some raw time series for some areas for the slow change before and after applying the diffusive filter.

**The term slow changes includes pluri-annual and decadal changes. The term "slow" has been replaced by "pluri-annual and decadal" at each use (including in the title, abstract and conclusions) for more accuracy.**

**We only removed seasonal signals, modelled as the sum of annual and semi-annual sinusoids whose phase and amplitude are estimated at each grid cell (section 2.5). Linear trends contribute significantly to the decadal variability in TWS, therefore we do not remove it (Fig. 4).**

**We determine the time scales of the residual TWS changes using high-pass (cutoff at 1.5 years), band-pass (cutoffs at 1.5 and 10 years) and lowpass (cutoff 10 years) filters, allowing to extract sub-annual, pluri-annual and decadal signals respectively (section 3). We show that TWS residuals are dominated by pluri-annual and decadal signals (Fig. 3).**

**We also compute and display the power spectral density associated with TWS time-series estimated with GRACE and global hydrological models over numerous regions of the globe (Fig. 5 to 12, Appendix C). The comparison of power spectral densities (PSD) shows that TWS changes are systematically underestimated at pluri-annual and decadal time scales by global hydrological models when compared to GRACE (PSD are not normalised).**

2. Authors did a good job in combining multiple GRACE data, but I feel these efforts wasted by just using the ensemble of all data, also I don't think averaging the mascons and solutions together is a good strategy.

**All GRACE solutions are remarkably consistent one with another, which is evidenced by the small dispersion between the solutions (Fig 1b). The differences between the average GRACE solution and global hydrological models (Fig. 1e and f) are much larger than the dispersion between the solutions (Fig 1b, section 3.1). As suggested by the reviewer, we compared the mascon and spherical harmonic sub-ensembles to global hydrological models separately and found remarkably consistent results (supplementary material). We also added a couple of sentences in section 3.1 to make that clearer.**

**The manuscript is already very long, so we do not wish to add to it. Therefore, we only show the average GRACE-based solution in Fig 1. to 4. The same analyses carried out with mascons and spherical harmonic separately are available in supplementary material. We show the difference between mascons (red) and spherical harmonic (magenta) solutions for the regional analyses (Fig. 5 to 12, Appendix C).**

3. Are all GRACE solutions have the same performance for the slow changes in TWS?

**Yes. See point 2.**

4. It will be more informative to have two ensembles; one for the mascons and another for the spherical harmonics, and compare their performance.

**Both sub-ensembles (mascons and spherical harmonics) are remarkably similar (see point 2). Extremely similar results are found when comparing global hydrological models with mascons and spherical harmonics separately (supplementary material).**

5. Paragraph #30. "located at the Earth's surface", not correct phrasing

**We replaced "mass anomalies as a layer of water of variable thickness in space and time located at the Earth's surface" by "changes in surface density (i.e. changes in mass per unit area) as a layer of water of variable thickness in space and time" to be closer to the definition of Wahr et al., (1998) and Ditmar (2018)**

6. Paragraph #35. Do you mean seasonal components and the trend? Or by the decadal change, the author meant the linear trend.

**In the study cited here (Humphrey et al., 2016), decadal changes are the long term changes including both linear trends and interannual changes. This has been rephrased and clarified in the revised version of the manuscript.**

7. Paragraph #46. " Multidecadal Atlantic oscillations" è Atlantic multidecadal oscillations

**Corrected**

8. Since the two hydrological models do not simulate the glaciers storage; comparing them with GRACE data is not fair; and so, the results over the glaciers regions. One suggestion for the authors is to remove the linear trend and limit the comparison to the interannual and decadal fluctuations.

**We agree that we cannot compare GRACE with hydrological models around glaciers. We indicate the limits of glaciers estimated with the sixth version of the Randolph Glacier Inventory by white contours in the revised version of Fig. 1. We indicate both in the text and figure legend that GRACE should not be compared with global hydrological models around glaciers.**

**However, linear trends constitute a significant part of decadal TWS variability, contributing to the biases observed between GRACE and global hydrological models. Therefore, this component should remain in the present assessment study.**

9.  Is the amplitude defined here as the max(TWS) – min (TWS)? if so, please define it clearly in the methods.

**The amplitude is defined as the range at 95% CL (confidence limit), calculated as the difference between the 97.5 and 2.5 percentiles of the TWS anomalies estimated in each grid cell. This allows an accurate estimation of the amplitude of non-seasonal TWS anomalies, avoiding extreme values. This is defined at the end of the method section 2.5. For more clarity, we also included this definition in the legend of Fig. 1.**

10. One suggestion on the discussion sections; please start with results for these areas, and then discuss their hydrological characteristics.

**We divided the discussion on each region in two subsections: i) study area; ii) comparison of global hydrological models with GRACE. This allows a clearer separation between the hydrological context and our results.**

**We thank the reviewer for his/her comments, which helped improve the manuscript.**

---

## Author Response (AR2)

Generally, the contribution treats a well known topic: the shortcoming of hydrological models in representing long term trends and decadal changes of TWS. This is a relevant topic also in the scope of better understanding the occurrence of extreme events. This contribution goes beyond the investigations of Scanlon et al. (2018) (https://www.pnas.org/doi/10.1073/pnas.1704665115), in particular by accomplishing detailed regional analyses and by looking not only at trends but also at decadal changes. Unfortunately, the study is limited to the time period 2003 to 2016 because of the availability of WGHM data. Due to this 6 years of GRACE/GRACE-FO data cannot be considered, which is quite a significant part of the 20 year time span. With respect to the title and the focus of the paper, I wonder whether it would not make sense to look at some other hydrological models (e.g. the GLDAS models also evaluated by Scanlon et al. (2018)) in order to exploit the whole GRACE time span and to give more value to the message that should be conveyed. However, please explain why you decided to focus your study on WGHM and ISBA.

**We thank the Reviewer 3 for his/her comments, that helped considerably improve the manuscript. We focused on global hydrological models rather than land surface models, allowing a more detailed representation of hydrological processes across continental areas. In particular, land surface models of the Global Land Data Assimilation System (GLDAS) such as VIC or NOAH , only take into account vertical fluxes and do not explicitly represent surface water storage and aquifers. While land surface models, such as NOAH, present the undeniable advantage of availability in near real time, allowing a longer period overlap with GRACE and GRACE-FO missions, their accuracy is lesser than those of global hydrological models such as ISBA-CTRIP or WGHM. We added the justification of our choice in the introduction (L76-82). We also provide in supplementary material the comparison of GRACE-based and NOAH-based TWS, which confirms the lower accuracy (larger residuals and lower $R^2$ over most continental areas) of NOAH in comparison to WGHM or ISBA-CTRIP.**

Overall, the manuscript is well written and easy to follow. In particular the part on the regional analyses is extremely interesting and provides new insights into possible reasons for shortcomings of the models. In some parts information should be more concise as commented below.

**Changes to the manuscript have been made as follows.**

Abstract:
- l14: changes with respect to what? **"with respect to the temporal average" added**
- l19: well correlated (please be more concise), how can differences in TWS be correlated with precipitation? **Rephrased as "consistent with precipitation" (i.e. a drop (rise) in precipitation is consistent with a drop (rise) in TWS)**

- l21: you should mention that this issue is well known and has already be investigated a lot. **This is discussed in the introduction at L54-69.**

Introduction:
- l. 54 onwards: I do not understand why this part about the water mass budget is relevant for this paper. **The paragraph has been removed from the introduction.**

2 Methods:
-l60: why do you truncate at degree 60? There is still some signal contained in the higher degree coefficients. **The signal contained in the higher degree coefficients is overpowered by noise, requiring specific mitigation only available in dedicated filters.**
-l157: do you average the three products? Please be more specific how you "estimate" precipitation. **We use two distinct precipitation products: GPCC and IMERG. GPCC is based on rain gauges measurements. IMERG is based on TRMM and GPM satellite measurements. We rephrased for clarity. We apply the same processing to all time series, as detailed in section 2.5.**
-l178: please define CL at its first occurrence. **Done (CL stands for Confidence Level).**

3 Results / 4 Discussion
- Fig. 1: please consider a different colorbar for Fig 1b. In the other subplots red is larger than yellow/white. It is confusing for interpretation. **The colorbar has been changed for Fig. 1b.**
- l 207: is it possible that the model variability is damped too strongly by the filter that you applied? **We use the same filter (radius at 250 km) on all fields to be able to compare them.**
- Fig. 3: very interesting figure! How should sub-annual to decadal signals in the Sahara region be interpreted? **As no significant hydrological signal is expected in the Sahara, this may be interpreted as the noise signature of the TWS residuals, which contains a high and a low frequency component. A sentence has been added at L262-265.**
- l273: what kind of anthropogenic influences and which kind of climate variability? Please be more specific and cite relevant studies. **These two categories include a variety of processes that are discussed in section 4. Among anthropogenic influences, irrigation has the most impact on TWS changes. Among climate influences, precipitation (droughts/excess rainfall) has the most impact on TWS changes. These examples were added at L272-275. Relevant studies are abundantly cited in the discussion section.**
-l545: could the negative trend in the Black Sea Catchment be related to uncertainties in the water mass correction for lakes? **The lake correction is only added to WGHM and ISBA, not to GRACE. It cannot be responsible for the decreasing trend observed in GRACE-based TWS.**

5 Conclusion

-l587/589 parameterisation → parameterization. **Corrected.**

- l529: joint calibration against discharge and TWSA has been applied e.g. by Werth et al. 2009. **The reference has been added to the conclusion.**

- general: you could also indicate the potential benefit from GRACE data assimilation

**We added this information in the conclusion (L596-602).**

---

## Author Response (AR3)

Dear Authors, thank you for the revision of the manuscript, including the additional reflection considering GLDAS. Am proposing to take the manuscript forward for publication. In doing so, I would , however, raise the question on how to deal with the appendices, which are quite extensive. This makes the manuscript very long. Additional to the appendices there are also supplementary materials. To my understanding, including these as appendices in the main paper that is typeset, would mean that page charges will be applied. This could incur significant charges. As the paper is largely carried by the basins that are detailed in the discussion section, with the remaining basins mentioned in Appendix C. One could consider for example moving figures C2 through C41 to the supplementary materials as these are not explicitly referenced in the main manuscript nor the appendices. The decision is of course yours.

**We thank the editor for her advice. As suggested, we moved Appendices C and D to the supplementary materials.**

**Best regards,**
**Julia Pfeffer on behalf of all co-authors**